



# Long-term study on coarse mode aerosols in the Amazon rain forest with the frequent intrusion of Saharan dust plumes

Daniel Moran-Zuloaga[1], Florian Ditas[1], David Walter[1], Jorge Saturno[1], Joel Brito[2,a], Samara Carbone[2,b], Xuguang Chi[1,c], Isabella Hrabě de Angelis[1], Holger Baars[3], Ricardo H. M. Godoi[4], Birgit Heese[3], Bruna A. Holanda[1], Jošt V. Lavrič[5], Scot T. Martin[6,7], Jing Ming[1], Mira L. Pöhlker[1], Nina Ruckteschler[1], Hang Su[1,d], Yaqiang Wang[8], Qiaoqiao Wang[1,d], Zhibin Wang[1], Bettina Weber[1], Stefan Wolff[1,9], Paulo Artaxo[2], Ulrich Pöschl[1], Meinrat O. Andreae[1,10] and Christopher Pöhlker[1]

[1] *Multiphase Chemistry & Biogeochemistry Departments, Max Planck Institute for Chemistry, 55020 Mainz, Germany.*

[2] *Institute of Physics, University of São Paulo, São Paulo 05508-900, Brazil.*

[3] *Leibniz Institute for Tropospheric Research, Permoserstraße 15, 04318 Leipzig, Germany.*

[4] *Environmental Engineering Department, Federal University of Parana, Curitiba PR, Brazil.*

[5] *Department of Biogeochemical Systems, Max Planck Institute for Biogeochemistry, 07701 Jena, Germany.*

[6] *John A. Paulson School of Engineering and Applied Sciences, Harvard University, Cambridge, MA 02138, USA.*

[7] *Department of Earth and Planetary Sciences, Harvard University, Cambridge, MA 02138, USA.*

[8] *State Key Laboratory of Severe Weather & Key Laboratory of Atmospheric Chemistry of CMA, Chinese Academy of Meteorological Sciences, Beijing, China.*

[9] *Instituto Nacional de Pesquisas da Amazonia, Manaus-AM, CEP 69083-000 Brazil.*

[10] *Scripps Institution of Oceanography, University of San Diego, La Jolla, CA 92037, USA.*

[a] *now at: Laboratory for Meteorological Physics, Université Clermont Auvergne, F-63000 Clermont-Ferrand, France.*

[b] *now at: Federal University of Uberlândia, Uberlândia-MG, 38408-100, Brazil.*

[c] *now at: Institute for Climate and Global Change Research & School of Atmospheric Sciences, Nanjing University, Nanjing, 210093, China.*

[d] *Institute for Environmental and Climate Research, Jinan University, China.*

*Correspondence to*: D. Moran-Zuloaga (daniel.moran@mpic.de) and C. Pöhlker (c.pohlker@mpic.de)



**Abstract**

In the Amazonian atmosphere, the aerosol coarse mode comprises a complex, diverse, and variable mixture of bioaerosols emitted from the rain forest ecosystem, long-range transported Saharan dust, marine aerosols from the Atlantic Ocean, and coarse smoke particles from deforestation fires. For the rain forest, the coarse mode particles are of significance with respect to biogeochemical and hydrological cycling as well as ecology and biogeography. However, detailed knowledge on the physicochemical and biological properties as well as the ecological role of the Amazonian coarse mode is still sparse. This study presents results from multi-year coarse mode measurements at the remote Amazon Tall Tower Observatory (ATTO) site. It combines online aerosol observations, selected remote sensing and modelling results, as well as dedicated coarse mode sampling and analysis. The focal points of this study are a systematic characterization of aerosol coarse mode abundance and properties in the Amazonian atmosphere as well as a detailed analysis of the frequent, pulse-wise intrusion of African long-range transport (LRT) aerosols (comprising Saharan dust and African biomass burning smoke) into the Amazon Basin.

We find that, on a multi-year time scale, the Amazonian coarse mode maintains remarkably constant concentration levels (with 0.4 cm$^{-3}$ and 4.0 µg m$^{-3}$ in the wet vs. 1.2 cm$^{-3}$ and 6.5 µg m$^{-3}$ in the dry season) with rather weak seasonal trends (in terms of abundance and size spectrum), which is in stark contrast to the pronounced biomass burning-driven seasonality of the submicron aerosol population and related parameters. For most of the time, bioaerosol particles from the forest biome account for a major fraction of the coarse mode background population. However, from Dec to Apr there are episodic intrusions of African LRT aerosols, comprising Saharan dust, sea salt particles from the transatlantic passages, and African biomass burning smoke. Remarkably, during the core period of this LRT season (i.e., Feb-Mar), the presence of LRT influence, occurring as a sequence of pulse-like plumes, appears to be rather the norm than an exception. The LRT pulses increase the coarse mode concentrations drastically (up to 100 µg m$^{-3}$) and alter coarse mode composition as well as its size spectrum. Efficient transport of the LRT plumes into the Amazon Basin takes place in response to specific mesoscale circulation patterns in combination with the episodic absence of rain-related aerosol scavenging *en route*. Based on a modelling study, we estimated a dust deposition flux of 5-10 kg ha$^{-1}$ a$^{-1}$ in the region of the ATTO site. Furthermore, a chemical analysis quantified the substantial increase of crustal and sea salt elements under LRT conditions in comparison to the background coarse mode composition. With these results, we estimated the deposition fluxes of various elements that are considered as nutrients for the rain forest ecosystem. These estimates range from few g ha$^{-1}$ a$^{-1}$ up to several hundreds of g ha$^{-1}$ a$^{-1}$ in the ATTO region.

The long-term data presented here provide a statistically solid basis for future studies of the manifold aspects of the dynamic coarse mode aerosol cycling in the Amazon. Thus, it may help to understand its biogeochemical relevance in this ecosystem as well as to evaluate to what extent anthropogenic influences have altered the coarse mode cycling already.





## 1 Introduction

The Amazon rain forest is of particular relevance in Earth system science. It represents a vulnerable ecosystem of global importance, which is increasingly disturbed by the combination of climate change and agricultural as well as infrastructural expansion (Davidson et al., 2012). In fact, the Amazon has been ranked as one

of the potential tipping points in the global climate system (Lenton et al., 2008). Furthermore, it represents a unique location to study the human influence on the atmosphere as it represents one of the few remaining continental places where episodes with a near-pristine atmospheric state can be found (Martin et al., 2010a; Pöschl et al., 2010).

Since the late 1980's, various field campaigns have been conducted in the Amazon region, which focused

of specific aspects of the complex atmospheric cycling for time periods of weeks, months, and in some cases up to years (e.g., Andreae et al., 1988; Talbot et al., 1988; Harriss et al., 1990; Talbot et al., 1990; Artaxo et al., 1993; Andreae et al., 2004; Martin et al., 2010a; Artaxo et al., 2013b; Brito et al., 2014; Andreae et al., 2015a; Martin et al., 2016; Wendisch et al., 2016). In 2010, the Amazon tall tower observatory (ATTO) has been established ~150 km northeast (NE) of the city of Manaus, Brazil, for continuous and

detailed observation of meteorology, trace gases, aerosols, and ecology in order to study long-term trends of the Amazonian hydrological and biogeochemical cycling in relation to the increasing extent of man-made perturbations (Andreae et al., 2015).

Generally speaking, the central Amazonian atmosphere swings back and forth between a mostly clean wet season (typically February to May) and a substantially polluted dry season (typically August to Novem-

ber) with corresponding transition periods in between (Martin et al., 2010b; Andreae et al., 2015). This oscillating atmospheric state is determined by the position of the intertropical convergence zone (ITCZ) and the corresponding trade wind circulation as well as the strong seasonality in biomass burning in Africa and South America (Martin et al., 2010a; 2010b; Fuchs and Cermak, 2015; Abdelkader et al., 2017). Figure 1 serves as a general illustration of the large-scale atmospheric circulation as well as the predominant aerosol

and trace gas emissions patterns in the Atlantic region that govern the overall atmospheric seasonality in the Amazon Basin. In particular, the contrasting atmospheric states of the wet vs. dry season are emphasized.

In Fig. 1a and b, the cloud top temperature maps visualize areas with deep convective clouds (i.e., cold cloud tops), which correspond – particularly over the ocean – with the position of the ITCZ. During the wet season, the ITCZ is located (slightly) south of the ATTO site, which entails a northeasterly (NE) trade wind

advection as visualized by back trajectory (BT) ensembles. Here, the trades typically bring air masses from the Atlantic Ocean, which pass over extended and almost untouched forest areas while traveling into the Amazon Basin. During the dry season, the ITCZ is located north of the ATTO site and the trade winds arrive from southeasterly (SE) directions. The SE trades travel over urban and extended agricultural areas in South America and, thus, are prone to bring substantial amounts of anthropogenic pollution into the central Ama-

zon. A detailed description of the corresponding land cover types in the ATTO site's footprint can be found elsewhere (C. Pöhlker et al., 2017). Note that the ITCZ-related changes between Northern and Southern Hemispheric influences are most pronounced for the central Amazon region, which experiences the ITCZ





overpasses. However, in the Northern and Southern Amazon regions, the hemispheric air masses changes
are less pronounced.

Figure 1c and d show a superposition of precipitation and the BT ensembles, visualizing that the most
intense precipitation rates are observed co-located with the deep convection at the ITCZ belt. The combina-
tion of precipitation maps and BT ensembles further shows qualitatively the probability of rain-related scav-
enging of long-range transported aerosols in the advected air masses along their transport track. For the wet
season, the NE trajectories meet the ITCZ rain belt at about 03° N. This means that for the transatlantic pas-
sage north of 03° N (i.e., in the subtropical latitudes) no major wet deposition of the transported aerosol load
is expected. Once the BTs intersect the rain belt south of 03° N, aerosol scavenging becomes a substantial
aerosol loss mechanism, which acts as a barrier for southwards transport (Abdelkader et al., 2017) and "effi-
ciently scrubs out aerosols" from the advected air masses (Andreae et al., 2012). The dry season scenario is
different: Here the ITCZ rain belt is located mostly north of 02° N. This implies that the SE trajectories
mostly bypass the most intense rain fields. Accordingly, the probability of aerosol scavenging during the dry
season is much lower and there is a higher likelihood that (pyrogenic and other) LRT aerosols from Africa
are transported (far) into the central Amazon Basin.

Figures 1e and f combine aerosol optical depth (AOD) maps with the BT data to visualize the major Af-
rican and South American aerosol sources that influence the atmospheric state in the central Amazon. A
main fraction of the anthropogenic aerosol that is observed at the ATTO site is related to biomass burning
emissions from deforestation, savanna, grassland, and agricultural waste fires in Africa and South America.
These, mostly man-made, burning activities on both continents follow a pronounced seasonality: In northern
Africa, a winter fire maximum (i.e., Dec-Mar) is observed due to intense savanna and grassland fires in and
south of the Sahel belt (5°-15° N), while in southern Africa a pronounced summer maximum (i.e., Jun-Oct)
occurs in the Miombo woodlands (5°-20° S) (Barbosa et al., 1999; Edwards et al., 2006). Consequently, both
sources result in extended pollution plumes mixed with dust that travel westwards over the Atlantic Ocean,
as visible in Fig. 1e and f. In South America, a strong biomass burning activity due to agriculture and defor-
estation fires occurs from August to November mostly along the southern margin of the Amazon forest (the
so-called 'arc of deforestation') and in the extended agricultural and Cerrado savanna areas further south
(Edwards et al., 2006; C. Pöhlker et al., 2017).

In addition to the biomass burning smoke, various urban and industrial emissions in Africa (e.g., from
the oil rigs in the Gulf of Guinea) and South America (e.g., from the densely populated south-eastern Brazil-
ian coastline) may also contribute to the anthropogenic aerosol burden (see C. Pöhlker et al., 2017). Figures
1g and h, show the satellite-retrieved total carbon monoxide (CO) column, which highlights the locations of
major biomass burning activities, such as strong fire activity in the Sahel region during the Amazonian wet
season as well as fires in the tropical latitudes in both, Africa and South America, during the Amazonian dry
season.

Beside these man-made emissions, transcontinental advection of Saharan dust across the Atlantic Ocean
represents a major source of LRT aerosol that is relevant for the central Amazon. The boreal summer dust
plumes (August and later) are clearly visible in Fig. 1f as they pass over the Atlantic ocean in a latitude belt



between 12° and 20° N. The winter dust plumes (February-May) in Fig. 1e pass over the ocean in a latitude belt between 06° and 14° N. During the wet season conditions, the back trajectory bundle transects the dust-related AOD plume over the ocean, whereas no significant overlap is observed for the dry season months. This underlines that the Saharan desert represents a relevant aerosol source for the Amazon Basin almost

exclusively during the wet season. It has to be noted that the winter dust plumes are typically mixed with smoke from African savanna fires (Koren et al., 2006; Liu et al., 2008; Ben-Ami et al., 2010). Accordingly, the pronounced AOD plume in Fig. 1e likely represents a mixed signal from dust and smoke as discussed in more detail later in this work.

In the Amazonian atmosphere, a characteristic tri-modal aerosol size distribution prevails, which – in

terms of number concentrations – consists of a rather pronounced and persistent Aitken mode, a mostly dominant accumulation mode, and a comparatively weak coarse mode (Andreae et al., 2015). The multi-modal shape of the size distribution is the result of the interplay of different aerosol sources as well as com-plex aerosol transformation processes (for further information see Sect. S2.1) (Zhou et al., 2002; Martin et al., 2010b). The coarse mode[1] – which is the focus of the present study – originates from different aerosol

sources, such as direct emissions of primary biological aerosol particles (PBAP), marine aerosols, long-range transport (LRT) of Saharan dust plumes, and a coarse mode fraction of biomass burning aerosols (Martin et al., 2010b; Huffman et al., 2012). Coarse mode aerosols play several important roles in the rain forest ecosystem. They can act as ice nuclei (i.e., bioaerosol that are ice-active at high temperatures) and gi-ant cloud condensation nuclei in aerosol-cloud interaction (Prenni et al., 2009; Pöschl et al., 2010). The ad-

vected Saharan dust particles, which mostly occur in the coarse mode, are regarded as crucial fertilizers for the comparably poor Amazonian soils (e.g., by bringing iron and phosphorous) (Swap et al., 1992; Rizzolo et al., 2016). Furthermore, the highly abundant and diverse PBAP population ensures the spread of various organisms in the ecosystem (Womack et al., 2015; Fröhlich-Nowoisky et al., 2016). Further details on these major sources of Amazonian coarse mode particles and their relevance can be found in supplementary sec-

tion S2.2.

The majority of aerosol studies from the Amazon Basin has focused on the life cycle and processing of the submicron size range of the aerosol population and its relevance for the Earth's climate system (Martin et al., 2010b; Artaxo et al., 2013a; Andreae et al., 2015a). Its seasonal variability is largely governed by the extent of biomass burning activities in Africa and South America as well as the transport patterns of the

emitted aerosols. Accordingly, the anthropogenic impact on the fine mode aerosol, including its links to bio-geochemical and hydrological cycling, is particularly pronounced. On the other hand, the atmospheric life cycle of the aerosol coarse mode is not primarily driven by a pollution-related seasonality, but rather defined by the emission and transport of natural aerosols (i.e., desert dust, sea spray, and bioaerosols), which are re-leased and dispersed on different spatiotemporal scales. Whether and to what extent human activities, such

as deforestation fires and land use change, have already altered the coarse mode cycling in the Amazon compared to the preindustrial state, is largely unknown. Furthermore, whether and to what extent coarse

---

[1] Note that different definitions of the coarse mode have been used in the literature (e.g., ≥1 μm, ≥2 μm, and ≥2.5 μm). Here we used the most common definition, which specifies all particles ≥1 μm as coarse mode aerosol.





mode particles (e.g., as IN or giant CCN) have a direct influence on clouds, precipitation, and thus the hydrological cycle in the Amazon also bears unanswered questions. So far, only a few studies have provided first insights into the coarse mode properties and variability in the central Amazon (Huffman et al., 2012; Artaxo et al., 2013b; Womack et al., 2015; Whitehead et al., 2016).

This study aims to partly fill this knowledge gap by presenting a systematic overview of the coarse mode variability from a three-year measurement period at a remote Amazonian site. However, a comprehensive analysis of the complex coarse mode cycling in the Amazon is clearly beyond the scope of this manuscript. Accordingly, we focused on the following two goals: (i) First, we aim to present a general overview of the characteristic seasonal variability in the coarse mode concentration and size distribution to highlight

annually recurring patterns. This general characterization is supposed to serve as baseline for follow-up studies with in-depth analyses of aspects that are only briefly addressed here. (ii) Second, we selected the frequent transport of African dust into the Amazon Basin, which can be observed very clearly in the coarse mode variability. We highlight the atmospheric conditions that explain the episodic transport of African dust into the central Amazon in relation to the characteristic signals that are observed in the coarse mode aerosol

population as well as related parameters. Although the primary focus of this work is the aerosol coarse mode, also selected aspects of the accumulation mode variability have been included in the analysis and discussion, where they help to clarify the overall picture of the relevant atmospheric conditions and processes.

## 2.   Materials and methods

### 2.1   Aerosol observations at the Amazon Tall Tower Observatory (ATTO) site

The aerosol data discussed in this study have been collected at the ATTO site (S 02° 08.602', W 59° 00.033', 130 m above sea level), which is located ~150 km NE of Manaus, Brazil. The ATTO site has been established as a long-term research station for aerosol, trace gas, meteorological, and ecological studies in the central Amazon forest. A comprehensive set of information on the ATTO site can be found in an over-

view paper by Andreae et al. (2015). In 2014 and 2015, the ATTO site was part of the international large-scale field campaign GoAmazon2014/5 that was conducted in and around the city of Manaus from 1 January 2014 until 31 December 2015 (Martin et al., 2016; 2017). During GoAmazon2014/5, the ATTO site served as the clean background site (T0a). Two intensive observation periods (IOPs) took place: IOP1 from 01 Feb to 31 Mar 2014 and IOP2 from 15 Aug to 15 Oct 2014. Furthermore, the measurement period of this study

overlapped with the German-Brazilian ACRIDICON-CHUVA measurement campaign in September 2014 (Machado et al., 2014; Wendisch et al., 2016), where detailed ground-based and aircraft measurements were performed over a large area of the Amazon Basin.

In this study, we mostly focus on measurement data obtained at the ATTO site. However, some selected results (i.e., analysis filter samples, see Sect. 2.4) from the 'neighbor' ZF2 site, which is located 60 km north

of Manaus and about 100 km southwest of the ATTO site, were taken into account (Martin et al., 2010a; Artaxo et al., 2013a). A comparison of results from the ATTO and ZF2 sites is justified by the fact that on average similar atmospheric conditions and air mass advection patterns prevail at both locations as shown in C. Pöhlker et al. (2017). Also the aerosol (optical) properties, analyzed in Saturno et al. (2017a), are compa-



rable. However, some degree of uncertainty is associated with this comparison and has to be taken into account carefully.

### 2.2 Long-term measurements with optical particle sizing and data analysis

This study is based on measurements with an optical particle sizer (OPS, model 3330, TSI Inc. Shoreview, MN, USA), which has been operated continuously at the ATTO site since 30 January 2014. It covers 38 months of OPS data from 30 January 2014 until 30 April 2017. The instrument performs single particle counting and sizing (in the range of 0.3 to 10 μm, divided into 16 size bins) based on aerosol light scattering. All aerosol particle sizes throughout this study represent particle diameters. The OPS size range covers the

aerosol coarse mode as well as the 'tail' of the accumulation mode, which peaks at ~0.15 μm (Huffman et al., 2012; Andreae et al., 2015). The measured aerosol number size distributions ($dN/d\log D_o$) were converted into surface size ($dS/d\log D_o$) and volume size distributions ($dV/d\log D_o$) assuming spherical particles with a shape factor of 1 and a density of 1 g cm$^{-3}$. The sampling intervals and, therefore, the time resolution of the measurements, were set to 5 min. The OPS has been operated inside a measurement container at the base of

a triangular mast, getting ambient air from a 25 mm stainless steel inlet line with a total suspended matter (TSP) inlet head at 60 m a.g.l. and ~30 m above canopy height (for further details see Andreae et al., 2015; M. Pöhlker et al., 2016). The sample air was dried by silica diffusion dryers to a relative humidity (RH) of about 40 %. From 30 Jan 2014 until 02 Feb 2015 the drying was based on frequent manual exchanges of the silica gel cartridges. Since 02 Feb 2015, drying has been based on an automated drying system as described

in Tuch et al. (2009). The OPS data were recorded by the software package 'aerosol instrument manager' (AIM, version 9.0, TSI Inc.). Further analysis and processing were done with the software packages IGOR Pro (version 6.3.7.2, Wavemetrics, Inc.; Portland, OR, USA) and R (version 3.2.0, R Development Core Team, 2010). Periods with biased data (i.e., due to changes of silica gel in the dryers, leaks, local contamination) have been flagged and were not included in the present analyses. The aerosol data reported here were

converted to standard temperature (0 °C) and pressure (1013 hPa) (STP). The time throughout this study is shown as coordinated universal time (UTC). Local time (LT or UTC-4) has only been used for the analysis of diurnal cycles and is marked accordingly.

Optical aerosol sizing techniques have been widely used in aerosol research. However, the following aspects have to be kept in mind: (i) optical aerosol sizing – based on the correlation of particle size and the

intensity of light scattering pulses – is one of three widely used approaches to retrieve equivalent diameters of coarse mode particles. The other experimental strategies rely on geometric or aerodynamic particle sizing (Gwaze et al., 2007; Huffman et al., 2012). (ii) The resulting equivalent optical ($D_o$), geometric ($D_p$), and aerodynamic ($D_a$) diameters and corresponding aerosol size distributions typically deviate substantially due to systematic experimental biases. (iii) In a direct comparison of $D_o$, $D_p$, and $D_a$ retrievals, Reid et al.

(2003a) stated that optical sizing has the largest biases of all three techniques. Specifically, it tends to oversize particles (up to factor 2) and results in a broadening of the coarse mode. These aspects have to be critically considered throughout the subsequent discussion of the OPS-derived results in this study.





### 2.3  Inlet aspiration and transmission efficiency with particle loss corrections

Online aerosol analysis and sampling in the field typically rely on dedicated inlet systems that transport the ambient air to the instruments and/or samplers. However, sampling through an inlet is always non-ideal and, thus, creates sampling artifacts as well as biases the measurement results (von der Weiden et al., 2009). Rep-

5 resentative aerosol analysis requires high aspiration rates (i.e., isoaxial and isokinetic conditions) and high tube transmission efficiencies (i.e., minimized losses) (von der Weiden et al., 2009; Byeon et al., 2015). For the tube transmission, the most relevant particle loss mechanisms (i.e., diffusion, sedimentation, and inertial deposition) are strongly size dependent, and coarse mode aerosol particles, which are the focus of the present study, are particularly prone to the deposition- and inertial-related artifacts. Since the overall sampling

efficiency is critical for the representativeness and interpretation of the coarse mode results reported here, we conducted a detailed analysis of relevant particle losses in the inlet system and applied a corresponding particle loss correction to the OPS data. Details can be found in supplement section S1.1.

### 2.4  Aerosol filter sampling for gravimetric and x-ray fluorescence analyses

Aerosol filters for gravimetric, x-ray fluorescence, and further analysis are collected at the ATTO site on a continuous basis. Prior to Feb 2016, the samples were collected with a two-stage impactor. After Feb 2016, the samples were collected with an automated Partisol$^{TM}$ filter sampler (model 2025i, Thermo Fisher Scientific, Waltham, MA, USA). The samplers were placed on a platform on the walk-up tower at a height ~50 m above ground level (Andreae et al., 2015). The sampling was conducted size-segregated on a 'fine' mode

filter (with $D < 2.5$ µm) and a 'coarse' mode filter (with $2.5 < D < 10$ µm)[2]. The samples were mostly collected on a 5-7 day basis with a low volume sampler (16.7 l min$^{-1}$) with 47 mm diameter and 0.4 µm pore size polycarbonate filters. A similar continuous filter sampling has been conducted at the ZF2 site as well.

The gravimetric analysis was conducted according to a protocol of the US Environmental Protection Agency. Mass concentrations were obtained gravimetrically using an electronic microbalance with a reada-

25 bility of 1 µg (Mettler Toledo, model MX5) in a controlled-atmosphere room under defined conditions (i.e., RH 35% and 20 °C). Filters were equilibrated for 24 h prior weighing. Electrostatic charges were controlled with radioactive $^{210}$Po sources. The experimental error in the elemental mass concentrations is specified as 5 % (see details in Artaxo et al., 2002; 2013b).

Energy dispersive x-ray fluorescence (EDXRF) analysis was used to quantify elements with atomic

numbers ≥11 (Na and heavier) on the filters (Arana et al., 2014). The elements sodium (Na), magnesium (Mg), aluminum (Al), silicon (Si), phosphorus (P), sulfur (S), chlorine (Cl), potassium (K), calcium (Ca), titanium (Ti), manganese (Mn), and iron (Fe) have been analyzed in detail in the context of this study. The detection limits for the individual elements are shown in Table S2, based on the work by Arana et al. (2014). The coloring of the elements in the corresponding figures was done according to the Corey-Pauling-Koltun

(CPK) color schema (http://jmol.sourceforge.net/jscolors/, last access 12 Oct 2017). For the chemical analysis of LRT aerosols as well as aerosols in the absence of strong LRT influence (called non-LRT conditions

---

[2] Note that the definition of fine and coarse mode size ranges in the context of the gravimetric filter analysis (mode separation at $D = 2.5$ µm) is different from the definition that we used throughout this study (mode separation at $D = 1.0$ µm).



here), the EDXRF results from 5 LRT filters and 4 non-LRT filters samples were taken into account. The selection of filter samples for this analysis is based on the online data analysis: LRT samples span LRT episodes according to Table 1 and non-LRT samples are free of detectable LRT influence. Samples spanning both LRT and non-LRT conditions were omitted in this analysis accordingly. The LRT filters were collected on (i) 06 Feb 16:18 (UTC) to 09 Feb 16:52 2015, (ii) 27 Feb 15:10 to 02 Mar 17:16 2015, (iii) 11 Mar 17:23 to 13 Mar 15:54 2015, (iv) 20 Mar 16:41 to 23 Mar 16:34 2015, and (v) 02 Apr 16:41 to 08 Apr 16:48 2015. The non-LRT filters were collected on (i) 23 Feb 17:16 to 27 Feb 14:57 2015, (ii) 27 Feb 15:51 to 02 Apr 16:28 2015, (iii) 15 Apr 17:21 to 20 Apr 16:10 2015, and (iv) 20 Apr 16:22 to 24 Apr 15:43 2015. We report the obtained elemental results for the $D < 2.5$ µm and $2.5 < D < 10$ µm size fractions as well as total concentrations/fractions, corresponding to the sum of both size fractions. The shown elemental mass concentrations for the LRT case represent average values based on the five LRT filters and the non-LRT results represent averages of the four non-LRT filters. Elemental mass fractions were calculated relative to the gravimetrically determined total mass loading of the filters. For the estimation of elemental deposition fluxes, the difference in mass concentration $M_{LRT}(X) - M_{non\text{-}LRT}(X)$ for a certain element X as well as the difference in total mass concentration $M_{LRT,total} - M_{non\text{-}LRT,total}$ were taken into account.

### 2.5 Comparison of OPS-retrieved aerosol masses and gravimetric analysis

The calculated aerosol mass concentrations, which are based on the OPS measurements and the PLC (see Sect. 2.3), were validated by means of a comparison with a gravimetric filter analysis as outlined in Sect. 2.4. For the comparison, the following two periods with gravimetric results were available: (i) a period from 06 Sep to 29 Nov 2014, comprising 10 filters and (ii) a period from 30 Sep to 08 Nov 2015, comprising 14 filters. The results of the comparison are summarized in Fig. S2, which shows the coarse mode aerosol mass time series (Fig. S2a and b) as well as a scatter plot with the combined results (Fig. S2c). Figure S2 confirms that the aerosol masses from the direct gravimetric approach and the indirect OPS-based retrieval agree comparatively well. We regard the gravimetric analysis as reference measurement, which confirms that the PLC of the OPS data and the chosen density (i.e., $\rho_{1.0}$) are appropriate to yield reliable OPS-based aerosol mass concentrations at the ATTO site in the absence of major dust influence.

### 2.6 Supplementary aerosol measurements and instrument comparison

A broad set of aerosol and trace gas instrumentation is being operated at the ATTO site on a continuous basis (Andreae et al., 2015). In addition to the OPS data analysis, which is the focus of the present study, we used supplementary data from the following online instruments: a condensation particle counter (CPC, model 5412, Grimm Aerosol Technik, Ainring, Germany), a scanning mobility particle sizer (SMPS, model 3082, TSI Inc., Shoreview, MN, USA), an ultra-high sensitivity aerosol spectrometer (UHSAS, DMT, Longmont, CO, USA), a wide-band integrated bioaerosol sensor (WIBS, model 4A, DMT), a three-wavelength integrating nephelometer (Ecotech Aurora 3000, λ = 450, 525, and 625 nm), a multi-angle aerosol absorption photometer (MAAP, model 5012, Thermo Electron Group, λ = 670 nm), carbon dioxide ($CO_2$), methane ($CH_4$), and carbon monoxide (CO) monitors based on cavity ring-down spectroscopy





(G1301, G1302 analyzers, Picarro Inc, USA) and a ceilometer (CHM15kx, Jenoptik, Germany). The equivalent black carbon ($BC_e$) mass concentrations, $M_{BCe}$, were obtained from the MAAP aerosol absorptivity measurements based on ATTO-specific mass absorption coefficients (MAC) as outlined in Saturno et al. (2017a). The ceilometer has been operated at the ATTO site since Dec 2014. It has been installed with an

inclination angle of 15 degrees to minimize direct sun light. Frequent comparisons of the sizing instruments have been conducted at the ATTO site to ensure comparability of the individual data sets. The supplement Fig. S3 presents the results of such a comparison for the instruments OPS (0.3–10 µm), SMPS (0.01-0.43 µm), UHSAS (0.06-1 µm), and WIBS (1.5-10 µm). The good agreement of the size distributions shows that the results of all instruments are consistent. The ceilometer data was processed using a signal noise ratio

correction by Heese et al., (2010), specifically dedicated to detect aerosol layers in the free troposphere up to ~ 4 km height during daytime hours.

### 2.7   Backward trajectory analysis

Our investigation of the atmospheric transport in this study is based on a systematic backward trajectory

(BT) analysis, which has been adopted from a recent study where details can be found (C. Pöhlker et al., 2017). Briefly, Fig. S4 shows 15 clusters of 3-day BT ensembles, which describe the spatiotemporal variability of air mass movement towards the ATTO site over the NE Amazon Basin. The choice of 15 clusters is explained in C. Pöhlker et al. (2017). The choice of 3-day BTs is based on the following rationales: (i) The mesoscale circulation that transports the African LRT aerosol plumes from the Atlantic coast into the Ama-

zon Basin are of primary interest for this study. In this regard, the 3-day BTs sufficiently cover the relevant NE fetch across the Basin. (ii) Moreover, the region of interest $ROI_{off}$ (see details in Sect. 2.8) is relevant for rain-related aerosol scavenging (i.e., of LRT aerosol, see Fig. 1c and d), which is of particular relevance for the analyzed phenomena. Figure S4 illustrates that the air masses at the ATTO site arrive almost exclusively in a rather narrow easterly wind sector (between 45° and 120°). Within this sector, four BT subgroups can be

identified: (i) a northeasterly (NE) track including the clusters NE1, NE2, and NE3; (ii) an east-northeasterly (ENE) track including the clusters ENE1, ENE2, ENE3, and ENE4; (iii) an easterly (E) track including the clusters E1, E2, E3, and E4; and (iv) a group of 'inland' trajectories in east-southeasterly (ESE) directions including clusters ESE1, ESE2, and ESE3 as well as one cluster towards the south-west (i.e., SW1).

### 2.8   Satellite data analysis

The satellite data products used in this study were obtained from the Giovanni web-based application by the Goddard Earth Sciences Data and Innovation web interface (http://giovanni.gsfc.nasa.gov/, last accessed 04 Jul 2017) (Acker and Leptoukh, 2007). The following satellite products were used: (i) aerosol optical depth (AOD) at a wavelength of 550 nm from the moderate resolution imaging spectroradiometers (MODIS) on

the satellites Terra and Aqua (combined dark target deep blue AOD products MOD08_D3_V6 and MYD08_D3_V6), (ii) cloud top temperature data from the atmospheric infrared sounder (AIRS) instruments on board of Terra and Aqua (AIRX3STD_v006 product), (iii) precipitation data from the tropical rainfall measuring mission (TRMM) mission (TRMM_3B42_Daily_v7 product), and (iv) CO total column data



from the AIRS instruments on board of Terra and Aqua (AIRS3STD_v006 product). These data sets were used as time-average maps as well as time series for specified regions of interest (ROI). The time series of MODIS and TRMM satellite data products were obtained as area averages within two ROIs: (i) a ROI in front of the NE coast of the Basin ('offshore'), called $ROI_{off}$ (49° W to 37° W; 0° N to 12° N) and (ii) a ROI

covering the ATTO region, called $ROI_{ATTO}$ (59° W to 54° W; 3.5° S to 2.4° N). Both ROIs are displayed in Fig. S4. The data files were exported as netcdf (Network Common Data Form, version 3.x) or ascii files. Further data processing was conducted in IGOR Pro. Cloud Lidar Aerosol Infrared Pathfinder Satellite (CALIPSO) data products – i.e., lidar profiles from the cloud-aerosol lidar with orthogonal polarization (CALIOP) – were obtained from the website: https://www-calipso.larc.nasa.gov/ (last access 15 Jun 2017).

Daily wind data and precipitation were obtained from National Center for Environmental Prediction NCEP (http://www.esrl.noaa.gov/psd/data, last access 20 Sep 2016) and using the software MeteoInfo (see details in Y. Wang, 2014).

### 2.9  GEOS-Chem modelling

The modelling results used here are based on GEOS-Chem version 9-02 (http://www.geos-chem.org/, last access 17 May 2017). GEOS-Chem is a chemical transport model with a global 3-D model of atmospheric composition driven by assimilated meteorological data GEOS-5 FP from the NASA Global Modeling and Assimilation Office (GMAO). Aerosol types simulated in GEOS-Chem include carbonaceous aerosols (fine mode), sulfate–nitrate–ammonium aerosols (fine mode), fine and coarse mode sea salt, and mineral dust in

four size classes. For details, we refer the reader to a previous study by Q. Wang et al. (2016), which showed that GEOS-Chem successfully captured the observed variation in aerosol properties in the Amazon Basin during Jan-Apr of 2014.

### 2.10  Amazonian seasonality, nomenclature, and definition of LRT episodes

This study utilizes the definition of the Amazonian seasons according to M. Pöhlker et al. (2016) as follows: (i) the "wet season" spans Feb to May and shows the cleanest atmospheric state, (ii) the "transition period from wet to dry season" spans Jun and Jul, (iii) the "dry season" months Aug to Nov show the highest pollution levels, and (iv) the "transition period from dry to wet season" spans Dec and Jan. For the present analysis, all results referring to "transition period" include both, the transition period from wet to dry season and

the transition period from dry to wet season. Results referring to the "LRT episodes" of a certain year or for the entire measurements period represent averaged results from the individual LRT episodes as listed in Table 1. Note that LRT episodes in this study refer exclusively to LRT during the Amazonian wet season, when African dust and smoke has been transported to the Amazon. During the dry season, LRT aerosols (mostly smoke without dust) from Southern Africa arrive in the Amazon Basin via similar transport mecha-

nisms. These dry season LRT events, which typically do not transport substantial dust loads and, thus, are less relevant for the coarse mode, have not been addressed in the context of this study. Details regarding dry season LRT aerosols can be found in Saturno et al., (2017a; 2017b). Results referring to the "wet season



without LRT" represent average data from the wet season time frame with the corresponding LRT episodes in Table 1 being excluded.

As LRT episodes, we defined periods with continuous dust influence at the ATTO site, when the following three criteria were fulfilled: (i) increased AOD values were detected by the spaceborne MODIS instru-

ments (see examples in Fig. 11 and S10) for clear detection of African dust outbreaks, transatlantic passage, and arrival at the South American coast, (ii) the maximum of daily averaged coarse mode mass concentrations, $M_{1\text{-}10}$ exceeded 9 µg m$^{-3}$ (average wet season concentration + 2 standard deviations), as sensitive marker for the abundance of supermicron particles, and (iii) the $M_{1\text{-}10}$ time series showed a peak-like increase that lasted longer than one day. Episodes with chemical information on aerosol composition (i.e.,

based on weekly filter samples, see Sect. 2.4) confirm that the afore mentioned criteria are sufficient to identify major and medium dust events. The observed LRT episodes typically span periods from 2 to 20 days. However, it has to be noted that the definition of the (precise) beginning and end of each LRT episodes may be considered arbitrary to some extent because of the comparatively high variability of the $M_{1\text{-}10}$ background.

## 3   Results and Discussion

### 3.1   Time series of the aerosol coarse mode and related parameters

The overview Figure 2 shows coarse mode aerosol data from almost two and a half years (Feb 2014 until Jun 2016)[3]. It combines this data with selected meteorological and aerosol time series and, thus, puts the

coarse mode trends in the context of the overall atmospheric seasonality at the ATTO site. Figure 2a displays the daily frequency of occurrence of the individual BT clusters, $f_{\text{BT,cluster}}$ (see Fig. S4 for BT cluster overview and color coding). This shows the clear prevalence of northeasterly (NE) and east-northeasterly (ENE) trajectories during the wet season (i.e., Feb-Apr), followed by a rather sudden shift to easterly (E) and east-southeasterly (ESE) trajectories towards the end of the wet season (i.e., May). During the subsequent

transition period (i.e., Jun-Jul) and most of the dry season (i.e., Aug-Oct), E and ESE trajectories prevail. Eventually, at the end of the dry season (i.e., Nov) and in the subsequent transition period (i.e., Dec-Jan), the dominant wind direction gradually swings back from SE to NE directions.

In general, the seasonality in the atmospheric composition in the central Amazon is largely driven by the seasonal cycle of deforestation fire activity in combination with the changing air mass transport patterns.

This biomass burning seasonality is detectable in most aerosol parameters (Andreae et al., 2015; M. Pöhlker et al., 2016). As examples in Fig. 2b and c, the accumulation mode abundance, $M_{\text{BCe}}$, and the aerosol scattering coefficient, $\sigma_{\text{sp}}$, time series showed pronounced maxima, with highest concentrations during the core months of the dry season (i.e., Aug-Oct). In this overall picture, the coarse mode seasonality shows different trends. This can be seen in the plot of the surface size distribution[4] (i.e., from 1 to 10 µm, Fig. 2b) as well as

---

[3] Figure 2 covers the LRT episodes 2014, 2015, and 2016. Note that we also analyzed the LRT episodes in 2017 for this work, however, decided to limit the time frame of Fig. 2 to only three LRT episodes for the sake of clarity (for details see Sect. 2.2).

[4] We chose the surface size distribution as an adequate representation of the aerosol population, since the corresponding image plot shows the aerosol concentrations in the sub- and supermicron ranges at comparable intensities.





in Fig. 2e as time series of total aerosol mass concentration covering the size ranges 0.3-10 μm ($M_{0.3-10}$), 1-10 μm ($M_{1-10}$), and 0.3-1 μm ($M_{0.3-1}$). Particularly, $M_{1-10}$ represents a sensitive indicator for the overall abundance of coarse particles (i.e., >1 μm). The coarse mode aerosol mass concentration $M_{1-10}$ was generally low and ranged below 10 μg m$^{-3}$ for most of the measurement period. The trend was only interrupted by annually

reoccurring and well defined peaks, which mostly occurred in the period Dec to Apr as detectable in Fig. 2b and e.

The observed peaks in $M_{1-10}$, which Fig. 2 highlights by grey vertical shadings, represent the frequent intrusion of African LRT dust and combustion aerosol plumes (called here LRT episodes). The identification of major LRT episodes in the ATTO data was rather straightforward as they are associated with pronounced

increases in the aerosol parameters (i.e., $M_{1-10}$ and $M_{BCe}$). However, the detection of minor and/or 'diluted' LRT episodes turned out to be difficult in certain cases with respect to the variable coarse mode background, which is mostly driven by PBAP cycling. The criteria that were used for the definition of LRT episodes are outlined in Sect. 2.10. Table 2 summarizes all detected[5] major and medium LRT episodes from Feb 2014 to Apr 2017. Note that the $M_{1-10}$ peaks clearly coincide with corresponding signals in $M_{BCe}$ and $\sigma_{sp}$, underlining

that the LRT aerosols typically represent mixtures of Saharan dust (mostly in the coarse mode), biomass burning smoke (mostly in the accumulation mode), and sea spray (in the coarse and accumulation modes) (Quinn et al., 1996; O'Dowd et al., 2008; Wang et al., 2016; Aller et al., 2017; Huang and Jaeglé, 2017). Accordingly, the image plot of the aerosol size distribution shows (pronounced) LRT pulses in both, the super- and submicron size ranges. Note in this context that also the scattering Ångström exponent, $å_{sca}$, is a sensi-

tive indicator for the presence of coarse mode particles. Qualitatively, this effect can be seen in Fig. 2c by means of a decreasing difference of the three $\sigma_{sp}$ time series during LRT episodes. Quantitatively, this effect is shown by Saturno et al. (2017a) in a multi-year $å_{sca}$ time series from the ATTO site with clear $å_{sca}$ decreases during the presence of African dust aerosols. Mostly, the LRT episodes occurred in a rather defined time window from Dec to Apr, which was described previously as 'Amazonian dust season' (Andreae, 1983;

Swap et al., 1992; Formenti et al., 2001; Rizzolo et al., 2016).

The pulse-wise arrival of African LRT aerosol plumes at the ATTO site, as shown in Fig. 2, appears to be controlled by the following three factors: (**1**) arrival and availability of LRT aerosol plumes at the South American coast, (**2**) atmospheric circulation in the NE Basin and its efficiency to transport dust from the coast towards the ATTO site, and (**3**) the extent of wet deposition of the aerosol load *en route*. Note that (**2**)

and (**3**) are related to some extent since aerosol deposition by precipitation plays a central role in both of them. However, we decided to outline both aspects separately for the sake of clarity.

Relating to (**1**): Saharan dust outbreaks are frequent and circulation patterns and aerosol deposition over the Atlantic Ocean define if and to what extent the LRT plumes arrive at the NE margins of the Basin (Gläser et al., 2015). To analyze this arrival of dust plumes on a temporal scale with area-averaged satellite

data, we defined an (offshore) region of interest (ROI$_{off}$) NE of the Amazon River delta over the Atlantic Ocean as displayed in Fig. S4. This ROI$_{off}$ intersects with the main tracks of the wet season trajectory clus-

---

[5] Due to several data gaps, particularly between Dec 2014 and May 2015, the coverage of the LRT episodes in 2015 is incomplete. As LRT episodes likely occurred during the data gaps, these events were not covered by the present analysis and, thus, are not listed in Table 1.



ters (i.e., NE and ENE) and further covers the arriving LRT plumes. The resulting satellite-derived and ROI$_{off}$-averaged AOD time series (AOD$_{ROI, off}$) is displayed in Fig. 2d. Note that the transport time of the air masses over the last ~1500 km from the ROI$_{off}$ to the ATTO site takes on average 3 days (C. Pöhlker et al., 2017). In order to (qualitatively) compare the $M_{1-10}$ signals at the ATTO site with the AOD$_{ROI, off}$ variability

of arriving dust plumes at the coast, we 'synchronized' both time series via lagging the AOD$_{ROI, off}$ data by 3 days. In other words, the direct comparison of the shifted AOD$_{ROI, off}$ time series shows the amount of dust that potentially arrived at the ATTO site (Fig. 2d) vs. the $M_{1-10}$ time series that indicates the amount of dust that actually arrived (Fig. 2e). This comparison (Fig. 2d vs. 2e) indicates that only a rather small fraction of the dust load at the coast actually reached the ATTO site (i.e., the majority of AOD$_{ROI,off}$ peaks did not result

in $M_{1-10}$ signals). The following examples in Fig. 2 illustrate these trends: The pronounced dust pulses at the ATTO site around 18 February, 07 March, and 10 April 2014 were clearly related to corresponding AOD$_{ROI,off}$ increases. However, note that the intensities of $M_{1-10}$ and AOD$_{ROI,off}$ in this relationship did not necessarily correlate. For instance, the $M_{1-10}$ pulse on 06 April 2015, which represents one of the strongest signals observed during the entire measurement period, was associated with a comparatively weak signal in

AOD$_{ROI,off}$. Furthermore, all April-May periods in Fig. 2 showed pronounced and continuous AOD$_{ROI,off}$ signals, however only sparse dust transport towards the ATTO region as shown by the few occurring $M_{1-10}$ peaks. To further underline these aspects, the supplement Fig. S5 focuses specifically on a direct $M_{1-10}$ vs. AOD$_{ROI,off}$ comparison. In summary, the AOD$_{ROI,off}$ vs. $M_{1-10}$ time series underline that the LRT plume transport from the coast over the NE region of the Basin towards the ATTO site can be regarded as a 'bot-

tleneck', as most of the LRT aerosol load appears to be scavenged on its way here, which is subject of detailed discussion in the following paragraphs.

Relating to (**2**): A second key factor appears to be the efficiency of dust transport coming from the coast to the ATTO site. In this context, efficiency means a direct and fast transport of air masses from regions with enhanced arriving dust loads in front of the coast towards the ATTO site. This relationship can be illustrated

by means of the frequency of occurrence of the back trajectory clusters $f_{BT}$ (Fig. 2a). Specifically, the occurrence of $M_{1-10}$ peaks appears to be predominantly associated with high $f_{BT}$ of those BT clusters that reach furthest to the NE: namely NE2 and NE3. The majority of dust pulses in Fig. 2 followed this trend. To mention a few characteristic examples: All $M_{1-10}$ pulses in Feb to Apr 2014 precisely coincide with the purple-bluish peaks of high NE2 and NE3 $f_{BT}$ in Fig. 2a. The efficiency aspect of the transport can be further ex-

plained by the spatiotemporal trends in Fig. 3, which displays the longitude-averaged Hovmöller plots for satellite-derived AOD levels and precipitation rate $P_{TRMM}$. The geographic dimensions of the Hovmöller plots are based on the ROI$_{off}$: The longitudinal range (49° W to 37° W), which is averaged in the Hovmöller representation, is identical in the ROI$_{off}$. Moreover, the latitudinal range of the Hovmöller plot (10° S to 30° N) includes the ROI$_{off}$ (0° N to 12° N), however, it reaches further north and south. Figure 3a illustrates

the annual latitudinal shifts in the LRT plume position (Huang et al., 2010). Thus, at the beginning of the year during February and March, the southernmost position was observed with an AOD maximum at ~06° N. The northernmost position is observed in July and August with an AOD maximum at ~17° N. In parallel, a pronounced spatiotemporal shift can be found in $P_{TRMM}$: particularly in March the southernmost



position of the $P_{TRMM}$ maximum is observed at ~00° N, whereas the northernmost position occurs in September at ~07° N, which corresponds with the ITCZ shifts as previously showed in Fig. 1c and d.

In the light of Fig. 3, the high dust transport efficiency of the BT clusters NE2 and NE3 can be explained as follows: (i) Both clusters reached comparatively far to the north and, thus, had the highest overlap with the densest LRT plume regions (i.e., both clusters transect the AOD maximum at ~07° N; compare Fig. 1e and 3a). Thus, they tended to transport (on average) the highest LRT aerosol loads into the Basin. (ii) Both clusters are comparatively 'long', which reflects high air mass velocities and, thus, minimizes the probability of aerosol scavenging. (iii) Moreover, both clusters bypassed the densest rain fields in the north and, thus, tended to avoid intense scavenging (compare Fig. 1c and 3b). Overall, this study clearly show that the clusters NE2 and NE3 are particularly efficient dust transporters, as they have maximum overlap with the high-AOD region and, at the same time, a rather small overlap with high-$P_{BT}$ regions. In contrast, the ENE clusters have less overlap with the dense AOD fields and higher probability to receive larger amounts of precipitation. Accordingly, periods with high frequency of occurrence of the ENE clusters were typically not associated with increased $M_{1-10}$ levels. The influence of the air mass transport track and rain fields is discussed in further detail by means of a case study in Sect. 3.5

Relating to (**3**): Wet deposition is the dominant aerosol loss mechanism in tropical latitudes because of their intense precipitation (Huang et al., 2009; Martin et al., 2010a; Abdelkader et al., 2017). According to Fig. 1c, comparatively small scavenging rates are expected for most of the dust's transatlantic passage (i.e., north of 03° N), whereas precipitation rates (on average) increase instantaneously when the air masses meet the ITCZ rain belt. As a measure for the extent of scavenging rates that the air masses experience in the NE Basin, we calculated the cumulative precipitation along the 3-day BT tracks, $P_{BT}$ , (shown as daily averages).[6] In other words, the intense precipitation in the NE Basin defined if and to what extent the LRT plumes reached the ATTO region. The $P_{BT}$ time series shown in Fig. 2d and its comparison with $M_{1-10}$ in Fig. 2e clearly underlines this relationship: Virtually all $M_{1-10}$ pulses correspond with relative minima in the $P_{BT}$ time series. This shows, expectedly, that the dust burden that arrived at the ATTO site was inversely related to the cumulative amount of rain that the corresponding air masses received. In other words, only dust plumes that survived the intense rain-related scavenging had a chance to arrive in the ATTO region. Good examples for this relationship (among many others) are the dust pulses around 18 February 2014 and 06 April 2015.

Based on the time series in Fig. 2, we calculated mean seasonal cycles of selected aerosol, trace gas, and meteorological parameters, which were combined into Fig. 4. Meteorologically, Fig. 4a shows the seasonal trends in BT patterns. The annual oscillation between wet vs. dry season BTs stands out clearly. More specifically, the BT patterns illustrate the timing of changes in the dominant wind direction, such as its swing from NH to SH and back upon latitudinal passage of the ITCZ. Further details related to Fig. 4a can be found elsewhere (C. Pöhlker et al., 2017). Figure 4b shows the seasonality of two different precipitation parameters: $P_{BT}$ and the TRMM-derived precipitation rate $P_{TRMM}$ in the ROI$_{ATTO}$ (see Fig. S4). The $P_{BT}$ data

---

[6] A direct comparison of $P_{BT}$ for 3-day *versus* 9-day back trajectories confirms most of the cumulative precipitation results from the last 3 days of the air mass journey, which underlines that the region of the ITCZ belt is most important for aerosol wet deposition (see Fig. 2c).



represents a measure for the aerosol scavenging in the transported air masses and, thus, provides important information about (LRT) aerosol removal *en route*. In contrast, the $P_{TRMM}$ data represents a regional characterization of the precipitation in an area around ATTO. The seasonality in $P_{TRMM}$ shows a rather broad maximum spanning most of the wet season (i.e., Feb-May), whereas the seasonal trends in $P_{BT}$ show a compara-

tively narrow and well pronounced maximum in Apr and May. Figure 4c shows the seasonality of the pollution tracers $M_{BCe}$ and $c_{CO}$, reflecting the pronounced biomass burning seasonality in South America as well as Africa as a major source of LRT pollution. The cleanest episodes in terms of pollution aerosols (see $M_{BCe}$) occurred between Apr and May, which could be explained by the maximum in aerosol scavenging (see highest $P_{BT}$ values in Apr and May) (compare also near-pristine periods in Pöhlker et al., 2017). In parallel, a

minimum in biomass burning occurrence is typically found during this period (i.e., the Amazonian burning season has not started yet and the frequency of savanna fires in Africa declines after February) (Andreae et al., 2015). The seasonality of the MODIS-derived parameter, $AOD_{ROI,off}$, in Fig. 4d, representing arriving LRT plumes in the NE coast of the Basin, shows a comparatively broad peak with a maximum around March.

Figure 4e shows the seasonality of the coarse mode aerosol abundance (represented by $M_{1-10}$) at ATTO, which is displayed in two modifications: (i) as a data set that includes the *entire* time series and, thus, reflects the $M_{1-10}$ seasonality with all coarse mode-relevant aerosol sources ('$M_{1-10}$ with LRT') and (ii) as a data set *excluding* all LRT periods as defined in Table 2 and, thus, reflecting the $M_{1-10}$ seasonality without (most of) the African LRT influence ('$M_{1-10}$ without LRT'). The '$M_{1-10}$ with LRT' data expectedly shows its

highest levels in the dust season (Dec-Apr) with weekly average concentrations frequently exceeding 10 µg m$^{-3}$. In contrast, the '$M_{1-10}$ without LRT' data shows a rather modest seasonal cycle with average concentrations around 6-7 µg m$^{-3}$ during the dry season (i.e., broad maximum spanning Aug-Oct) and around 4 µg m$^{-3}$ during the wet season (i.e., with a minimum in Apr). In the absence of African LRT influence, the coarse mode mostly comprises bioaerosols from local/regional sources (see also Pöschl et al., 2010;

Huffman et al., 2012). Accordingly, Fig. 4e indicates that the Amazonian atmosphere contains a rather constant coarse mode background, in which bioaerosols likely account for a dominant fraction. As possible explanation for this (modest) seasonality of the '$M_{1-10}$ without LRT' background, we suggest that this could be determined by (i) differences in aerosols scavenging frequency (see seasonality of $P_{BT}$), (ii) a certain fraction of coarse mode particles originating from biomass burning plumes, and/or (iii) different bioaerosol emis-

sions patterns and strength in the wet vs. dry season.

### 3.2   Seasonally averaged aerosol concentrations and size distributions

Figure 5 and Table 2 provide a statistical summary of selected aerosol concentrations, resolved by season. Figure 5a and b show the seasonal levels of $N_{total}$ and $M_{BCe}$, reflecting the characteristic biomass burning driven trends: the wet season shows clean background concentrations (i.e., $N_{total} = 336 \pm 209$ cm$^{-3}$ and $M_{BCe} = 0.02 \pm 0.03$ µg m$^{-3}$, mean $\pm$ 1 std. dev.), whereas highest concentration levels occur in the dry season



(i.e., $N_{total} = 1508 \pm 785$ cm$^{-3}$ and $M_{BCe} = 0.35 \pm 0.20$ µg m$^{-3}$). The transitions periods represent an intermediate state in between these extremes (i.e., $N_{total} = 790 \pm 547$ cm$^{-3}$ and $M_{BCe} = 0.17 \pm 0.20$ µg m$^{-3}$). During the LRT season, we observed a clear $M_{BCe}$ enhancement in comparison to the wet season background (i.e., $M_{BCe} = 0.17 \pm 0.15$ µg m$^{-3}$ vs. $M_{BCe} = 0.02 \pm 0.03$ µg m$^{-3}$), due to the smoke fraction in the advected African

LRT plumes. Remarkably, the $N_{total}$ levels for wet vs. LRT episodes show no statistically significant difference. This suggests that $M_{BCe}$ is a sensitive indicator to discriminate near-pristine episodes and periods that are influenced by long-range transported African pollution, whereas the use of $N_{total}$ as a pollution tracer may be misleading (see also M. Pöhlker et al., 2017).

      The statistics of the multi-year OPS data is presented as number and mass concentrations for two size

ranges: The range from 0.3 to 1µm in Fig. 5c and d covers the large-particle tail of the accumulation mode and, thus, to certain extent follows the biomass burning seasonality, similar to $N_{total}$ and $M_{BCe}$. Note that the parameters, which are sensitive to biomass burning pollution (i.e., $N_{total}$, $M_{BCe}$, $N_{0.3-1}$, $M_{0.3-1}$) generally show a wide range of statistical scattering (see large std. dev.), which can be explain by the fact that dry season aerosol time series are characterized by a sequence of high-concentration peaks due to the plume-wise advec-

tion of biomass burning emissions (see M. Pöhlker et al., 2017; Saturno et al., 2017a). In contrast, the range from 1 to 10 µm in Fig. 5e and f represents the coarse mode seasonality, with differs from the biomass burning trends. The $N_{1-10}$ and $M_{1-10}$ levels show a modest increases from the wet season ($N_{1-10} = 0.42 \pm 0.34$ cm$^{-3}$, $M_{1-10} = 4.04 \pm 2.72$ µg m$^{-3}$) over the transition periods ($N_{1-10} = 0.81 \pm 0.75$ cm$^{-3}$, $M_{1-10} = 5.24 \pm 3.46$ µg m$^{-3}$) to the dry season ($N_{1-10} = 1.15 \pm 0.81$ cm$^{-3}$, $M_{1-10} = 6.47 \pm 2.69$ µg m$^{-3}$). The highest $N_{1-10}$ and $M_{1-10}$ levels

clearly occurred during African LRT influence ($N_{1-10} = 2.03 \pm 1.87$ cm$^{-3}$, $M_{1-10} = 11.28 \pm 9.05$ µg m$^{-3}$).

      To discuss our observed $N_{1-10}$ and $M_{1-10}$ concentrations in the context of previous measurements, we summarized the results from related studies in Table 3. A certain number of previous studies from the central and Southern Amazonian region reported coarse mode concentrations, which agree well with the $N_{1-10}$ and $M_{1-10}$ levels in this work (Artaxo et al., 2002; Graham et al., 2003; Guyon et al., 2003; Huffman et al.,

2012; Artaxo et al., 2013b; Whitehead et al., 2016). For comparison, we added a few studies from other locations and ecosystems (i.e., boreal and semi-arid forest sites) to Table 3, which suggest that the coarse mode (background) concentrations in different forested ecosystems are remarkably similar and range around 0.5 cm$^{-3}$ (Schumacher et al., 2013).

      Figure 6 illustrates the seasonal differences in the aerosol size distributions obtained from the multi-year

OPS measurements. Similar to the number and mass concentration trends in Fig. 5e and f, we observed the strongest coarse mode during African LRT episodes, followed by the dry season, the transition periods, and lastly the wet season. In addition to these concentration trends, the coarse mode also reveals seasonal characteristic shapes (see surface and volume size distributions in Fig. 6b and c). Under wet season conditions, the coarse mode maximum is shifted towards large diameters (~2.8 µm in the surface and ~4.2 µm in the vol-

ume size distributions) and the entire mode can be characterized as a broad monomodal distribution. In contrast, during the dry season, the coarse mode shape is clearly different. It has a multimodal appearance and the strongest, rather narrow peak is located at ~1.7 µm (surface size distribution) and ~2.0 µm (volume size distribution), respectively. Towards larger diameters, a pronounced shoulder indicates the presence of one or



two further modes. During the transition periods, the coarse mode resembles a mixture of the wet and dry season size distributions. The coarse mode shape during Saharan dust influence appears monomodal with its maximum at ~2.0 µm (surface size distribution) and ~2.4 µm (volume size distribution), respectively. A comparison of the LRT aerosol size distribution after the transatlantic transport (see Fig. 6) with Saharan

dust size distributions on or nearby the African continent (e.g., Di Biagio et al., 2017) shows – expectedly – that the dust population has been depleted in large particles (i.e., >2 µm) due to gravitational settling.

For comparison, we added coarse mode size distributions from two previous Amazonian studies to Fig. 6. It has to be kept in mind that optical particle sizing may deviate from geometric and/or aerodynamic sizing approaches (see also Sect. 2.2). Huffman et al. (2012) conducted a multi-week measurement in the

central Amazon using an ultra-violet aerodynamic particle sizer (UV-APS) to probe wet season conditions with African dust influence. The resulting campaign average number and volume size distributions agree well with our observations for particle sizes <3 µm (Fig. 7a and c).[7] For diameters >3 µm, the OPS-based results shows much higher particles abundances than the UV-APS-based distribution. This deviation can probably be explained by a combination of different reasons. An important aspect could be that optical parti-

cle sizing tends to oversize aerosol particles relative to aerodynamic particle sizing (i.e., UV-APS) (Reid et al., 2003b). Moreover, optical sizing is often associated with a broadening of the distributions. Both of these tendencies are consistent with our observations in Fig. 6a and have also been reported previously for the Amazonian coarse mode (Martin et al., 2010a). However, a systematic and long-term comparison of optical and aerodynamic particles properties of Amazonian aerosols is subject of ongoing work. The second study

for comparison was conducted by Whitehead et al. (2016) using a WIBS (optical sizing) to probe several weeks of the transition period from wet to dry season (Fig. 6a). The WIBS data agree well with our observations for the particle size range >2 µm. Below 2 µm, the detection efficiency of the WIBS drops, which probably explains the deviation between WIBS and OPS in this range.

**3.3   Comparison of experimental data and GEOS-Chem model results**

For the wet season and LRT episodes of the years 2014 and 2015, a comparison of the experimental data (i.e., $M_{0.3-10}$) and GEOS-Chem model results along the lines of the study by Q. Wang et al. (2016) was conducted. Measured and modeled results show a rather accurate agreement, particularly for the timing and mass concentrations of the LRT episodes. The corresponding time series and a bivariate regression fit are

shown in Fig. S6 and S7. The good agreement indicates that the relevant factors, controlling the dust transport into the Basin, are accurately covered by the model. Based on this convincing model result validation, we extracted further data products from the model runs, which highlight relevant atmospheric and ecological aspects of the dust deposition in the Amazon Basin.

The effective dust deposition, which introduces essential micro- and macronutrients onto the low-

fertility Amazonian soils, is a relevant aspect from an ecological perspective (e.g., Swap et al., 1992; Okin et

---

[7] During the UV-APS measurement period (i.e., the AMAZE-08 campaign), several African LRT episodes occurred (Martin et al. 2010). Accordingly, the campaign average size distributions, which were reported by Huffman et al. (2012) and are shown in Fig. 6a and c, represent a LRT/wet season mixture. This is consistent with fact that the resulting UV-APS size distributions are located in between the OPS-derived wet and LRT season states.





al., 2004; Bristow et al., 2010; Rizzolo et al., 2016). Figure 7a displays a map of the modelled dust deposition flux into the Basin. For the Amazon region and during the time period Jan to Apr 2014, the model predicts average deposition fluxes from 5 to 500 ng m$^{-2}$ s$^{-1}$, which is in good agreement with previous studies, e.g., by Yu et al. (2015), who indicated a seven-year-average deposition flux in the range from 25 to

160 ng m$^{-2}$ s$^{-1}$ for the entire Amazon area. Note that the dust deposition occurs heterogeneously across the Basin. The highest deposition fluxes (about 100-500 ng m$^{-2}$ s$^{-1}$) can be found in the NE of the Basin (i.e., the Guiana shield and the region around the Amazon River delta), whereas deposition fluxes decrease towards the southwest.

      According to the model, the belt within the deposition gradient, which includes the ATTO region

(shown in yellow), received an average effective dust deposition of about 50-100 ng m$^{-2}$ s$^{-1}$ during the 2014 dust season (i.e., Jan - Apr 2014). Since Jan to Apr represent the LRT core months and include most of the LRT episodes, we assume that most of the dust deposition occurs within this time window. Accordingly, the deposited mass from Jan to Apr is regarded as a good representation of the total annual deposition. Thus, based on the average deposition flux, we obtained a total annual deposited dust mass of about 0.5-1 g m$^{-2}$ or

5-10 kg ha$^{-1}$ for 2014, respectively. This is in good agreement with the total deposited mass of 8-50 kg ha$^{-1}$ a$^{-1}$ reported by Yu et al. (2015) as well as the study by Swap et al. (1992), in which the authors estimate that the total mass of introduced dust "may amount to as much as 190 kg ha$^{-1}$ a$^{-1}$" in the northeastern Basin. We propose that the results obtained here for the year 2014 can be regarded as representative for a typical dust deposition scenario in the Amazon region, since 2014 was generally an 'average' year without

pronounced precipitation and circulation anomalies (M. Pöhlker et al., 2016; C. Pöhlker et al., 2017). Moreover, the four LRT seasons (2014-2017) analyzed in this study show generally similar trends and patterns. Given that the atmospheric input of essential dust-related nutrients plays a crucial role in rain forest ecology, differences in forest structure and diversity (e.g., total biomass) may reflect the spatial difference in dust deposition as shown in Fig. 7a. This link between atmospheric nutrient input and forest ecology has already

been subject of previous studies, however, it requires further research to answer various open questions.

      As an additional aspect, Fig. 7b emphasizes that the deposition of dust into the Basin is predominantly driven by wet processes (i.e., rain out and wash out), which is consistent with previous aspects in this study (e.g., the rain-related pulse-wise modulation of dust plume transport in the Basin). Swap et al. (1992) similarly emphasized that in the Amazonian wet season atmosphere, "precipitation scavenging is the principal

removal mechanisms of Saharan dust". This result further underlines that potential changes in precipitation patterns in the Amazon region also impact the dust deposition into the ecosystem.

### 3.4   Quantification of black carbon fraction in the African LRT plumes

A characteristic feature of the analyzed LRT plumes arriving in the Amazon Basin is their 'smokiness', due to the fact that substantial amounts of pyrogenic aerosols from fires in West Africa are mixed into the Saharan dust aerosols as illustrated in Fig. 1. This is a result of the fact that the biomass-burning season in West Africa coincides with the period of frequent LRT to the Amazon Basin. The presence of pyrogenic





aerosols resulted in high BC concentrations during the LRT episodes in comparison to the wet season background as shown in Fig. 5. Here, we analyze the relative BC fractions and their variability in more detail. For most LRT plumes, a positive correlation between $M_{1-10}$ and $M_{BCe}$ was found, underlining the joint arrival of BC and dust aerosols in the mixed plumes. Selected examples of $M_{1-10}$ vs. $M_{BCe}$ scatter plots can be found

in Fig. S8 with linear bivariate regression fits (with offset), in which the slopes represent the BC fraction relative to the dust aerosol mass. Based on the data shown in Fig. S8, slopes of $0.009 \pm 0.001$ and $0.016 \pm 0.015$ (mean slope $\pm 1$ std. dev.) were found.

Generally, our analysis showed that the relationship between $M_{1-10}$ and $M_{BCe}$ is characterized by a comparatively high variability due to the spatiotemporal heterogeneity of the LRT plumes depending on their

source regions and transport paths from Africa to South America. Along these lines and in order to analyze the temporal trends of the smoke fraction in the arriving plumes, we conducted a systematic linear regression analysis based on the entire dataset analyzed here (i.e., the time frame February to May for the years 2014 to 2017). The corresponding results, which are shown in Fig. 8, clearly indicate that the LRT events in the early wet season (i.e., Feb) comprise substantially higher smoke fractions (reaching up to 0.05) than the LRT

events in the later wet season (i.e., Apr), when the mean slope converges against ~0.005. This observation is consistent with concurrently decreasing fire intensities in Africa as described by Barbosa et al. (1999) and supports lidar-based findings from the wet season 2008 (Baars et al., 2011).

### 3.5  Case studies with detailed analysis of specific LRT events

So far, this manuscript presented the overall and long-term trends in coarse mode variability and the pulsewise Saharan dust advection. In the following sections, we zoom into selected time periods and specific case studies to highlight relevant short-term aspects and observations beyond what has been discussed so far. In particular, we will focus on the LRT episodes of the years 2014 and 2015.

The wet season 2014, which includes seven LRT episodes (2014_1 to 2014_7, see Table 1), is of partic-

ular interest since it overlaps with several intensive observation activities (Fig. 9a): (i) The first IOP of the GoAmazon2014/5 campaign, which targeted clean wet season conditions, took place from 01 Feb to 31 Mar 2014 (Martin et al., 2017). The GoAmazon2014/5 IOPs are subject of intensive analysis of various facets of atmospheric composition (see https://www.atmos-chem-phys.net/special_issue392.html, last access 18 Aug 2017). According to our analysis, the IOP1 spanned five LRT episodes (i.e., 2014_1 to 2014_5, see Table 1),

which interrupted the Amazonian background conditions with 37 of the 59 IOP1 days being classified as LRT impacted. Accordingly, LRT condition are not an exceptional but rather the predominant atmospheric state during this time period. (ii) Q. Wang et al. (2016) conducted an in-depth GEOS-Chem modelling study on light-absorbing aerosols with African LRT plumes being an important source of pyrogenic and dust aerosols. (iii) The CCN studies by M. Pöhlker et al., (2016; 2017) cover three LRT episodes in 2014 and analyze

the CCN-relevant properties of the advected aerosol population (i.e., the LRT episode 2015_7 is discussed in detail there). (iv) Intensive aerosol sampling targeting LRT conditions in Feb and Mar 2014 was conducted and the corresponding results using microspectroscopic techniques are currently prepared for a follow-up manuscript on the morphology, mixing state, and composition of the LRT aerosol population.





For comparison with the conditions in 2014 as shown in Fig. 9, an analogous overview for the LRT episodes in 2015, including five LRT episodes (2015_1 to 2015_5, see Table 1) can be found in the supplement Fig. S9. Generally, Figs. 9 and S9 – which display the online aerosol data at hourly resolution in contrast to daily resolution in Fig. 2 – clearly show the characteristic conditions of efficient dust transport towards AT-

5 TO as discussed in detail in Sect. 3.1: All observed LRT episodes, represented by $M_{1\text{-}10}$, are associated with elevated $AOD_{ROI,off}$ levels, relative minima in $P_{BT}$, and a predominance of the back trajectory clusters NE2 and NE3.

In addition to the multi-day LRT peaks in coarse mode abundance, a pronounced diurnal pattern can be recognized in the coarse-mode-relevant time series (i.e., Fig. 9c and f) as well as in $M_{BCe}$ and $\sigma_{sp}$. In order to

10 characterize these observations in more detail, we extracted diurnal cycles for two contrasting states: wet season conditions without LRT influence (Fig. 10a) and for LRT episodes only (Fig. 10b). In the absence of LRT influence, our results are consistent with previous observations by Huffman et al. (2012) and White-head et al. (2016), showing a maximum in coarse mode abundance during the night (i.e., around 01:00-02:00 LT) and a minimum in coarse mode abundance during afternoon hours (i.e., around 12:00-13:00 LT). It has

15 been suggested by Huffman et al. that these trends are driven by a combination of variable dispersal of bio-logical aerosols, which is "strongly tied to environmental variables, such as solar radiation, temperature, and moisture", as well as the oscillating height of the atmospheric boundary layer that concentrates local emis-sions during night and dilutes them convectively during the day. Along these lines, Fig. 10a clearly illus-trates coherent diurnal patterns of temperature, relative humidity, radiation, and $M_{1\text{-}10}$. Remarkably, Fig. 10a

20 and the results by Huffman et al. showed consistently a secondary maximum at 08:00 LT, which could indi-cate increased sporulation rates due to the onset of solar radiation and continuously high relative humidity levels. The specific responses of PBAP emission mechanisms to micrometeorological conditions are subject of an ongoing analysis. In contrast to this scenario, the diurnal pattern during LRT episodes shows a differ-ent trend as shown in Fig. 10b. Here, the highest coarse mode abundance occurred around ~12:00 LT, collo-

25 cated with the maximum in incoming radiation. This observation suggests that the intrusion of LRT aerosols into the near-surface boundary layer occurred via convective downward mixing from higher altitudes, where the transport of the plumes mostly takes place. After sunset, also the $M_{1\text{-}10}$ levels decrease instantaneously, suggesting an efficient deposition of the LRT aerosol load to surfaces in the canopy space.

As a further step, we zoomed into two particular LRT episodes for a detailed analysis using satellite-

30 based remote sensing data: (i) the event 2014_7 from 08 to 14 April 2014 and (ii) the event 2015_5 from 02 and 10 April 2015. For the event 2014_7, Figures 11 and 12 provide a remote sensing characterization of the corresponding dust plume. The sequence of AOD maps in Fig. 11a-d shows the temporal evolution of the African dust outbreak as it passed over the Atlantic Ocean (04 to 05 Apr), arrived at north-eastern coast of South America (06 to 07 Apr), and traveled (deeply) into the Amazon Basin (08 to 11 Apr). Note that the

35 AOD data indicates a plume arrival on April 08, which agrees very well with our *in situ* observation of the actual plume arrival at ATTO as shown in Fig. 9. Figure 11c shows that the plume impacted a large area in-cluding Southern Venezuela, Guyana, Suriname, French Guiana, and Northern Brazil. Accordingly, the de-tailed presentation of the ATTO measurements for this particular event can be regarded as characteristic for



atmospheric conditions under LRT influence in a comparatively large area of the Northern Amazon Basin. For comparison, the corresponding AOD maps for the 2015_5 event can be found in Fig. S10.

  In addition to the MODIS characterization, Fig. 12 presents lidar data from two CALIPSO overpasses that characterized the dust plume in a rather young state during its transatlantic passage (on 05 Apr 2014) and at a later stage as it reached the ATTO region (on 10 Feb 2014). The CALIPSO data for the overpass on 05 April probed the dust plume in the middle of the Atlantic Ocean and emphasizes its large horizontal extent from 20° N towards the equator (Fig. 12a, b, c). It further illustrates that the aerosol layer is lofted above the marine boundary layer with a vertical extent up to altitudes around 4-5 km and a certain degree of stratification. Note in this context that the transatlantic dust transport has been found to characteristically occur in lamella-like stratifications (Ansmann et al., 2009). Although the CALIOP aerosol subtype categorization (Fig. 12c, f) is showing a thin marine layer close to the surface, shallow moist convection likely facilitated the dust layer also to reach down into the marine boundary layer. Further, the aerosol subtype categorization confirms that the Saharan dust outbreaks during this time of the year are typically mixed with substantial amounts of pyrogenic aerosol, in agreement with Fig. 8 and 9 and the related discussion. The CALIPSO overpass on 10 April shows a lidar profile relatively close to the ATTO site (Fig. 12d, e, f). In the region of the ITCZ belt with its deep convective clouds, the signal is completely attenuated (i.e., 0°-5° S). However, the cloud-free areas show the presence of a compact and relatively well mixed dust layer up to altitudes of 2-3 km, in agreement with Ansmann et al. (2009). For comparison, an analogous CALIPSO characterization for the LRT episode 2015_5 can be found in Fig. S11, shows consistent overall trends. Further note that *in situ* ceilometer measurements at the ATTO site in 2015 confirm that the LRT plumes arrive in the ATTO region as compact and mixed layers below 3 km (see Fig. S12).

  The first prerequisite for effective dust transport to ATTO is the arrival and availability of a dust plume at the South American coast (see Sect. 3.1). For the 2014_7 episode, the remote sensing products shown above indicate the plume arrival at the coast on 06 April. The subsequent effectiveness and the role of wet deposition during its transport over land towards ATTO is illustrated in Fig. 13, presenting a sequence of the average precipitation patterns and wind fields at the 925 hPa level in the respective regions of interest. On 02/03 April, we find an almost closed rain band clearly illustrating the position of the ITCZ, which effectively prevents dust transport further south. In the course of the following days, the rain band became disturbed and moved slightly south, leading to a decrease in precipitation over French Guyana, Suriname and the north-east Amazon Basin. Finally, on 09 and 10 April, precipitation stopped in the NE fetch of ATTO and the NE trade wind circulation established, opening the door for effective advection of dust towards the ATTO site. This interplay of the availability of dust, the timing of transport, and minimal wet deposition underlines the episodic but mesoscale character of dust intrusions into the north-eastern Basin (Swap et al., 1992). For comparison, an analogous and consistent characterization of the 2015_5 episode can be found in Fig. S13.



### 3.6 Chemical characterization of LRT aerosols

In this section, we present an analysis of the chemical composition of the LRT aerosols, which are advected to the ATTO site. This analysis is based on EDXRF data, which is available from the ZF2 site[8] for the LRT season 2015 (see details in Sect. 2.4). Nine multi-day filter samples have been selected that best represent

the conditions with and without LRT influence (see Fig. S9). The results are summarized in Fig. 14 for LRT episodes vs. conditions in the absence of strong LRT plumes (called non-LRT), both for $D < 2.5$ µm and $2.5 < D < 10$ µm. Figure 14 quantifies the elemental contributions to the total collected mass on the filters as well as the mass concentrations of the individual elements. It clearly emerges that the relative fractions and mass concentrations of the dust-related crustal elements Si, Al, Fe, Ti, and Ca as well as the sea salt-related

elements Na, Cl, and Mg are – expectedly – higher for LRT than non-LRT conditions. For instance, a total Si mass concentration of ~500 ng m$^{-3}$ (sum of $D < 2.5$ and $2.5 < D < 10$ µm size fractions) was observed under LRT influence, whereas ~35 ng m$^{-3}$ were found under non-LRT conditions. Similarly, total Na mass concentrations went up to ~130 ng m$^{-3}$ under LRT influence, in contrast to non-detectable amounts for the non-LRT case. Moreover, S and K concentrations are enhanced under LRT influence. Note in this context

that the EDXRF-retrieved sulfate concentrations of ~440 ng m$^{-3}$ (for $D < 2.5$ µm; obtained from conversion of S into $SO_4^{2-}$ mass) under LRT influence agrees well with the aerosol mass spectrometry-based LRT sulfate concentrations of $250 \pm 190$ ng m$^{-3}$ (for $D < 1$ µm) as reported in M. Pöhlker et al. (2017). Interestingly, only minor amounts of Cl were found in the $D < 2.5$ µm fraction, which could indicate a strong processing of small NaCl particles during atmospheric transport and an almost full replacement of the Cl$^-$ anion by

$NO_3^-$ and/or $SO_4^{2-}$ anions, which is in agreement with the comparatively high sulfate concentrations in the same size fraction (Laskin et al., 2012). As a further general trend, the fraction of the afore mentioned inorganic elements tends to be higher in the particle size fraction $D < 2.5$ µm compared to $2.5 < D < 10$ µm. The lighter elements with atomic numbers $< 11$, which account for the remaining part of the total mass, can predominantly be attributed to C, N, and O. For all four cases, the CNO contribution accounts for large frac-

tions, ranging from about 71 % (LRT case for $D < 2.5$ µm, Fig. 14a) to 97 % (non-LRT case for $2.5 < D < 10$ µm, Fig. 14d). Under non-LRT conditions, the elements K, P, and S, which are typically associated with biogenic particles, prevail besides the dominant CNO fraction.[9] Generally, the results in Fig. 14 agree very well with previous studies on the Amazonian aerosol composition, where detailed discussions can be found (Lawson and Winchester, 1979; Talbot et al., 1990; Graham et al., 2003; Guyon et al., 2003). Note

in the context of the high CNO fraction that coarse mode particles in the Amazon are typically coated by SOA (Pöschl et al., 2010).

The elemental data in Fig. 14 can be regarded as an estimate of the typical composition of African LRT aerosols, including Saharan dust, marine aerosols, and smoke, arriving at the ATTO site. This is valuable information as it can be linked to the retrieved dust deposition fluxes in Sect. 3.3. Accordingly, the combina-

tion of both results allows to obtain deposition fluxes of individual elements for different regions of the Ba-

---

[8] The atmospheric conditions at the ZF2 and ATTO sites can be considered as comparable as outlined in Sect. 2.1.
[9] Note that certain traces of dust-related elements (i.e., Si, Al, Fe) were also found in the non-LRT samples, which can be explained by the fact that the non-LRT samples also received minor amounts of dust from the onset and/or decay of LRT pulses before and afterwards (see Fig. S9).



sin (see Fig. 7). This is particularly relevant for those ecologically important elements that are regarded as essential micro- and macronutrients for the rain forest ecosystem, such as Fe, P, S, Ca, Mg, Na, Cl, and others (Swap et al., 1992; Okin et al., 2004; Rizzolo et al., 2016). For the ATTO region, our estimated elemental deposition fluxes are summarized in Table 4. This analysis suggests that the heavier elements with

the largest input fluxes are the crustal elements Si and Al (~410-810 g ha$^{-1}$ a$^{-1}$ and ~200-410 g ha$^{-1}$ a$^{-1}$), followed by sulfur (~140-270 g ha$^{-1}$ a$^{-1}$), and the sea salt elements, Cl and Na (~130-260 g ha$^{-1}$ a$^{-1}$ and ~110-230 g ha$^{-1}$ a$^{-1}$). For the ecologically important element Fe, an input of ~120-240 g ha$^{-1}$ a$^{-1}$ into the rain forest ecosystem was estimated. For P, our analysis results in comparatively small input fluxes of about (~10-20 g ha$^{-1}$ a$^{-1}$), since the P abundance is only slightly enhanced for the LRT (21 ng m$^{-3}$) in comparison to the

non-LRT case (11 ng m$^{-3}$). Swap et al. (1992) also provide deposition fluxes for some selected elements (i.e., Na, K, Cl, P) and ions (i.e., $NH_4^+$, $NO_3^-$, $SO_4^{2-}$, $PO_4^{3-}$). Their deposition fluxes are overall comparable to ours, however, appear to be systematically higher (see Table 4), which is consistent with their significantly higher estimate for the total deposited dust mass (Sect. 3.3). It has to be kept in mind that only part of the total deposited material is soluble and, thus, bioavailable. Accordingly, further dedicated studies on the bio-

available fractions of key nutrient, along the lines of the recent study by Rizzolo et al. (2016), will be needed to explore the link between Saharan dust-related nutrient input and rain forest ecology in more detail.

## 4  Conclusion

In this manuscript, the long-term trends of coarse mode aerosols in the Amazon rain forest are investigated

based on an extensive data set starting in 2014 at the Amazonian Tall Tower Observatory (ATTO) site. The coarse mode aerosols originate from different sources, such as direct emissions of primary biological aerosol particles, marine aerosols, long-range transport (LRT) of Saharan dust plumes, and a coarse mode fraction of biomass burning aerosols (Martin et al., 2010b; Huffman et al., 2012). Therefore, different aspects, such as the seasonal variability of the background coarse mode properties compared to frequent LRT

intrusions are highlighted.

    The complex meso-scale nature of annually re-occurring LRT events in the Amazon Basin is investigated based on a detailed air mass history cluster analysis and further remote sensing and *in-situ* data. Tracking a typical dust layer on its way from the African to the American continent reveals the most important prerequisites for the efficient advection of LRT aerosols: (1) arrival and availability of LRT aerosol plumes at

the South American coast, (2) atmospheric circulation in the NE Basin and its efficiency to transport dust from the coast towards the ATTO site, and (3) the extent of wet deposition of the aerosol load *en route*. Consequently, air mass trajectories with high average air mass velocity, northernmost tracks, and lowest integrated precipitation rates overlap best with off-shore areas of increased dust loading and tend to be the most efficient dust transporters to the ATTO site region.

In contrast to the sub-micron aerosol fraction, the atmospheric life cycle of the aerosol coarse mode is not primarily driven by a pollution-related seasonality. The emission and transport of natural aerosols released and dispersed on different spatiotemporal scales lead to a rather defined and surprisingly stable coarse mode mass concentration of 4-7 µg m$^{-3}$. The seasonal coarse mode number and mass concentration levels





($N_{1-10}$ and $M_{1-10}$) show a modest increase from the wet season ($N_{1-10} = 0.42 \pm 0.34$ cm$^{-3}$, $M_{1-10} = 4.04 \pm 2.72$ µg m$^{-3}$) over the transition periods ($N_{1-10} = 0.81 \pm 0.75$ cm$^{-3}$, $M_{1-10} = 5.24 \pm 3.46$ µg m$^{-3}$) to the dry season ($N_{1-10} = 1.15 \pm 0.81$ cm$^{-3}$, $M_{1-10} = 6.47 \pm 2.69$ µg m$^{-3}$). During the wet season, frequent intrusions of LRT aerosols significantly alter the particle number size distribution and chemical composition. Accordingly, the highest $N_{1-10}$ and $M_{1-10}$ levels clearly occurred during African LRT influence leading to average $N_{1-10} = 2.03 \pm 1.87$ cm$^{-3}$ and $M_{1-10} = 11.28 \pm 9.05$ µg m$^{-3}$. During major LRT events, the coarse mode mass concentration typically increases by about one order of magnitude, occasionally reaching peak concentrations of $M_{1-10} = 100$ µg m$^{-3}$.

Under wet season conditions (without LRT), the entire coarse mode can be characterized as broad monomodal distribution with large mean geometric diameter (4.2 µm in the volume size distributions). In contrast, during the dry season, the coarse mode appears to have a multimodal shape with a strong, rather narrow peak located at 2.0 µm (volume size distribution) and pronounced shoulder towards larger particles. The coarse mode shape during Saharan dust influence shows a monomodal distribution with its maximum at 2.4 µm (volume size distribution).

A closer look at the diurnal cycling of particle mass concentrations during wet season conditions (without LRT) reveals a maximum in coarse mode abundance during the night (i.e., around 01:00-02:00 LT) and a minimum during the afternoon hours, which is consistent with previous observations by Huffman et al. (2012) and Whitehead et al. (2016). As suggested by Huffman et al., these trends are driven by a combination of variable dispersal of biological aerosols, connected to environmental/meteorological variables, as well as the oscillating height of the atmospheric boundary layer that concentrates and dilutes local emissions. A pronounced secondary maximum in the coarse mode mass concentration at 08:00 LT could indicate increased sporulation rates due to the onset of solar radiation and continuing high relative humidity levels. In contrast, the diurnal pattern during LRT episodes shows highest coarse mode mass concentrations around 12:00 LT, collocated with the maximum in incoming solar radiation and increasing vertical mixing. This observation suggests that the intrusion of LRT aerosols into the near-surface boundary layer occurs via convective mixing with (lofted) aerosol layers at higher altitudes. After sunset, as soon as less efficient vertical mixing cuts further supply, an instantaneous decrease in $M_{1-10}$ levels suggests efficient deposition of the LRT aerosol load to surfaces in the canopy space.

The arrival of African LRT plumes clearly corresponds with increased equivalent black carbon mass concentrations ($M_{BCe}$) and light scattering coefficients ($\sigma_{sp}$), underlining that the LRT aerosols typically represent mixtures of Saharan dust, biomass burning smoke, and sea spray (Talbot et al., 1990; Quinn et al., 1996; O'Dowd et al., 2008; Wang et al., 2016; Aller et al., 2017; Huang and Jaeglé, 2017). The degree of 'smokiness' of the arriving LRT plumes decreases towards the end of the wet season (Apr) which is consistent with the decreasing biomass burning activity in Africa, and simultaneously marks the cleanest periods at the ATTO site.

The complex emission, transport and transformation processes involved in the LRT of African dust into the Amazon basin is well represented in a recent modelling study by Q. Wang et al. (2016). Measured and modelled results of dust mass concentrations are in good agreement and encourage quantifying regional



deposition fluxes of individual chemical components. Based on these results, we estimated a dust deposition flux of 5-10 kg ha$^{-1}$a$^{-1}$ in the ATTO region, which is in good agreement with previous studies (Swap et al., 1992; Yu et al., 2015). Furthermore, a chemical analysis of aerosol filters with and without LRT influence confirmed an increase of crustal and sea salt elements during the LRT events. With this compositional in-

5 formation, we estimated elemental deposition fluxes in the ATTO region, which is particularly relevant for those elements that are considered as dust-related nutrients for the rain forest ecosystem.

Overall, this study provides a comprehensive overview of the physical and chemical properties of coarse mode aerosols in the Amazon basin, highlighting background PBAP and LRT conditions. The results serve as a basis for further in-depth studies on the complex coarse mode aerosol composition and cycling as well

as its significance for atmospheric, biogeochemical, and ecological processes.

## 5 Data availability

The data of the key results presented here, such as aerosol size distributions and aerosol mass concentration time series, have been deposited in supplementary tables for use in follow-up studies. For specific data re-

15 quests or detailed information on the deposited data, please refer to the corresponding authors.



**Acknowledgements.**

This work has been supported by the Max Planck Society (MPG). For the operation of the ATTO site, we
acknowledge the support by the German Federal Ministry of Education and Research (BMBF contract
01LB1001A) and the Brazilian Ministério da Ciência, Tecnologia e Inovação (MCTI/FINEP contract

01.11.01248.00) as well as the Amazon State University (UEA), FAPEAM, LBA/INPA and
SDS/CEUC/RDS-Uatumã. This paper contains results of research conducted under the Technical/Scientific
Cooperation Agreement between the National Institute for Amazonian Research, the State University of
Amazonas, and the Max-Planck-Gesellschaft e.V.; the opinions expressed are the entire responsibility of the
authors and not of the participating institutions. We highly acknowledge the support by the Instituto Nacion-

al de Pesquisas da Amazônia (INPA). We would like to especially thank all the people involved in the tech-
nical, logistical, and scientific support of the ATTO project, in particular Jürgen Kesselmeier, Carlos Alberto
Quesada, Susan Trumbore, Reiner Ditz, Matthias Sörgel, Thomas Disper, Thomas Klimach, Andrew Cro-
zier, Uwe Schulz, Steffen Schmidt, Alessandro Araùjo, Antonio Ocimar Manzi, Alcides Camargo Ribeiro,
Hermes Braga Xavier, Elton Mendes da Silva, Nagib Alberto de Castro Souza, Adir Vasconcelos Brandão,

Amauri Rodriguês Perreira, Antonio Huxley Melo Nascimento, Feliciano de Souza Coelho, Thiago de Lima
Xavier, Josué Ferreira de Souza, Roberta Pereira de Souza, Bruno Takeshi, and Wallace Rabelo Costa. Fur-
ther, we thank the GoAmazon2014/5 team for the fruitful collaboration and discussions. Moreover, we thank
Tobias Könemann, Maria Praß, Jan-David Förster, Andrea Arangio, Emilio Rodríguez Caballero, Ovid O.
Krüger, and Oliver Lauer for their support and stimulating discussions. The authors gratefully acknowledge

the NOAA Air Resources Laboratory (ARL) for the provision of the HYSPLIT transport and dispersion
model and/or READY website (http://www.ready.noaa.gov) used in this publication.





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





**Table A1.** List of acronyms.

| Acronym | Description |
| --- | --- |
| AIRS | Atmospheric infrared sounder |
| AOD | Aerosol optical depth |
| $AOD_{ROI,off}$ | Aerosol optical depth with the region $ROI_{off}$ |
| ATTO | Amazon Tall Tower Observatory |
| $BC_e$ | Equivalent black carbon |
| BT | Back trajectory |
| CALIPSO | Cloud Aerosol Lidar and Infrared Pathfinder Satellite Observation |
| CALIOP | Cloud-Aerosol Lidar with Orthogonal Polarization |
| CCD | Charge coupled device |
| CCN | Cloud Condensation Nuclei |
| CN | Condensation Nuclei |
| CPC | Condensation Particle Counter |
| CA | Cluster Analysis |
| CO | Carbon Monoxide |
| E | East |
| EDXRF | Energy dispersive x-ray fluorescence |
| ENE | East-northeast |
| ENSO | El Niño-Southern Oscillation |
| ESE | East-southeast |
| FoO | Frequency of occurrence of trajectories |
| FWHM | Full width at half maximum |
| GEOS-Chem | Goddard Earth Observing System coupled with chemistry |
| GIOVANNI | Geospatial Interactive Online Visualization and Analyze Infrastructure |
| GoAmazon2014/5 | Green Ocean Amazon 2014/5 |
| GEOS-Chem | Goddard Earth Observing System with Chemistry |
| HYSPLIT | Hybrid Single Particle Lagrangian Trajectory Model |
| IN | Ice Nuclei |
| IOP | Intensive Observation Period |
| ITCZ | Intertropical convergence zone |
| $\log(x)$ | $\log_{10}(x)$, i.e. the logarithm of $x$ to base 10 |
| LRT | Long-Range Transport |
| LT | Local time |
| MAAP | Multi-Angle Absorption Photometer |
| MAC | Mass Absorption Coefficient |
| MODIS | Moderate Resolution Imaging Spectroradiometer |
| NanoMOUDI | *2nd generation Micro-Orifice Uniform Deposit Impactor* |
| NASA | *National Aeronautics and Space Administration* |
| NCEP | National Center for Environmental Prediction |
| NE | Northeast |
| NOAA | National Oceanic and Atmospheric Administration |
| OPS | Optical Particle Sizer |
| PBAP | Primary Biological Aerosol Particle |
| PSL | Polystyrene Latex |
| RH | Relative Humidity |
| $ROI_{off}$ | Offshore region of interest |
| $ROI_{ATTO}$ | ATTO region of interest |
| SAL | Saharan Air Layer |
| SE | Southeast |
| SOM | Secondary Organic Matter |
| SMPS | Scanning Mobility Particle Sizer |
| SOA | Secondary Organic Aerosol |
| TRMM | Tropical Rainforest Measuring Mission |
| TSP | Total Suspended Particles |
| UHSAS | Ultra-High Sensitive Aerosol Spectrometer |
| UTC | Coordinated Universal Time |
| UV-APS | Ultra-Violet Aerodynamic Particle Sizer |
| VOC | Volatile Organic Compounds |
| WIBS | Wideband Integrated Bioaerosol Sensor |



**Table A2.** List of symbols.

| Symbol | Description |
|---|---|
| $D$ | Aerosol particle diameter, µm |
| $D_o$ | Optical particle diameter, µm |
| $D_a$ | Aerodynamic particle diameter, µm |
| $D_p$ | Physical or geometric particle diameter, µm |
| $D_{cut}$ | Cut-size diameter of Nano-MOUDI |
| $f_{BT,clusters}$ | Frequency of occurrence of back trajectory clusters |
| $\lambda$ | Wavelength |
| $M$ | Aerosol mass concentration, µg m$^{-3}$ |
| $M_{0.3-1}$ | Aerosol mass concentration for particle from 0.3 to 1 µm, µg m$^{-3}$ |
| $M_{2.5-10}$ | Mass concentration from 2.5 to 10 µm, µg m$^{-3}$ |
| $M_{0.3-10}$ | Mass concentration from 0.3 to 10 µm, µg m$^{-3}$ |
| $M_{1-10}$ | Mass concentration from 1 to 10 µm (coarse mode), µg m$^{-3}$ |
| $M_{BCe}$ | Mass concentration of black carbon equivalent, µg m$^{-3}$ |
| $N$ | Number concentration, cm$^{-3}$ |
| $c_{CO}$ | Carbon monoxide, ppb |
| $N_{Ait}$ | Number concentration of Aitken mode, cm$^{-3}$ |
| $N_{acc}$ | Number concentration of accumulation mode, cm$^{-3}$ |
| $N_c$ | Number concentration of coarse mode, cm$^{-3}$ |
| $N_{0.3-1}$ | Number concentration from 0.3 to 1 µm, cm$^{-3}$ |
| $N_{0.3-10}$ | Number concentration from 0.3 to 10 µm, cm$^{-3}$ |
| $N_{1-10}$ | Number concentration from 1 to 10 µm, cm$^{-3}$ |
| $N_{total}$ | Total number concentration (>5 nm), cm$^{-3}$ |
| $dN/dlogD_o$ | Number size distribution (cm$^{-3}$) |
| $dS/dlogD_o$ | Surface size distribution (µm$^2$ cm$^{-3}$) |
| $dV/dlogD_o$ | Volume size distribution (µg m$^{-3}$) |
| $P_{BT}$ | Precipitation from HYSPLIT back trajectory model, mm |
| $P_{TRMM}$ | Precipitation rate from Tropical Rainfall Measurement Mission, mm h$^{-1}$ |
| $\rho_{0.85}$ | Aerosol density at 0.8 g m$^{-3}$, g cm$^{-3}$ |
| $\rho_{1.0}$ | Aerosol density at 1.0 g m$^{-3}$, g cm$^{-3}$ |
| $\rho_{1.2}$ | Aerosol density at 1.2 g m$^{-3}$, g cm$^{-3}$ |
| $\rho_{2.0}$ | Aerosol density at 2.0 g m$^{-3}$, g cm$^{-3}$ |
| $\sigma_{sp}$ | Aerosol light scattering coefficient, Mm$^{-1}$ |




**Table 1.** List of African LRT episodes observed at the ATTO site from February 2014 until April 2017. The mean and maximum $M_{1-10}$ values characterize the relative intensity of the dust pulses. For LRT episodes that contain larger data gaps, the $M_{1-10}$ values were put in parentheses. Note that the mean and maximum $M_{1-10}$ values shown here are based on a density of 1 g cm$^{-3}$, see further information in Sect. 2.3. Since typical densities of dust and sea salt components in the LRT plumes are higher (see Table 1), the shown $M_{1-10}$ values represent lower limit values (compare Table 2).

| LRT | Date [UTC] | | $M_{1-10}$ [µg m$^{-3}$] | |
|---|---|---|---|---|
| Episodes | Start | End | mean | max |
| 2014_1 | 2014-Feb-02 | 2014-Feb-05 | 7.60 | 22.32 |
| 2014_2 | 2014-Feb-09 | 2014-Feb-21 | (12.46) | (77.04) |
| 2014_3 | 2014-Mar-02 | 2014-Mar-12 | (14.90) | (41.67) |
| 2014_4 | 2014-Mar-20 | 2014-Mar-24 | 10.11 | 26.44 |
| 2014_5 | 2014-Mar-28 | 2014-Apr-01 | 7.07 | 15.30 |
| 2014_6 | 2014-Apr-03 | 2014-Apr-05 | 7.37 | 16.59 |
| 2014_7 | 2014-Apr-08 | 2014-Apr-14 | (10.84) | (24.40) |
| 2015_1 | 2015-Feb-02 | 2015-Feb-18 | (8.47) | (23.89) |
| 2015_2 | 2015-Feb-27 | 2015-Mar-09 | (9.99) | (38.81) |
| 2015_3 | 2015-Mar-11 | 2015-Mar-14 | 8.74 | 25.55 |
| 2015_4 | 2015-Mar-18 | 2015-Mar-25 | 9.04 | 39.75 |
| 2015_5 | 2015-Apr-02 | 2015-Apr-10 | (19.17) | (59.65) |
| 2016_1 | 2015-Dec-06 | 2015-Dec-12 | 17.80 | 71.34 |
| 2016_2 | 2015-Dec-22 | 2015-Dec-25 | 9.48 | 16.92 |
| 2016_3 | 2015-Dec-28 | 2016-Jan-02 | 11.64 | 30.75 |
| 2016_4 | 2016-Jan-04 | 2016-Jan-11 | 11.97 | 40.85 |
| 2016_5 | 2016-Jan-20 | 2016-Jan-31 | (8.14) | (20.83) |
| 2016_6 | 2016-Feb-05 | 2016-Feb-22 | (12.88) | (62.44) |
| 2016_7 | 2016-Mar-01 | 2016-Mar-05 | 10.08 | 37.49 |
| 2017_1 | 2016-Dec-28 | 2017-Jan-13 | 9.73 | 25.75 |
| 2017_2 | 2017-Feb-08 | 2017-Feb-16 | 16.25 | 46.71 |
| 2017_3 | 2017-Feb-25 | 2017-Feb-28 | (5.38) | (14.34) |
| 2017_4 | 2017-Mar-10 | 2017-Mar-15 | 6.51 | 28.35 |
| 2017_5 | 2017-Mar-31 | 2017-Apr-06 | 22.81 | 106.17 |
| 2017_6 | 2017-Apr-13 | 2017-Apr-15 | 6.45 | 31.51 |
| 2017_7 | 2017-Apr-20 | 2017-Apr-24 | 7.64 | 27.60 |





**Table 2.** Average aerosol number and mass concentrations for the size ranges of 0.3-1.0 μm, 1.0-10.0 μm, 0.3-10.0 μm, and different seasons in the period of Feb 2014 – Apr 2017: (i) wet season excluding long-range transport (LRT) episodes, (ii) LRT episodes, (iii) transition periods, (iv) dry season, and (v) average of the entire data period (see Sect. 2.11). Total aerosol number concentration (> 4 nm, CPC-based) and black carbon ($BC_e$) mass concentrations (MAAP-based) are included for comparison. All results are reported as mean, median, and one standard deviation (std. dev.). The OPS-retrieved mass concentrations were calculated for 3 different densities ($\rho$=0.85; 1.0 and 1.2 g m⁻³). For the LRT episodes, also $\rho$=2.0 g m⁻³ was taken into account. The mass concentrations for the different densities do not scale linearly since the particle loss correction according to Sect. 2.3 has been taken into account.

*Aerosol number concentrations*

| Season | Density [g cm⁻³] | $N_{0.3-1}$ [cm⁻³] mean | median | std. dev. | $N_{1-10}$ [cm⁻³] mean | median | std. dev. | $N_{0.3-10}$ [cm⁻³] mean | median | std. dev. | $N_{1-10}/N_{0.3-10}$ [%] Mean | median | $N_{total}$ [cm⁻³] mean | median | std. dev. |
|---|---|---|---|---|---|---|---|---|---|---|---|---|---|---|---|
| Wet |  | 6.11 | 4.96 | 5.06 | 0.42 | 0.33 | 0.34 | 6.53 | 5.32 | 5.28 | 6.43 | 6.20 | 336 | 285 | 209 |
| LRT | 1.0 | 23.16 | 15.76 | 34.54 | 2.03 | 1.47 | 1.87 | 25.19 | 17.56 | 35.50 | 8.06 | 8.37 | 347 | 302 | 234 |
| Trans |  | 31.83 | 15.20 | 66.71 | 0.81 | 0.65 | 0.75 | 32.65 | 16.01 | 67.25 | 2.48 | 4.06 | 790 | 666 | 547 |
| Dry |  | 70.69 | 57.80 | 57.60 | 1.15 | 1.10 | 0.81 | 71.85 | 58.92 | 58.13 | 1.60 | 1.87 | 1508 | 1339 | 785 |
| All |  | 32.95 | 23.43 | 40.98 | 1.10 | 0.89 | 0.94 | 34.06 | 24.45 | 41.54 | 4.64 | 5.13 | 745 | 648 | 444 |

*Aerosol mass concentrations*

| Season | Density [g cm⁻³] | $M_{0.3-1}$ [μg m⁻³] mean | median | std. dev. | $M_{1-10}$ [μg m⁻³] mean | median | std. dev. | $M_{0.3-10}$ [μg m⁻³] mean | median | std. dev. | $M_{1-10}/M_{0.3-10}$ [%] mean | median | $M_{BCe}$ [μg m⁻³] mean | median | std. dev. |
|---|---|---|---|---|---|---|---|---|---|---|---|---|---|---|---|
| Wet | 0.85 | 0.17 | 0.14 | 0.14 | 3.30 | 2.83 | 2.21 | 3.47 | 3.03 | 2.26 | 95.10 | 93.54 | 0.02 | 0.01 | 0.03 |
|  | 1.00 | 0.20 | 0.16 | 0.16 | 4.04 | 3.47 | 2.72 | 4.25 | 3.70 | 2.78 | 95.06 | 93.78 |  |  |  |
|  | 1.20 | 0.24 | 0.19 | 0.19 | 5.21 | 4.45 | 3.54 | 5.45 | 4.73 | 3.61 | 95.59 | 94.16 |  |  |  |
| LRT | 0.85 | 0.74 | 0.63 | 1.15 | 10.94 | 8.78 | 8.78 | 11.82 | 9.56 | 10.08 | 92.55 | 91.84 | 0.17 | 0.13 | 0.15 |
|  | 1.00 | 0.87 | 0.63 | 1.15 | 11.28 | 9.05 | 9.05 | 12.16 | 9.84 | 10.37 | 92.76 | 91.97 |  |  |  |
|  | 1.20 | 1.04 | 0.76 | 1.39 | 14.26 | 11.45 | 12.46 | 12.75 | 10.34 | 10.91 | 93.18 | 92.26 |  |  |  |
|  | 2.00 | 1.74 | 1.26 | 2.32 | 38.72 | 31.06 | 35.94 | 40.48 | 32.66 | 36.76 | 95.65 | 95.10 |  |  |  |
| Trans | 0.85 | 0.81 | 0.41 | 1.71 | 4.29 | 4.02 | 2.80 | 5.11 | 4.56 | 3.82 | 84.03 | 88.08 | 0.17 | 0.10 | 0.20 |
|  | 1.00 | 0.95 | 0.47 | 2.01 | 5.24 | 4.90 | 3.46 | 6.19 | 5.56 | 4.62 | 84.65 | 88.13 |  |  |  |
|  | 1.20 | 1.14 | 0.58 | 2.41 | 6.76 | 6.30 | 4.50 | 7.90 | 7.10 | 5.83 | 85.56 | 88.68 |  |  |  |
| Dry | 0.85 | 1.71 | 1.37 | 1.56 | 5.29 | 5.20 | 2.19 | 7.00 | 6.79 | 3.23 | 75.58 | 76.60 | 0.35 | 0.30 | 0.20 |
|  | 1.00 | 2.01 | 1.61 | 1.84 | 6.47 | 6.36 | 2.69 | 8.48 | 8.25 | 3.88 | 76.30 | 77.09 |  |  |  |
|  | 1.20 | 2.41 | 1.93 | 2.21 | 8.32 | 8.15 | 3.47 | 10.73 | 10.44 | 4.84 | 77.52 | 78.05 |  |  |  |
| All | 0.85 | 0.86 | 0.64 | 1.14 | 5.95 | 5.21 | 4.18 | 6.85 | 5.99 | 4.85 | 86.81 | 87.51 | 0.18 | 0.14 | 0.15 |
|  | 1.00 | 1.01 | 0.72 | 1.29 | 6.76 | 5.95 | 4.68 | 7.77 | 6.84 | 5.41 | 87.19 | 87.74 |  |  |  |
|  | 1.20 | 1.21 | 0.86 | 1.55 | 8.04 | 7.11 | 5.47 | 9.21 | 8.15 | 6.30 | 87.96 | 88.29 |  |  |  |



**Table 3.** Coarse mode number and mass concentrations from selected earlier studies in the Amazon and other forested locations.

| Location and year of measurement | Aerosol size range [μm] | Instrument/technique | Season | N [cm⁻³] (Mean ± 1 std. dev.) | M [μg m⁻³] | Reference | Comments |
|---|---|---|---|---|---|---|---|
| Primary rain forest site, Reserva Biologica Jarú, Rondônia, Brazil, 1999 | ~2 - 10 | Gravimetric filter analysis | Dry season (Sep - Oct) | -- | 6.5 ± 6.1 | (Artaxo et al., 2002) | Comprehensive LBA study in biomass burning region on the effect of fires on aerosol load |
| | | | Wet season (Jan - May) | -- | 5.1 ± 2.6 | | |
| Pasture site, Rondônia, Brazil, 1999 | | | Dry season (Sep - Oct) | -- | 17.8 ± 11.7 | | |
| | | | Wet season (Jan - May) | -- | 5.7 ± 3.1 | | |
| Primary rain forest site, Reserva Biologica Jarú, Rondônia, Brazil, 1999 | ~2 - 10 | Gravimetric filter analysis | Dry season (Sep - Oct) | -- | 6.6 ± 3.0 | (Guyon et al., 2003) | Systematic analysis of Amazonian aerosol composition in wet and dry season |
| | | | Wet season (Apr - May) | -- | 3.8 ± 1.3 | | |
| Primary rain forest site, Balbina site, Amazonia, Brazil, 2001 | ~2 - 10 | Gravimetric filter analysis | Transition period wet to dry season (Jul) | -- | 3.9 ± 1.4 | (Graham et al., 2003) | Short-term study on diurnal aerosol variability |
| Primary rain forest site, ZF2 site, Amazonia, Brazil, 2008-2011 | ~2 - 10 | Gravimetric filter analysis | Dry season | -- | 4.4 ± 2.4 | (Artaxo et al., 2013b) | 4-year study at remote central Amazonian site |
| | | | Wet season | -- | 5.0 ± 2.0 | | |
| Rain forest, Amazon Basin (Rondonia) | ~2 - 10 | Gravimetric filter analysis | Dry season (Jun - Dec) | -- | 10.2 ± 9.0 | (Artaxo et al., 2013b) | Multi-year study (2009-2012) in Amazonian region that is subject to strong land-use change and biomass burning |
| | | | Wet season (Jan - May) | -- | 8.8 ± 5.3 | | |
| Primary rain forest site, ZF2 site, Amazonia, Brazil, 2008 | ~0.7 - 10 | UV-APS | Wet season (Feb-Mar), "high dust" conditions | 0.93 | 3.89 | (Huffman et al., 2012b) | FBAP analysis during AMAZE-08 |
| | | | Wet season (Feb-Mar), "low dust" conditions | 0.26 | 1.63 | | |
| | | | Wet season (Feb-Mar), campaign average | 0.55 | 2.49 | | |
| Primary rain forest site, ZF2 site, Amazonia, Brazil, 2013 | > 1 | WIBS | Transition period wet to dry season (Jul) | 0.46 | -- | (Whitehead et al., 2016) | Short-term campaign of FBAP |
| Boreal forest site, SMEAR2 station, Hyytiälä, Finland | ~0.7 - 10 | UV-APS | Spring | 0.43 | -- | (Schumacher et al., 2013) | Long-term (>1 year) FBAP study in boreal forest environment |
| | | | Summer | 0.45 | -- | | |
| | | | Fall | 0.41 | -- | | |
| | | | Winter | 0.47 | -- | | |
| | | | Campaign average | 0.44 | -- | | |
| Semi-arid forest site, Manitou experimental forest, Colorado, USA | ~0.7 - 10 | UV-APS | Spring | 0.73 | -- | (Schumacher et al., 2013) | Long-term (>1 year) FBAP study in semi-arid forest environment |
| | | | Summer | 0.44 | -- | | |
| | | | Fall | 0.28 | -- | | |
| | | | Winter | 0.20 | -- | | |
| | | | Campaign average | 0.41 | -- | | |



**Table 4.** Estimated annual deposition of LRT-related elements in the ATTO region. The results are based on the modelled total dust aerosol deposition of 5-10 kg ha$^{-1}$ a$^{-1}$ as discussed in Sect. 3.3 in combination with the EDXRF results in Fig. 14. Note that these numbers represent only the part of the dust deposition gradient in Fig. 7 that includes the ATTO site (yellow area). For comparison, elemental deposition flux reported by Swap et al. (1992) are reported as well.

| Element | Estimated flux deposition [g ha$^{-1}$ a$^{-1}$] | |
| --- | --- | --- |
| | **This study** | **Swap et al. (1992)** |
| **Na** | 110 - 230 | 800 - 3400 [a] / 730 - 2900 [b] |
| **Mg** | 89 - 180 | -- |
| **Al** | 200 - 410 | -- |
| **Si** | 410 - 810 | -- |
| **P** | 9 - 17 | 11 - 46 [a] / 3 - 39 [b] |
| **S** | 140 - 270 | 140 - 2340 [a] / 400 - 1040 [b] |
| **Cl** | 130 - 260 | 2500 - 4900 [a] / 1400 - 5100 [b] |
| **K** | 110 - 230 | 230 - 870 [a] / 410 - 2300 [b] |
| **Ca** | 54 - 110 | -- |
| **Ti** | 17 - 33 | -- |
| **Mn** | 6 - 12 | -- |
| **Fe** | 120 - 240 | -- |

[a] Dust intrusion estimate
[b] Precipitation estimate





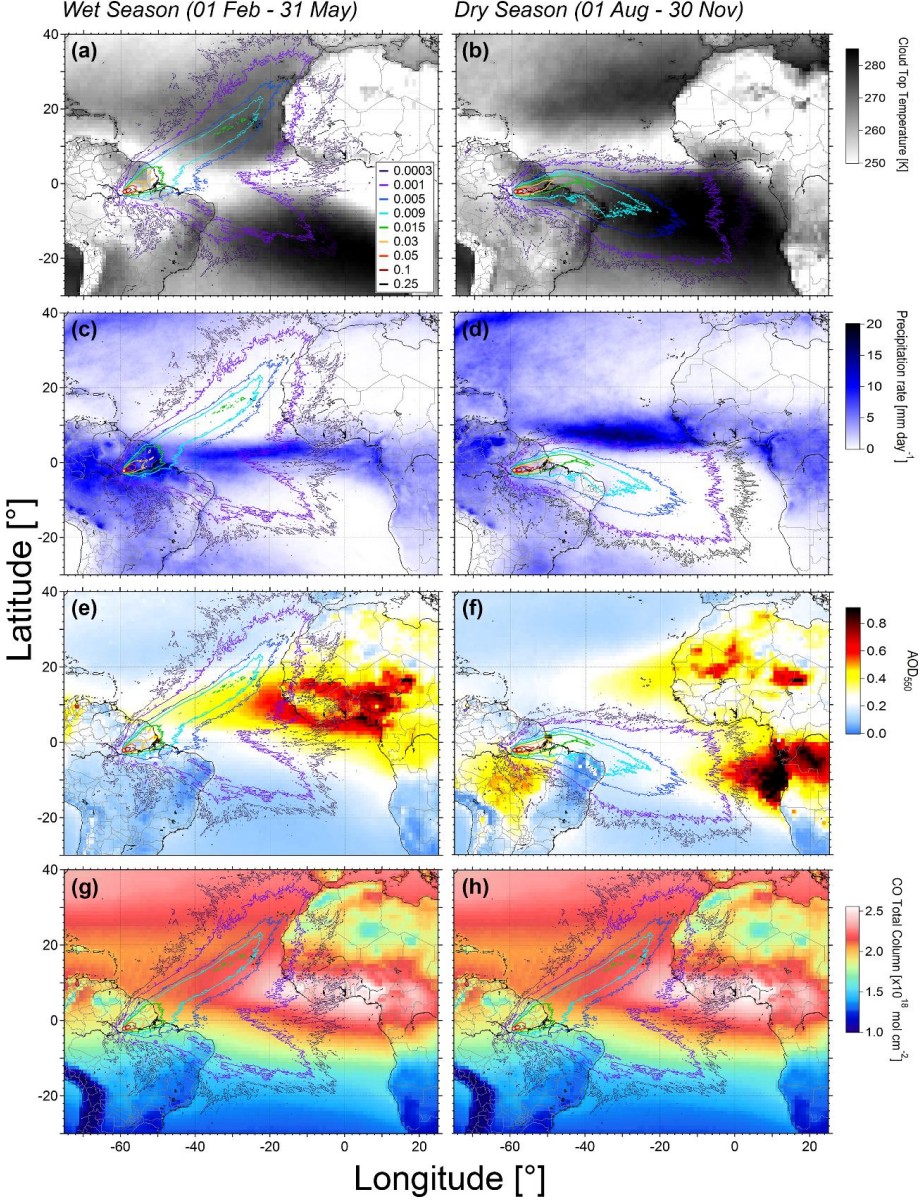

**Figure 1** Composite maps combining back trajectory ensembles with satellite data products contrasting wet and dry conditions in Amazonia. Multi-year time periods from 2008 to 2016 were averaged to retrieve representative results. All wet season data shown here represent averages of the months Feb-May and the dry season data represent averages of the months Aug-Nov. Back trajectory ensembles are shown as contour plots with contour lines representing fraction of occurrence of overpassing trajectories in a certain region as described in C. Pöhlker et al. (2017) (see legend in **a**). (**a** and **b**) MODIS-derived cloud top temperature data showing location of ITCZ belt with deep convective clouds and corresponding cold cloud tops. (**c** and **d**) TRMM-derived precipitation rate showing regions where strong rain-related aerosol scavenging is expected. The belt with strongest precipitation rates is collocated with the ITCZ. (**e** and **f**) MODIS-derived aerosol optical depth (AOD) illustrating the dominant aerosol sources in Africa and South America relevant for the ATTO site measurements. (**g** and **h**) AIRS-derived CO maps illustrating hot spots of biomass burning with strong CO emissions.


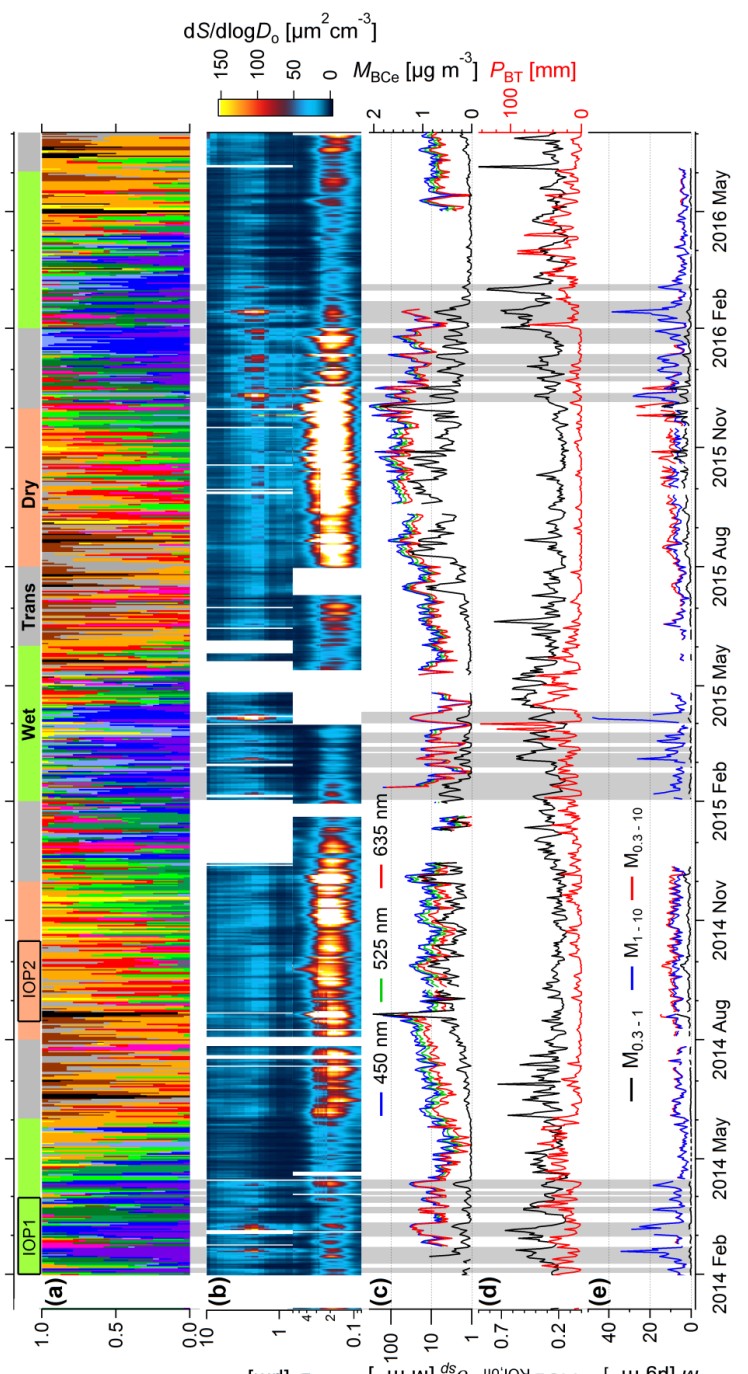

**Figure 2.** Overview of the long-term meteorological and aerosol time series emphasizing coarse mode data, in the context of atmospheric seasonality at the AT-TO site, with time series representing daily averages. (**a**) Frequency of occurrence, $f_{BT,clusters}$, of 15 back trajectory clusters as displayed in Fig. S4 (with identical color coding). (**b**) Image plot of aerosol surface size distribution for the size range from 80 nm to 10 μm. (**c**, left axis) Time series of the aerosol scattering coefficients, $\sigma_{sp}$, at three different wavelengths. (**c**, right axis) Time series of equivalent black carbon (BC$_e$) with mass absorption cross-sections (MAC) at 637 nm that represent conditions in the ATTO region (wet season: 11.4 m$^2$ g$^{-1}$; dry season: 12.3 m$^2$ g$^{-1}$). (**d**, left axis) Satellite-retrieved aerosol optical depth at 550 nm, area-averaged over offshore region of interest (ROI$_{off}$; see Fig. S4). The AOD$_{ROI,off}$ time series represents the average of the MODIS data sets from the satellites Aqua and Terra (**d**, right axis) HYSPLIT-retrieved accumulated precipitation $P_{BT}$ along the trajectory tracks. (**e**) Aerosol mass concentrations in the size ranges 0.3-10 μm ($M_{0.3-10}$), 1-10 μm ($M_{1-10}$), and 0.3-1 μm ($M_{0.3-1}$). Colored shading in the top of the figure visualizes the Amazonian seasons (see Sect. 2.11) as well as the intensive operation periods (IOPs) 1 and 2 of the GoAmazon2014/5 project. Grey vertical bands mark episodes when Saharan LRT aerosol was measured at the ATTO site (see Table 1).





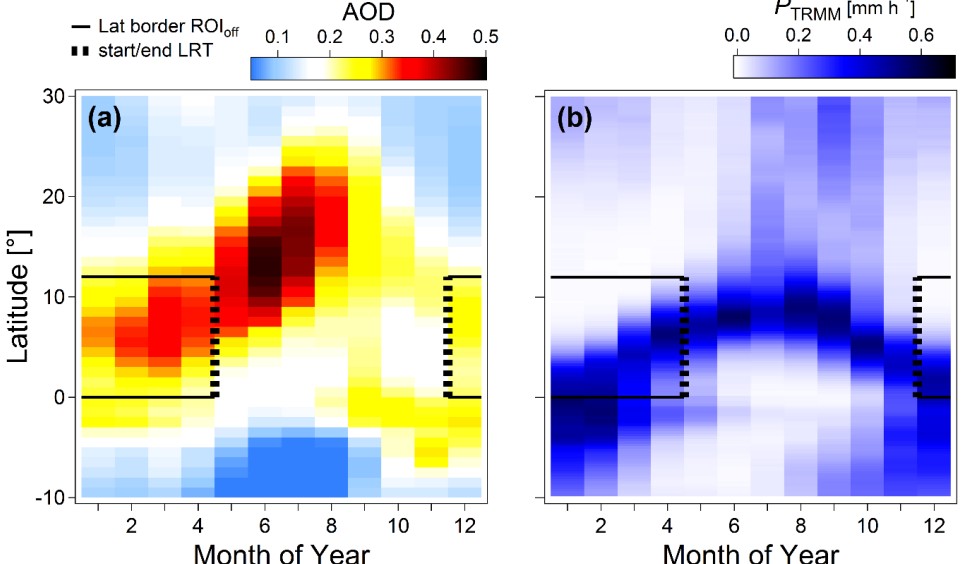

**Figure 3.** Longitude-averaged Hovmöller plots for (**a**) MODIS-derived aerosol optical depth (AOD) and (**b**) TRMM-derived precipitation rate, $P_{TRMM}$. The average AOD data represents a time period from Jul 2002 to Jun 2016. The average precipitation rate data represent a time period from Dec 1997 to Mar 2016. Long time periods have been chosen (maximum of available satellite data) to extract representative seasonal trends. The averaged longitude range of the plots from 49° W to 37° W corresponds with the longitude dimension of the ROI_off in Fig. S4. The latitudinal range of the Hovmöller plots spans from 10° S to 30° N. The solid black lines mark the latitudinal margins of the ROI_off (0° N to 12° N) and the black dashed lines mark start (Dec) and end (Apr) of the period when most African LRT aerosol transport towards the ATTO site has been observed.





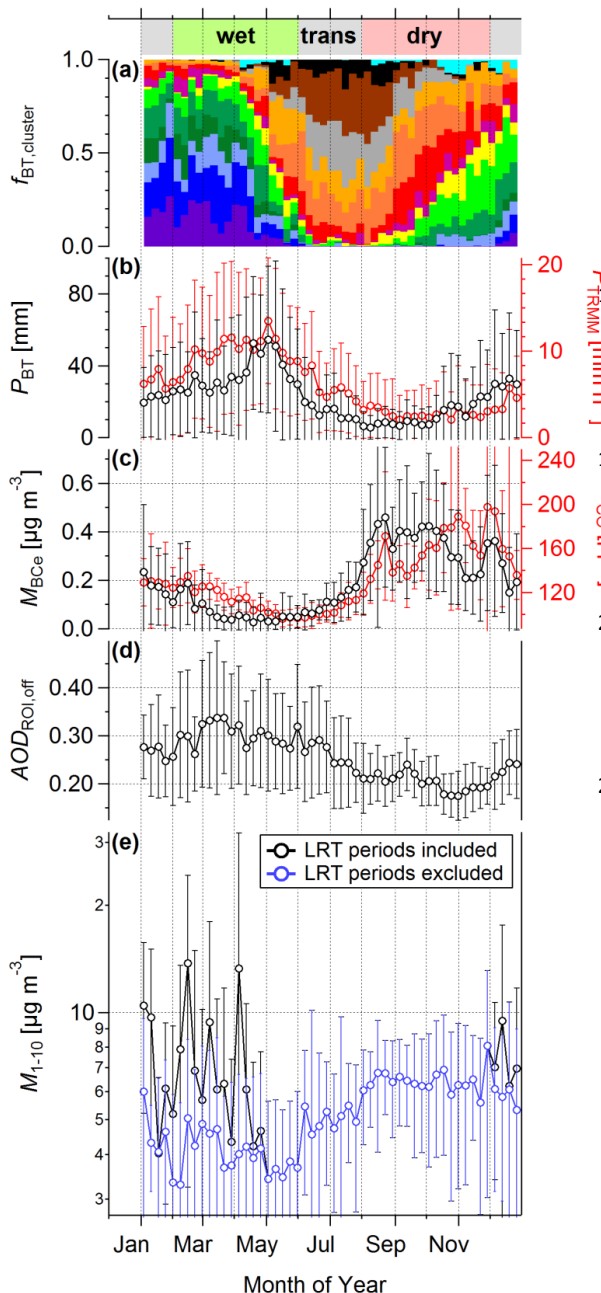

**Figure 4.** Seasonal cycle of selected meteorological, trace gas, and aerosol parameters in the central Amazon. (**a**) Weekly frequency of occurrence of back trajectory clusters according to C. Pöhlker et al. (2017). (**b**) Precipitation products $P_{BT}$, representing cumulative precipitation along back trajectory tracks, and $P_{TRMM}$, representing TRMM-derived precipitation within $ROI_{ATTO}$ (see Fig. S4). (**c**) Pollution tracers $M_{BCe}$ and $c_{CO}$. The $BC_e$ data includes ATTO and ZF2 site measurements, spanning from 2008 to 2017. CO data includes ATTO site measurements from 2012 to 2017. (**d**) MODIS-derived AOD data within $ROI_{off}$ as defined in Fig. S4, spanning from 2000 to 2016. (**e**) OPS-based coarse mode aerosol mass in two configurations: with and without LRT peaks according to Table 1. Data is shown as weekly averages with error bars representing one standard deviation.



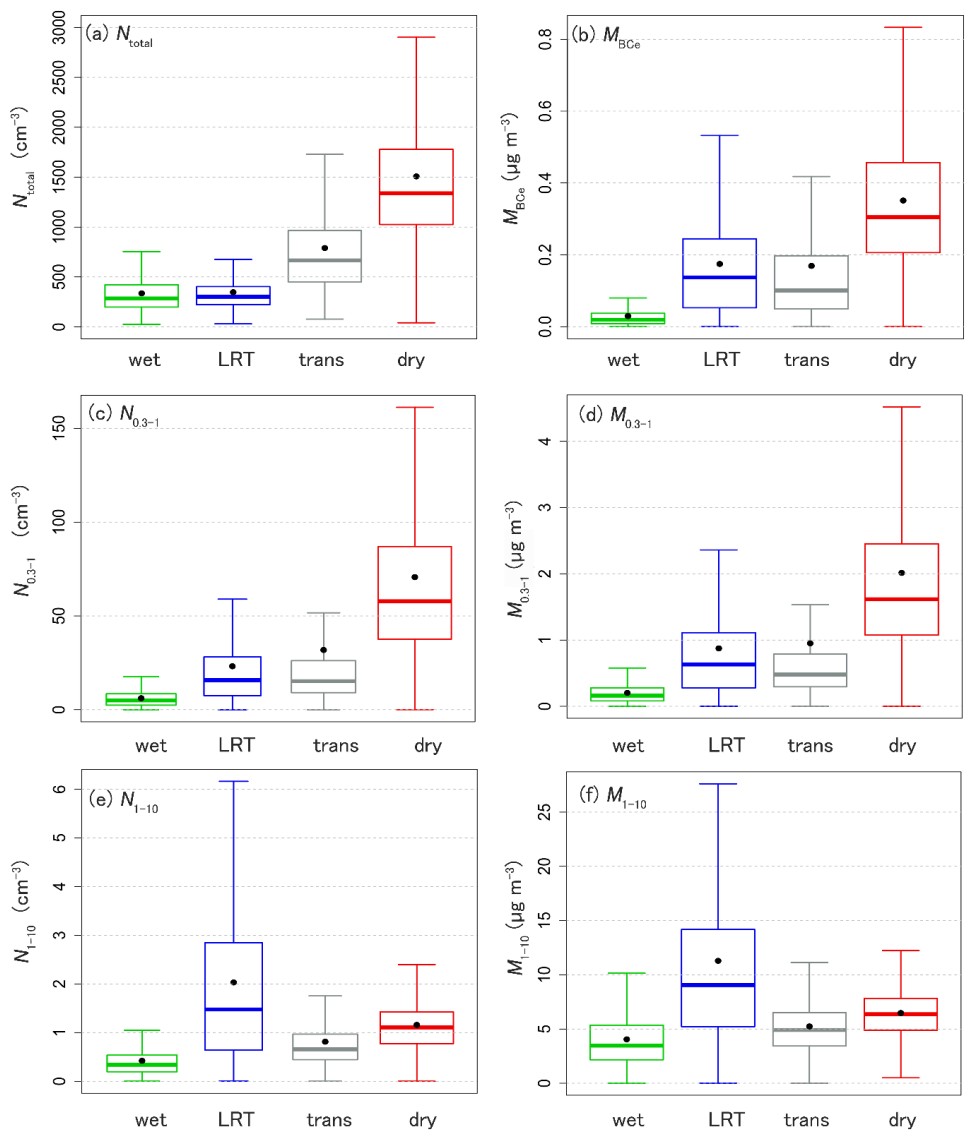

**Figure 5.** Statistical distributions (box-and-whisker plots) of (**a**) total aerosol number concentration (> 4 nm, CPC derived), (**b**) $BC_e$ mass concentration, $M_{BCe}$, (**c**) aerosol number concentration in the size range 0.3-1 µm, (**d**) aerosol mass concentration in the size range 0.3-1 µm, (**e**) aerosol number concentration in the size range 1-10 µm, and (**f**) aerosol mass concentration in the size range 1-10 µm. Wet season, LRT episodes, transition periods and dry season were defined according to Sect. 2.11. Data included here spans from Feb 2014 to Apr 2017. The boxes show the median (thick horizontal line), mean (black dot), 25 percentile (Q1, lower border of the box) and 75 percentile (Q3, upper border). The range between Q1 and Q3 is called interquartile range: IQR=Q3-Q1. The lower whisker shows the lowest measured value still within 1.5 IQR of the lower quartile, and the highest measured value still within 1.5 IQR of the upper quartile, see Tukey, 1977.



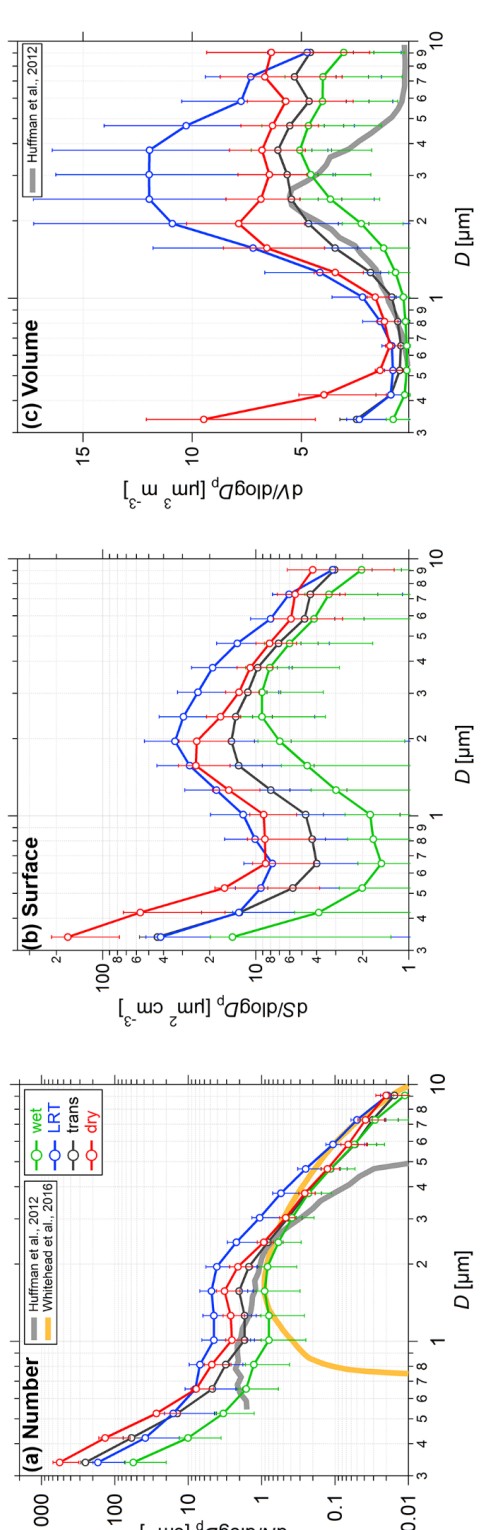

**Figure 6.** Seasonal variation of coarse mode aerosol size distributions, average over OPS data from Feb 2014 until Apr 2017: **(a)** Number size distributions, $dN/d\log D$, **(b)** surface size distributions, $dS/d\log D$, and **(c)** volume size distributions, $dV/d\log D$. The $dS/d\log D$ and $dV/d\log D$ distributions were calculated based on $dN/d\log D$ assuming spherical particles with standard density, $\rho_{1.0}$ (see Sect. 2.2). Markers represent median values with error bars as $25^{th}$ and $75^{th}$ percentiles. The size range of the OPS covers both, the coarse mode (1-10 µm) and the large diameter tail of the accumulation mode (0.3-1 µm). In terms of the coarse mode, quantitative information on its location, intensity, and shape were obtained. The accumulation mode information is semi-quantitative, since only a part of the mode is covered, which suffices to estimate the overall accumulation mode strength. For definition of the seasons refer to Sect. 2.11. Particle size distributions from the Amazonian studies by Huffman et al. (2012) and Whitehead et al. (2016) were included for comparison.



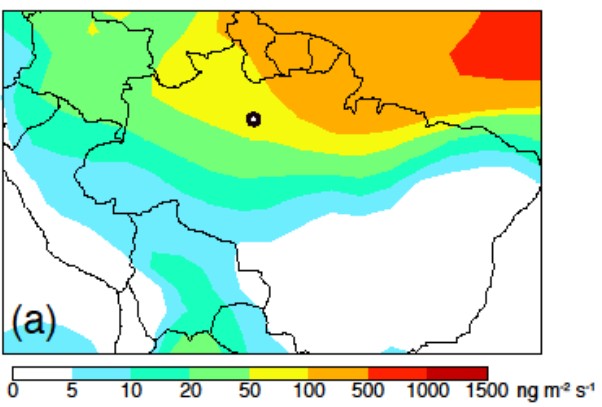

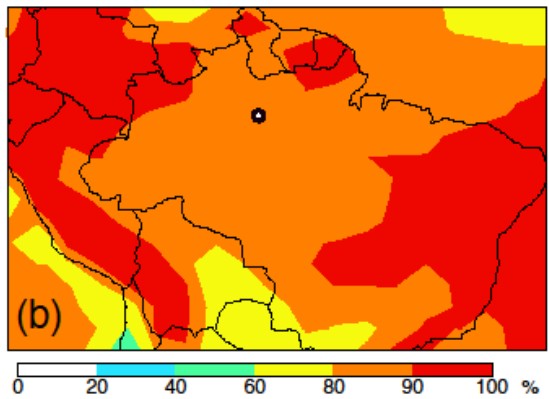

**Figure 7.** GEOS-Chem model results representing: (**a**) dust aerosol deposition flux and (**b**) contribution of wet dust deposition (rain out and wash out) relative to total deposition flux. Both data products were calculated from Jan to Apr 2014. The location of the ATTO site is represented by a black marker.



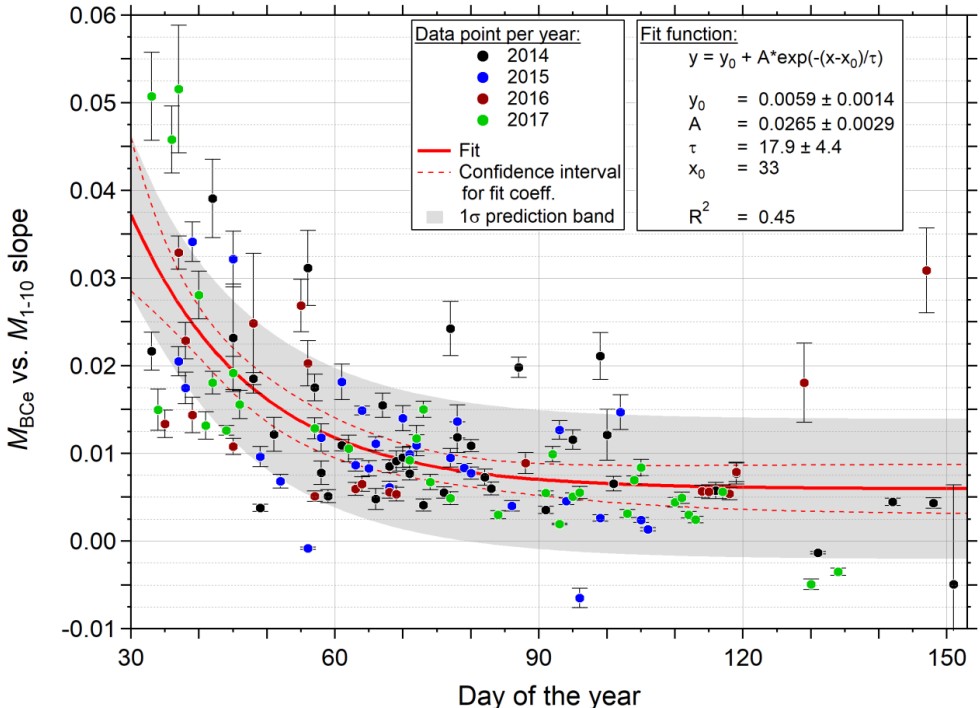

**Figure 8.** Temporal trend of the slopes from correlations between $M_{BCe}$ and $M_{1-10}$ from systematic bivariate regression fit analysis during the wet season period. Bivariate regression fits (with offset) have been performed every 1 d during wet season months Feb to May for the years 2014 to 2017. The results have been filtered and only those slope data points from regression fits with a clear correlation (i.e., with a correlation factor $R^2 > 0.5$) are shown here. The error bars represent the standard error of the slope. To emphasize the overall trend, an exponential fit has been added with its $1\,\sigma$ prediction band.



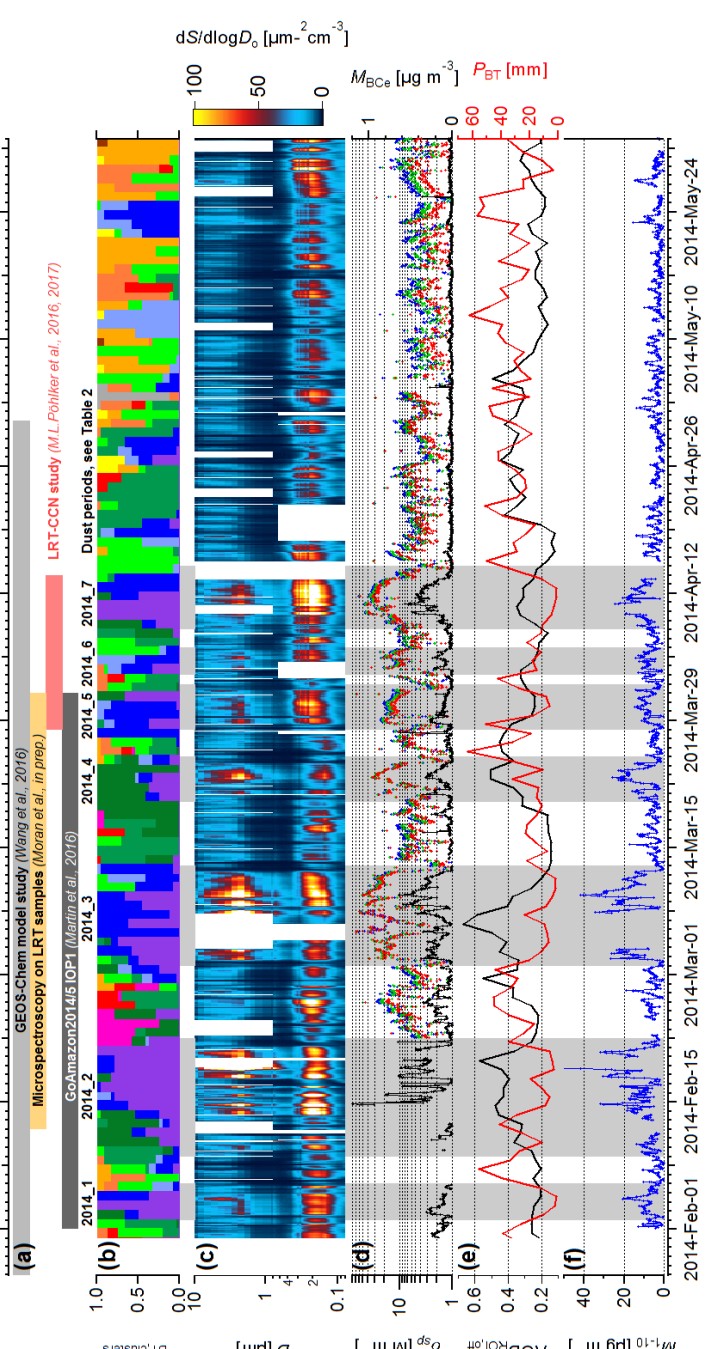

**Figure 9.** Meteorological and aerosol time series during the wet season 2014 case study. (**a**) Colored markers represent time periods of relevant intensive observation activities during 2014 case study. (**b**) Frequency of occurrence, $f_{BT, clusters}$, of 15 back trajectory clusters as displayed in Fig. S4 (with identical color coding). (**c**) Image plot of aerosol size surface distribution for the size range from 80 nm to 10 μm. (**d**, left axis) Time series of the aerosol scattering coefficients, $σ_{sp}$, at three different wavelengths. (**d**, right axis) Equivalent black carbon mass concentration, $M_{BCe}$, calculated assuming a mass absorption cross Section MAC of 11.4 $m^2$ $g^{-1}$ during the wet season. (**e**, left axis) Satellite-retrieved aerosol optical depth at 550 nm, area-averaged over offshore region of interest (ROI$_{off}$, see Fig. S4). The AOD$_{ROI,off}$ time series represents the average of the MODIS data sets from the satellites Aqua and Terra. (**e**, right axis) HYSPLIT-retrieved accumulated precipitation, $P_{BT}$, along the trajectory tracks. (**f**) Aerosol mass concentrations in the coarse mode 1-10 μm ($M_{1-10}$). Time series represent hourly averages, except $f_{BT,clusters}$, AOD$_{ROI,off}$, and $P_{BT}$. Grey vertical bands mark episodes when Saharan LRT aerosol was measured at the ATTO site (see Table 1). The $f_{BT, clusters}$, AOD$_{ROI,off}$ and $P_{BT}$ time series are shown as daily averages. The image plot of the aerosol size surface distribution as well as the $σ_{sp}$, $M_{BCe}$, and OPS mass concentrations time series are shown as daily averages.


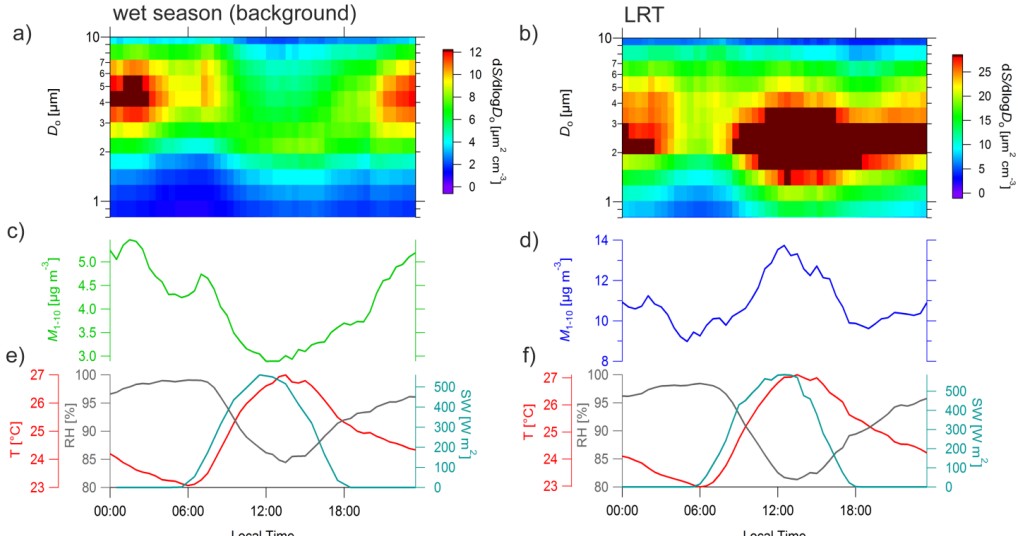

**Figure 10.** Diurnal cycles in selected aerosol and meteorological parameters during the wet season without major dust influence vs. LRT episodes within the time frame shown in Fig. 9. (**a**) and (**b**) image plots of aerosol size surface distribution for the size range from 0.8 to 10 µm. (**c**) and (**d**) coarse mode aerosol mass concentration $M_{1\text{-}10}$. (**e**) and (**f**) micrometeorological parameters, temperature, RH, and incoming short-wave solar radiation.



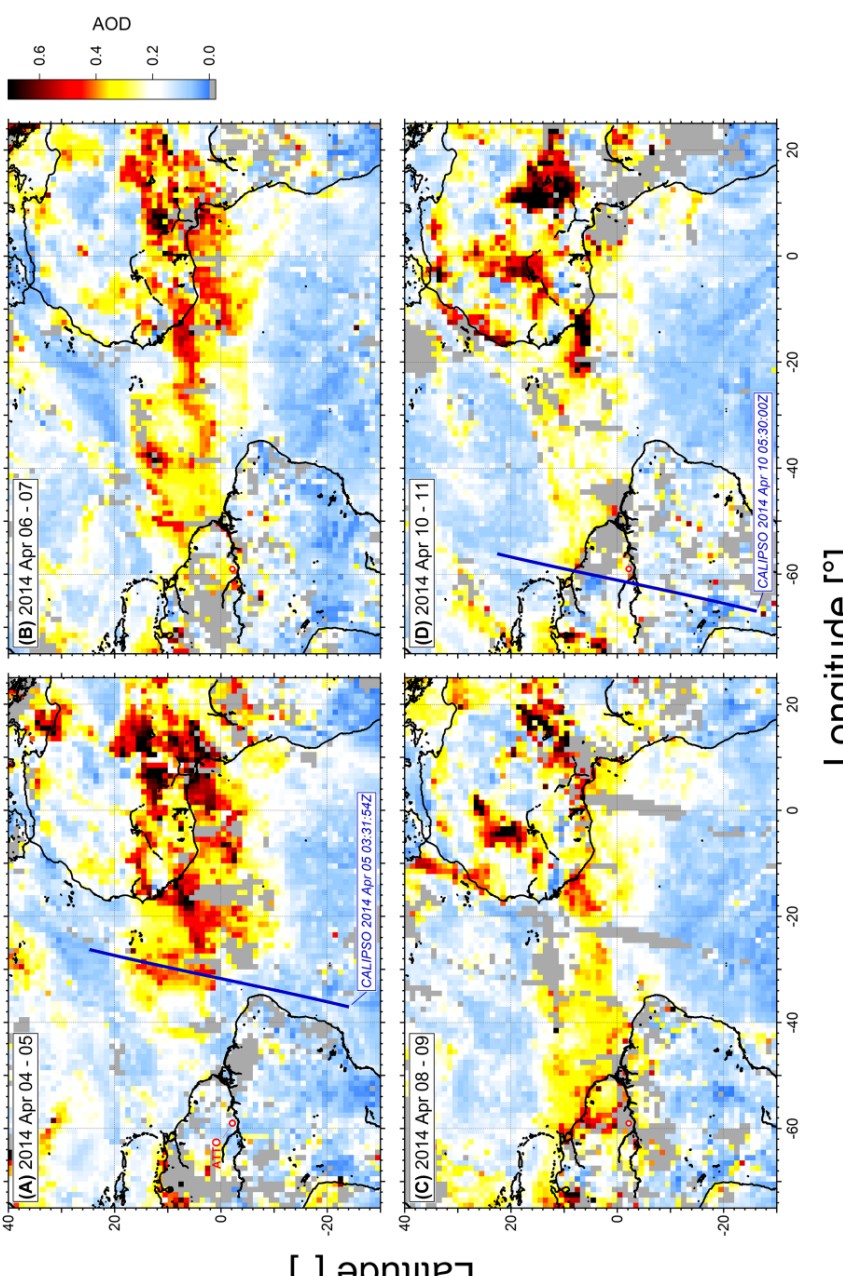

**Figure 11.** MODIS-derived AOD data showing Saharan dust outbreak, transatlantic passage, and intrusion into the Amazon Basin in the beginning of April 2014. Selected orbits of the CALIPSO spacecraft on April 05 and 10 are shown in (**a**) and (**d**). Corresponding CALIOP lidar profiles transecting the dust plume are displayed in Fig. 12.





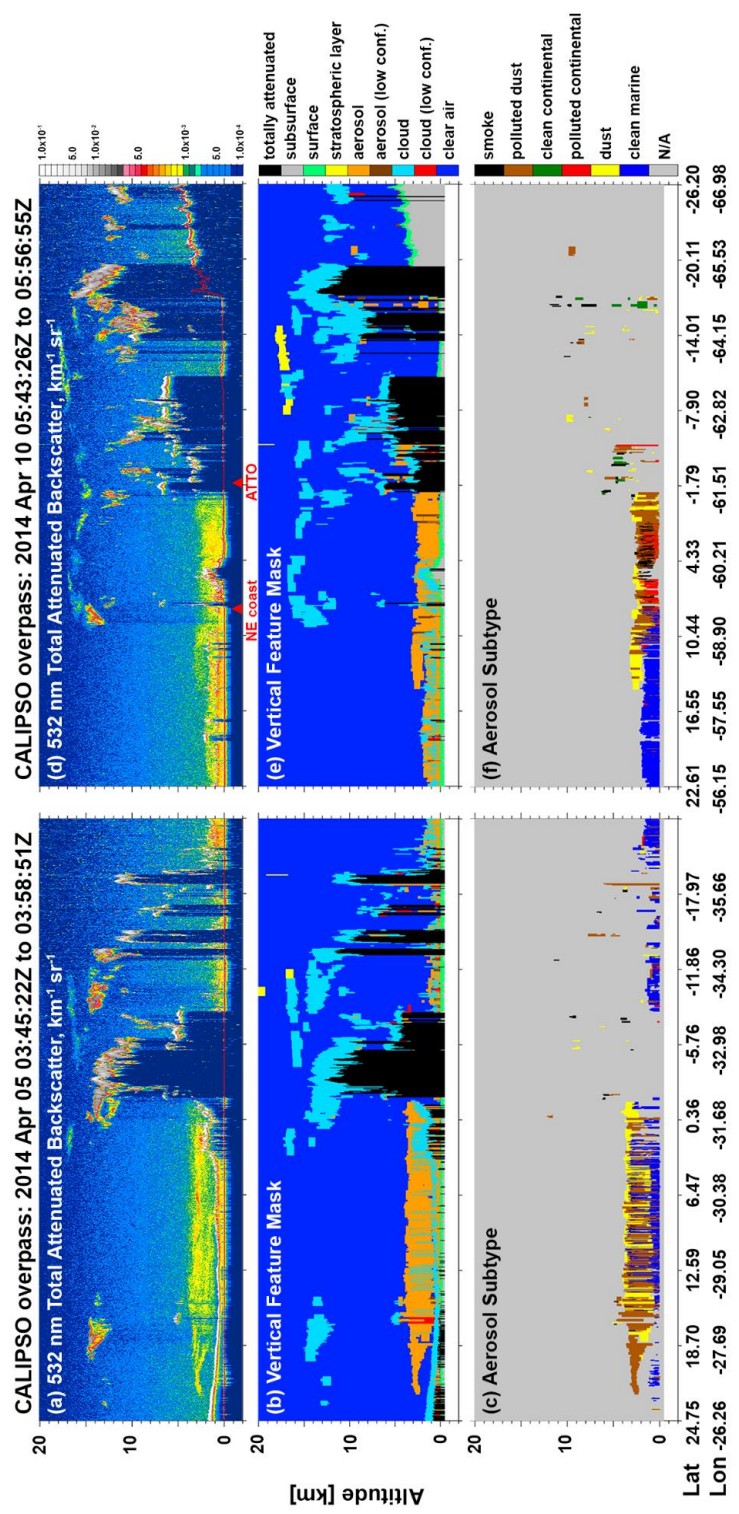

**Figure 12** CALIOP lidar profiles of the African dust plume from 05 and 10 April 2014. The corresponding satellite orbits are show as overlay with the MODIS AOD maps in Fig. 11. Red markers in (**b**) show position of the ATTO site and the north-east South American coast.





**Figure 13.** Wind data and precipitation daily averages from NCEP satellite data at a spatial resolution of 2.5 degrees from 02 to 13 April 2014 corresponding to 2014_7 period shown in Fig. 11.

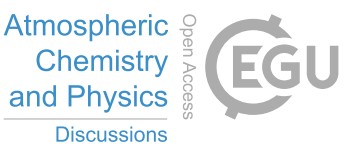

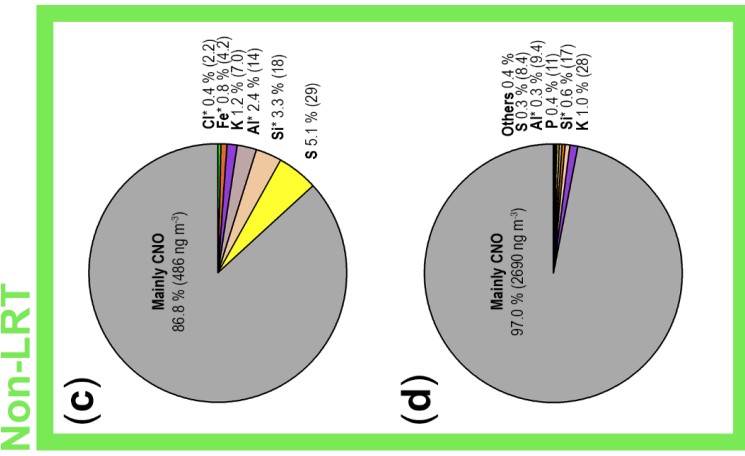

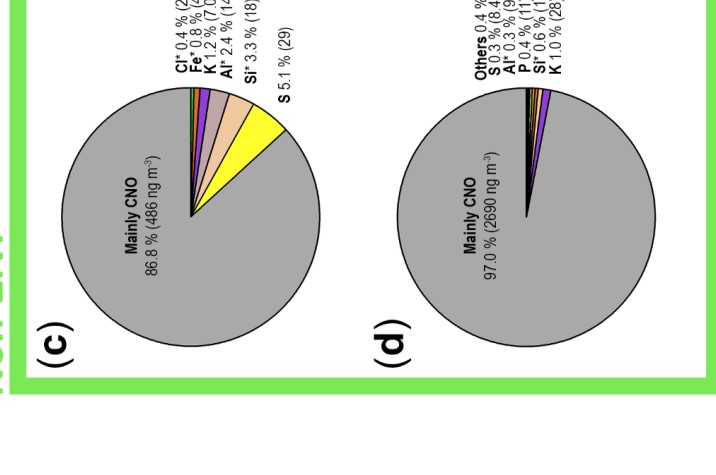

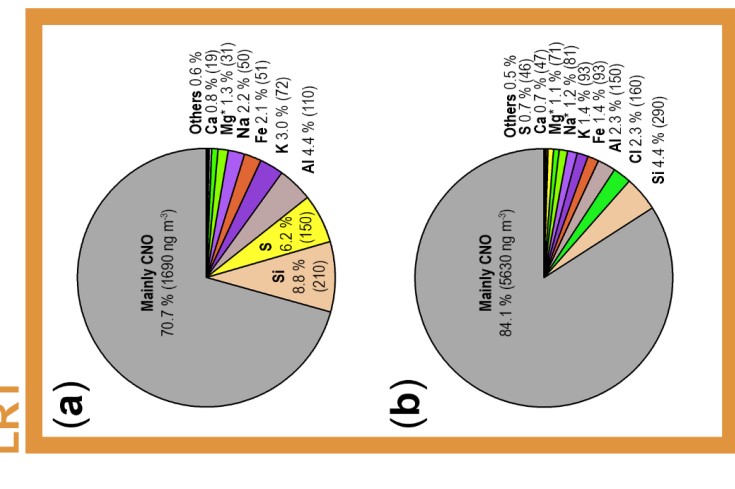

**Figure 14.** Mean relative mass fractions (in percent) and elemental mass concentrations (in ng m$^{-3}$, in parenthesis) of aerosol particles in the size fractions >2.5 µm and <2.5 µm for LRT episodes vs. periods without LRT influence (called non-LRT here). The sampling periods of the 5 LRT filters and the 4 non-LRT filters are specified in Sect. 2.4 and plotted in Fig. S9. The average elemental fractions are quantified relative to the average deposited mass on the LRT vs. non-LRT filters, respectively. The EDXRF analysis allows to quantify the masses of elements with atomic numbers larger 10 (i.e., starting with Na). Accordingly, the lighter elements in the periods 1 and 2 of the periodic table of the elements account for the remaining mass. The elements C, N, and O represent the predominant contributors here. The results visualized here are provided in detail in the corresponding Table S2.