# Peer review of "Long-term study on coarse mode aerosols in the Amazon rain forest with the frequent intrusion of Saharan dust plumes"

_Atmospheric Chemistry and Physics, 2017_

## Author Comment (AC1) · 14 Dec 2017

On figure 1, the composite maps the lower panel (g) "CO Total Column" is now corrected with the corresponding dry season map in the figure attached.

[Figure]

[Figure]

**Fig. 1.**

---

## Referee Comment (RC1) · Anonymous Referee #1 · 16 Feb 2018

General comments

In past the aerosol science has mostly been concentrated on fine particles, but scientists are starting to recognize the importance of the coarse fraction in biogeochemical cycling and also atmospheric processes such as ice nucleation.

The paper synthesizes a large amount of data on coarse particles gathered in the Amazon Tall Tower Observatory.

Large fraction of coarse aerosol in Amazon is of biogenic origin, consisting of fungal spores, bacteria, and plant debris. The manuscript provides a valuable source of information to reduce the large uncertainties that exist in understanding the amount and

emission fluxes of these particles.

The episodic nature of the transport of Saharan dust has made it challenging to quantify the amounts transported and deposited to Amazon Basin. The manuscript provides a detailed description of the necessary conditions for these transport episodes to occur.

Impressive amount of data has been brought together and analysed in this manuscript. The paper is well written and clearly understandable. I suggest publishing it with minor revisions.

Specific comments

1) Section 2.5, Fig S2 The comparison between the gravimetric and OPS derived aerosol concentrations is made for periods in dry season, where majority of the coarse mode particles is expected to be of primary biogenic origin with densities close to 1g/cm3. Would it be possible to compare the OPS derived and gravimetric mass concentrations also for the wet season or specific LRT episodes? Or at least estimate the implications of the densities of both major contributors to the long range transported coarse aerosol - dust and sea salt - being substantially higher than 1 g/cm3?

2) Page 15, lines 16-29. Is the precipitation along the back trajectories considered at ground level or 3D? Do you take into account the option of the dust plume transport above the rain clouds followed by downwards mixing?

Technical corrections

Page 3, line 10 – correct to "focused on "

Page 15, line 9 – correct "this study clearly shows"

---

## Referee Comment (RC2) · Anonymous Referee #2 · 19 Feb 2018

General Comments

This manuscript presents an unique and unprecedented multi-year time series of coarse mode aerosol measurements in Central Amazon. In the ATTO tower, it is even more representative of real remote aerosol characteristics. The manuscript also compares aerosol measurements with observations made on orbital platforms, and large-scale weather conditions, which further enriches the results presented here.

The paper is well written and deserves to be published in ACP. All proposed revisions by this referee are intended to improve the comprehension of the scientific results reported in this study.

[Figure]

Specific comments

- For most of figures: the authors preferred to differentiate curves in plots using different colors. There are recommendations to avoid this approach due to color-blinded readers. So, whenever possible, change and prefer figures using different gray tones and different line patterns (dot, crosses, diamonds, trace-dot, etc.). This is the case of the following figures in the manuscript: S1, S2a and S2b, S3, S4, S5, S6 S9, S12; 1, 2, 4a (colors), 4b-d (symbols are equal), 6, 7, 8 (use different symbols), 9, 10e-f.

Reference: https://www.nature.com/articles/nmeth.1618

- C. Pohlker et al. (2017) is cited several times in the text and is related to important features of this study. However, it is in reference list as "to be submitted". It is not correct to use so often a paper that is not even submitted. The same holds for Saturno et al. (2017) despite it is much less cited than C. Pohlker et al. 2017

- Change "Amazonian" to "amazonian" (first letter lower case) in the whole text. Amazonian is an adjective, not a location. Amazon is a location.

- p.5 lines 1 and 5: the author refers to "winter dust plumes". It is confusing because the manuscript is dealing with a phenomenon that happens in both hemispheres. Winter in one hemisphere is summer in the other. Instead use the name of specific months you are referring. Another option, use wet or dry season (with respect to Central Amazonia) whose months are well known.

- p.5 lines 34 and 36: repeated to close in the text the expression "whether and to what extent".

- p.13 line 28: "NE basin" to "NE Amazon basin".

- p.15 line 12: P_BT was not defined before.

- p.16 last line: there is no meaning in M_BCe = 0.02 +- 0.03. This discrepancy is due to the use of Standard Deviation in non-normal data. Use median and Interquartile

Range to avoid negative values in the confidence interval.

- p.19 lines 18-20: "We propose that the results obtained here for the year 2014 can be regarded as representative for a typical dust deposition scenario in the Amazon region, since 2014 was generally an 'average' year without 20 pronounced precipitation and circulation anomalies (M. Pöhlker et al., 2016; C. Pöhlker et al., 2017)." »>To state that 2014 was an "average year" is a strong affirmation, and should not use a "to be submitted" reference for such an assumption.

References: Rizzolo et al. (2016) was already published as final revised article in ACP: see https://www.atmos-chem-phys.net/17/2673/2017/acp-17-2673-2017.pdf and update it.

- Figure 2: it contains a lot of information in a single figure and it is somehow confusing to understand author's analysis and discussions due to the small size of curves. Improve it separating in different figures with larger sizes.

This figure also cites figure S4. To understand the figure it is mandatory to the reader to see supplement. So, it should not be in the supplement but in the main text. The same holds for Figure 3, 4, 9, 14. Also, (c) lacks BC equivalent in legend.

- Figure 4:

(a) very confusing to understand the actual meaning of colors in the figure. (c) and (e): change the experimental point symbol for the sake of clarity

- Figure 5: here you explain M_BCe, but it was previously used in Fig. 4 without any explanation. So, move it to previous figure.

- Figure 8: why some points show negative slope, with slope +- uncertainty not compatible with zero?

- Figure 9: in the upper legend it cites Moran et al. (in preparation). Not adequate.

- Figure 11: include in caption the meaning of gray areas in the map.

- Figure 12: in the caption: "are show overlay" > "are shown overlay"...

and (b) actually refers to figure (d).

- Figure 13, in caption: did you mean "NCEP Reanalysis" when writing "NCEP satellite"?

Supplementary material:

The manuscript cites too often figures containing in supplementary material. If a figure has to be cited frequently and it is important to the actual comprehension of the article context of the article it should be moved to the main text. This is the case of figures S4, S6 and S9, which should be inserted in the principal text.

Figure S2: at figure (c) the fitting seems to have forced parameter a = 0. It should be explained and/or justified it. Visual (separately) inspection of black and white experimental points does not seem to be statistically compatible with zero.

Figure S9: Really hard to understand the meaning of colors in part (b).

---

## Author Comment (AC2) · 20 Apr 2018

We appreciate the very positive comments by Referee #1. We addressed his/her comments as outlined below. [1.1] Referee comment: In section 1) Section 2.5, Fig S2 The comparison between the gravimetric and OPS derived aerosol concentrations is made for periods in dry season, where majority of the coarse mode particles is expected to be of primary biogenic origin with densities close to 1g/cm3. Would it be possible to compare the OPS derived and gravimetric mass concentrations also for the wet season or specific LRT episodes? Or at least estimate the implications of the densities of both major contributors to the long range transported coarse aerosol - dust and sea

salt - being substantially higher than 1 g/cm3? Author Response: Although a gravimetric validation of the OPS results during the wet season and, in particular during the LRT episodes, would have been very useful for the study, no gravimetric results were available for these periods, unfortunately. We have added the following statement to the experimental part (page 9, line 22) to clarify this explicitly: "No corresponding gravimetric results were available for the wet season and LRT episodes." With respect to the requested estimate of the implications of substantially higher densities for e.g. Saharan dust or sea salt for the OPS-based aerosol mass retrieval, we added the following statement in page 9, line 27: "This implies that during substantial influence of dust and/or sea salt, which are characterized by higher aerosol particle densities (∼2 g cm-3, see table S1), no one-to-one agreement between OPS-based mass retrieval and gravimetry can be expected. In fact, the OPS-derived aerosol mass would underestimate the actual mass. The influence of different densities on the aerosol mass concentration is addressed later in this study in more detail (refer to Table 2)." [1.2] Referee comment: 2) Page 15, lines 16-29. Is the precipitation along the back trajectories considered at ground level or 3D? Do you take into account the option of the dust plume transport above the rain clouds followed by downwards mixing? Author Response: The referee brings up important aspects and we agree that this is not sufficiently clarified in the manuscript. With respect to the first question (precipitation at ground level or 3D): The HYSPLIT model provides the precipitation in the grid cells independent of altitude and, thus, not in 3D. To clarify this aspect, we have added the following statement in the experimental part (page 10, line 28): "Note that the HYSPLIT precipitation output is provided "at the grid cell where the trajectory is located and does not depend on the cloud value at the height of the trajectory" (comment by G. Rolph at https://hysplitbbs.arl.noaa.gov/viewtopic.php?t=577, last accessed 15 Mar 2018)." Furthermore, a second aspect (dust plumes being transported over the rain clouds), we modified the following original paragraph (page 15, line 16): " Relating to (3): Wet deposition is the dominant aerosol loss mechanism in tropical latitudes because of their intense precipitation (Huang et al., 2009; Martin et al., 2010a; Abdelkader et al.,

2017). According to Fig. 1c, comparatively small scavenging rates are expected for most of the dust's transatlantic passage (i.e., north of 03° N), whereas precipitation rates (on average) increase instantaneously when the air masses meet the ITCZ rain belt. As a measure for the extent of scavenging rates that the air masses experience in the NE Basin, we calculated the cumulative precipitation along the 3-day BT tracks, PBT , (shown as daily averages). In other words, the intense precipitation in the NE Basin defined if and to what extent the LRT plumes reached the ATTO region. The PBT time series shown in Fig. 2d and its comparison with M1-10 in Fig. 2e clearly underlines this relationship: Virtually all M1-10 pulses correspond with relative minima in the PBT time series. This shows, expectedly, that the dust burden that arrived at the ATTO site was inversely related to the cumulative amount of rain that the corresponding air masses received. In other words, only dust plumes that survived the intense rain-related scavenging had a chance to arrive in the ATTO region. Good examples for this relationship (among many others) are the dust pulses around 18 February 2014 and 06 April 2015." to the following revised version to explicitly address and clarify the aspects brought up be the referee: "Relating to (3): Wet deposition is the dominant aerosol loss mechanism in tropical latitudes because of their intense precipitation (Huang et al., 2009; Martin et al., 2010a; Abdelkader et al., 2017). According to Fig. 1c, comparatively small scavenging rates are expected for most of the dust's transatlantic passage (i.e., north of 03° N), whereas precipitation rates (on average) increase instantaneously when the air masses meet the ITCZ rain belt. A direct comparison of the cumulative precipitation, PBT, for 3-day vs. 9-day BTs shows that most of the rain (on average ~75 %, see Fig. S6) occurred during the last 3 days of the air mass journey, underlining that the region of the ITCZ belt is most important for aerosol wet deposition. Along these lines, the daily averages of PBT along the 3-day BT tracks represent a measure for the extent of scavenging rates that the air masses experience in the NE Basin. The intense precipitation in the NE Basin defines if and to what extent the LRT plumes reached the ATTO region. The PBT time series shown in Fig. 2d and its comparison with M1-10 in Fig. 2e clearly underlines this relationship: Virtually all

[Figure]

M1-10 pulses correspond with relative minima in the PBT time series and vice versa. This shows, expectedly, that the dust burden arriving at ATTO was inversely related to the cumulative amount of rain that the corresponding air masses experienced. In other words, only dust plumes that survived the intense rain-related scavenging had a chance to arrive in the ATTO region. Good examples for this relationship (among many others) are the dust pulses around 18 Feb 2014 and 06 Apr 2015. Note that the HYSPLIT precipitation output, which is calculated per grid cell, does not depend on altitude and cloud cover (see sect. 2.7). Thus, this analysis does not exclude that dust is transported at high altitudes over the precipitating clouds and mixed downwards in the ATTO region. However, the clear inverse relationship between the PBT and M1-10 variability underlines empirically that PBT can be used as a simple but reliable proxy for the extent of rain-related scavenging." [1.3] Referee comment: Page 3, line 10 – correct to "focused on " Author Response: Thanks for pointing out this typo. We have changed it from "focused of" to "focused on". [1.4] Referee comment: Page 15, line 9 – correct "this study clearly shows" Author Response: We also corrected this typo from "this study clearly show" to "this study clearly shows".

[Figure]

**Figure S6.** Scatter plot of cumulative precipitation, $P_{BT}$, (daily averages) along 3-day vs. 9-day back trajectories. Trajectories within the time period from 01 Jan 2014 to 31 Dec 2014 are analyzed here. Comparison shows that (on average) 75 % of the rain that the transported air parcels receive occurs during the last three days on their way to ATTO.

**Fig. 1.**

---

## Author Comment (AC3) · 20 Apr 2018

We thank Referee #2 for all the comments and suggestions that indeed will helped us to improve the manuscript. The referees' comments and our responses are outlined in detail below: [1.1] Referee comment: For most of figures: the authors preferred to differentiate curves in plots using different colors. There are recommendations to avoid this approach due to color-blinded readers. So, whenever possible, change and prefer figures using different gray tones and different line patterns (dot, crosses, diamonds, trace-dot, etc.). This is the case of the following figures in the manuscript: S1, S2a and S2b, S3, S4, S5, S6 S9, S12; 1, 2, 4a (colors), 4b-d (symbols are equal), 6, 7, 8 (use

different symbols), 9, 10e-f. Reference: https://www.nature.com/articles/nmeth.1618 Author Response: Thanks for pointing this out. We have modified the figures whenever possible. To keep the manuscript consistent with C. Pöhlker et al. (2018) we prefer to keep the original color code of the back trajectory clusters. In figures 4, 6, 8, S3, S7 (formerly S6), and S10 (formerly S9) different marker types were used. The colors in figures 1 and 7 were modified. [1.2] Referee comment: 2) - C. Pohlker et al. (2017) is cited several times in the text and is related to important features of this study. However, it is in reference list as "to be submitted". It is not correct to use so often a paper that is not even submitted. The same holds for Saturno et al. (2017) despite it is much less cited than C. Pohlker et al. 2017 Author Response: The studies have been submitted in the meantime and corresponding references have been updated as follows: - Saturno, J., Ditas, F., Penning de Vries, M., Holanda, B. A., Pöhlker, M. L., Carbone, S., Walter, D., Bobrowski, N., Brito, J., Chi, X., Gutmann, A., Hrabe de Angelis, I., Machado, L. A. T., Moran-Zuloaga, D., Rüdiger, J., Schneider, J., Schulz, C., Wang, Q., Wendisch, M., Artaxo, P., Wagner, T., Pöschl, U., Andreae, M. O., and Pöhlker, C.: African volcanic emissions influencing atmospheric aerosol particles over the Amazon rain forest, Atmos. Chem. Phys. Discuss., 2017, 1-32, 10.5194/acp-2017-1152, 2017a. - Saturno, J., Holanda, B. A., Pöhlker, C., Ditas, F., Wang, Q., Moran-Zuloaga, D., Brito, J., Carbone, S., Cheng, Y., Chi, X., Ditas, J., Hoffmann, T., Hrabe de Angelis, I., Könemann, T., Lavrič, J. V., Ma, N., Ming, J., Paulsen, H., Pöhlker, M. L., Rizzo, L. V., Schlag, P., Su, H., Walter, D., Wolff, S., Zhang, Y., Artaxo, P., Pöschl, U., and Andreae, M. O.: Black and brown carbon over central Amazonia: Long-term aerosol measurements at the ATTO site, Atmos. Chem. Phys. Discuss., 2017, 1-57, 10.5194/acp-2017-1097, 2017b. - Pöhlker, C., Walter, D., Paulsen, H., Könemann, T., Rodríguez-Caballero, E., Moran-Zuloaga, D., Brito, J., Carbone, S., Degrendele, C., Després, V., Ditas, F., Holanda, B., Kaiser, J. W., Lammel, G., Lavric, J. V., Ming, J., Pickersgill, D., Pöhkler, M., Praß, M., Ruckteschler, N., Saturno, J., Sörgel, M., Wang, Q., Weber, B., Wolff, S., Artaxo, P., Pöschl, U., and Andreae, M. O.: Land cover and its transformation in the backward trajectory footprint of the Amazon Tall Tower Observatory, Atmospheric Chemistry and Physics, acp-2018-323, submitted, 2018. [1.3] Referee comment: - Change "Amazonian" to "amazonian" (first letter lower case) in the whole text. Amazonian is an adjective, not a location. Amazon is a location. Author Response: In numerous landmark papers focusing on Amazon research, "Amazonian" is used capitalized (e.g., Cochrane and Laurance, 2008; Martin et al., 2010; Davidson et al., 2012; Andreae et al., 2015). We are convinced that this is correct. If not, it will be corrected by ACP's copyeditor. [1.4] Referee comment: - p.5 lines 1 and 5: the author refers to "winter dust plumes". It is confusing because the manuscript is dealing with a phenomenon that happens in both hemispheres. Winter in one hemisphere is summer in the other. Instead use the name of specific months you are referring. Another option, use wet or dry season (with respect to Central Amazonia) whose months are well known. Author Response: Agree. We have replaced "winter dust plumes" by "the wet season dust plumes". [1.5] Referee comment: - p.5 lines 34 and 36: repeated to close in the text the expression "whether and to what extent". Author Response: Agree. We deleted the "whether and" and replaced by "Furthermore, to what extent". [1.6] Referee comment: - p.13 line 28: "NE basin" to "NE Amazon Basin". Author Response: Done. "Amazon" has been added. [1.7] Referee comment: - p.15 line 12: P_BT was not defined before. Author Response: We have added the following sentence to the experimental section (p. 10, line 13) to define PBT: "Furthermore, the cumulative precipitation along the back trajectory tracks, PBT, has been calculated based on the HYSPLIT model output." [1.8] Referee comment: - p.16 last line: there is no meaning in M_BCe = 0.02 +- 0.03. This discrepancy is due to the use of Standard Deviation in non-normal data. Use median and Interquartile Range to avoid negative values in the confidence interval. Author Response: We have replace the mean +/- standard deviation by median with interquartile range (difference between 75 % and 25 % percentiles) throughout the text. Accordingly, the original paragraph in Sect. 3.1, on page 16: "... the wet season shows clean background concentrations (i.e., $N_{total}$ = 336 $\pm$ 209 cm-3 and $M_{BCe}$ = 0.02 $\pm$ 0.03 $\mu$g m-3, mean $\pm$ 1 std. dev.), whereas highest concentration levels occur in the dry season (i.e., $N_{total}$ = 1508 $\pm$ 785 cm-3 and

MBCe = 0.35 ± 0.20 $\mu$g m-3). The transitions periods represent an intermediate state in between these extremes (i.e., Ntotal = 790 ± 547 cm-3 and MBCe = 0.17 ± 0.20 $\mu$g m-3). During the LRT season, we observed a clear MBCe enhancement in comparison to the wet season background (i.e., MBCe = 0.17 ± 0.15 $\mu$g m-3 vs. MBCe = 0.02 ± 0.03 $\mu$g m-3), ..." has been replaced by "... the wet season shows clean background concentrations [i.e., median with interquartile range, IQR (25th–75th percentiles): Ntotal = 283 (197–420) cm-3 and MBCe = 0.02 (0.02– 0.04) $\mu$g m-3], whereas highest concentration levels occur in the dry season [i.e., Ntotal = 1337 (1021–1776) cm-3 and MBCe = 0.30 (0.21–0.46) $\mu$g m-3]. The transitions periods represent an intermediate state in between these extremes [i.e., Ntotal = 663 (448–963) cm-3 and MBCe = 0.10 (0.05–0.20) $\mu$g m-3]. During the LRT season, we observed a clear MBCe enhancement in comparison to the wet season background [i.e., MBCe = 0.14 (0.05–0.24) $\mu$g m-3 vs. MBCe = 0.02 (0.02–0.04) $\mu$g m-3] ..." Also in section 3.2, on page 17: "The N1-10 and M1-10 levels show a modest increases from the wet season (N1-10 = 0.42 ± 0.34 cm-3, M1-10 = 4.04 ± 2.72 $\mu$g m-3) over the transition periods (N1- 10 = 0.81 ± 0.75 cm-3, M1-10 = 5.24 ± 3.46 $\mu$g m-3) to the dry season (N1-10 = 1.15 ± 0.81 cm-3, M1-10 = 6.47 ± 2.69 $\mu$g m-3). The highest N1-10 and M1-10 levels clearly occurred during African LRT influence (N1-10 = 2.03 ± 1.87 cm-3, M1-10 = 11.28 ± 9.05 $\mu$g m-3)." has been updated to: "The wet season [N1-10 = 0.3 (0.2-0.5) cm-3, M1-10 = 3.5 (2.2-5.4) $\mu$g m-3] over the transition periods [N1-10 = 0.7 (0.4-1.0) cm-3, M1-10 = 4.9 (3.4-6.5) $\mu$g m-3]; meanwhile during the dry season [N1-10 = 1.1 (0.8-1.4) cm-3, M1-10 = 6.4 (4.9-7.8) $\mu$g m-3]. The highest concentrations for N1-10 and M1-10 levels clearly occurred during African LRT influence [N1-10 = 1.5 (0.6-2.8) cm- 3, M1-10 = 9.1 (5.2-14.2) $\mu$g m-3] respectively" [1.9] Referee comment: - p.19 lines 18-20: "We propose that the results obtained here for the year 2014 can be regarded as representative for a typical dust deposition scenario in the Amazon region, since 2014 was generally an 'average' year without 20 pronounced precipitation and circulation anomalies (M. Pöhlker et al., 2016; C. Pöhlker et al., 2017)." Âż>To state that 2014 was an "average year" is a strong affirmation, and should not use a "to be submitted" reference for such an

[Figure]

assumption. Author Response: The fact, that the year 2014 was an "average year" in terms of precipitation is shown and discussed in detail in M. Pöhlker et al., (2016). The manuscript by C. Pöhlker et al. (2018), which provides further details on hydrology and circulation patterns, has been submitted in the meantime and the following reference has been added: - Pöhlker, C., Walter, D., Paulsen, H., Könemann, T., Rodríguez-Caballero, E., Moran-Zuloaga, D., Brito, J., Carbone, S., Degrendele, C., Després, V., Ditas, F., Holanda, B., Kaiser, J. W., Lammel, G., Lavric, J. V., Ming, J., Pickersgill, D., Pöhkler, M., Praß, M., Ruckteschler, N., Saturno, J., Sörgel, M., Wang, Q., Weber, B., Wolff, S., Artaxo, P., Pöschl, U., and Andreae, M. O.: Land cover and its transformation in the backward trajectory footprint of the Amazon Tall Tower Observatory, Atmospheric Chemistry and Physics, acp-2018-323, submitted, 2018. [1.10] Referee comment: References: Rizzolo et al. (2016) was already published as final revised article in ACP: see https://www.atmos-chem-phys.net/17/2673/2017/acp-17-2673-2017.pdf and update it. Author Response: Thanks for pointing this out. The reference has been update: - Rizzolo, J. A., Barbosa, C. G. G., Borillo, G. C., Godoi, A. F. L., Souza, R. A. F., Andreoli, R. V., Manzi, A. O., Sá, M. O., Alves, E. G., Pöhlker, C., Angelis, I. H., Ditas, F., Saturno, J., Moran-Zuloaga, D., Rizzo, L. V., Rosário, N. E., Pauliquevis, T., Santos, R. M. N., Yamamoto, C. I., Andreae, M. O., Artaxo, P., Taylor, P. E., and Godoi, R. H. M.: Soluble iron nutrients in Saharan dust over the central Amazon rainforest, Atmos. Chem. Phys., 17, 2673- 2687, 10.5194/acp-17-2673-2017, 2017. [1.11] Referee comment: - Figure 2: it contains a lot of information in a single figure and it is somehow confusing to understand author's analysis and discussions due to the small size of curves. Improve it separating in different figures with larger sizes. This figure also cites figure S4. To understand the figure it is mandatory to the reader to see supplement. So, it should not be in the supplement but in the main text. The same holds for Figure 3, 4, 9, 14. Also, (c) lacks BC equivalent in legend. Author Response: The Fig. 2 underwent various iterations and changes in the course of the preparation of the manuscript. The main purpose of this figure is to directly compare the variability of the key time series. We are not sure if this works well after splitting the figure. Regarding the reference to

Fig. S4, we added a legend into Fig. 2, which provides the essential information (wind directions of back trajectories). Thus, it is not mandatory any more to refer to the supplement. The BC equivalent has been added to the legend in Fig. 2c. [1.12] Referee comment: - Figure 4: (a) very confusing to understand the actual meaning of colors in the figure. (c) and (e): change the experimental point symbol for the sake of clarity Author Response: We have changed the markers as requested. We also added a legend that specifies the back trajectory cluster colors. [1.13] Referee comment: - Figure 5: here you explain M_BCe, but it was previously used in Fig. 4 without any explanation. So, move it to previous figure. Author Response: We specified the pollution tracer also in the caption of Fig. 4 as follows: "Pollution tracers BCe mass concentration, MBCe, and carbon monoxide mole fraction, cCO." [1.14] Referee comment: - Figure 8: why some points show negative slope, with slope +- uncertainty not compatible with zero? Author Response: We have checked all six data points with a negative M10-1/MBCe slope carefully. It turned out that three data points (i.e., the two lowest of 2014 and the lowest of 2015) have data interruptions of the MBCe and M10-1 time series. It appears that the negative slopes resulted from the data gaps. These data points were removed from Fig. 8. The other three data points appear to be true outliers. In these cases, the MBCe was exceptionally low. These data points were left in Fig. 8. [1.15] Referee comment: - Figure 9: in the upper legend it cites Moran et al. (in preparation). Not adequate. Author Response: The reference has been removed. [1.16] Referee comment: - Figure 11: include in caption the meaning of gray areas in the map. Author Response: For clarification, the following sentence has been added to the captions of Fig. 11 and Fig. S10: "The gray areas represent pixels with no satellite data for the corresponding time periods. " [1.17] Referee comment: - Figure 12: in the caption: "are show overlay" > "are shown overlay"... and (b) actually refers to figure (d). Author Response: True. The typos have been corrected. [1.18] Referee comment: - Figure 13, in caption: did you mean "NCEP Reanalysis" when writing "NCEP satellite"? Author Response: Correct – thanks. We have changes "NCEP satellite" to "NCEP reanalysis". [1.19] Referee comment: Supplementary material: The manuscript cites too often figures containing

in supplementary material. If a figure has to be cited frequently and it is important to the actual comprehension of the article context of the article it should be moved to the main text. This is the case of figures S4, S6 and S9, which should be inserted in the principal text. Author Response: In general, we tried to put as many non-essential figures to the supplement as possible to keep the main text (which is already rather long) as short and concise as possible. Referring to the suggestion by the referee: A legend that specifies the color code in Fig. S4 has been added to the main text Figure 2, 4, and 9. Accordingly, it should not be necessary any more to refer to Fig. S4. The Fig. S6 is mentioned in the main text once and is not needed to follow the argumentation. Figure S9 is the overview figure of the second case study. The key results are discussed by means of the first case study, which is entirely placed in the main text. Case study 2 has been added to the supplement since it does not provide new aspects, but rather broadens the data basis and underlines certain aspects for readers that are interested in further details. Accordingly, we are convinced that all figures that are essential to follow the argumentation have been placed in the main text. [1.20] Refereecomment:FigureS2:atfigure(c)thefittingseemstohaveforcedparametera=0.It should be explained and/or justified it. Visual (separately) inspection of black and white experimental points does not seem to be statistically compatible with zero. Author Response: That is correct, the correlation has been forced through zero, mainly due to the rather small number of data point in the correlation plots. Moreover, both techniques have been validated to deliver zero in the absence of aerosol particles, which emphasizes that no axis intercept is expected. We added the following statement to the caption of Figure S2: "Linear regression fits (forced through zero) for both periods in (c) confirm overall agreement of both techniques." [1.21] Referee comment: Figure S9: Really hard to understand the meaning of colors in part (b). Author Response: We have added a legend (similar to Figures 2, 4, and 9) that specifies the wind directions of the backward trajectories and may help to increase readability.

---

## Author Comment (AC4) · 9 May 2018

Response to the referees (Daniel Moran-Zuloaga et al., Long-Term study on coarse mode aerosols in the Amazon rain forest with the frequent intrusion of Saharan dust plumes, ACP-2017-1043)

We appreciate the very positive comments by Referee #1. We addressed his/her comments as outlined below.

[1.1]   Referee comment: In section 1) Section 2.5, Fig S2 The comparison between the gravimetric and OPS derived aerosol concentrations is made for periods in dry season, where majority of the coarse mode particles is expected to be of primary biogenic origin with densities close to 1g/cm3. Would it be possible to compare the OPS derived and gravimetric mass concentrations also for the wet season or specific LRT episodes? Or at least estimate the implications of the densities of both major contributors to the long range transported coarse aerosol - dust and sea salt - being substantially higher than 1 g/cm3?

Author Response: Although a gravimetric validation of the OPS results during the wet season and, in particular during the LRT episodes, would have been very useful for the study, no gravimetric results were available for these periods, unfortunately. We have added the following statement to the experimental part (page 9, line 22) to clarify this explicitly:

> "No corresponding gravimetric results were available for the wet season and LRT episodes."

With respect to the requested estimate of the implications of substantially higher densities for e.g. Saharan dust or sea salt for the OPS-based aerosol mass retrieval, we added the following statement in page 9, line 27:

> "This implies that during substantial influence of dust and/or sea salt, which are characterized by higher aerosol particle densities ($\sim 2$ g cm$^{-3}$, see table S1), no one-to-one agreement between OPS-based mass retrieval and gravimetry can be expected. In fact, the OPS-derived aerosol mass would underestimate the actual mass. The influence of different densities on the aerosol mass concentration is addressed later in this study in more detail (refer to Table 2)."

[1.2]   Referee comment: 2) Page 15, lines 16-29. Is the precipitation along the back trajectories considered at ground level or 3D? Do you take into account the option of the dust plume transport above the rain clouds followed by downwards mixing?

Author Response: The referee brings up important aspects and we agree that this is not sufficiently clarified in the manuscript.

With respect to the first question (precipitation at ground level or 3D): The HYSPLIT model provides the precipitation in the grid cells independent of altitude and, thus, not in 3D. To clarify this aspect, we have added the following statement in the experimental part (page 10, line 28):

> "Note that the HYSPLIT precipitation output is provided "at the grid cell where the trajectory is located and does not depend on the cloud value at the height of the trajectory" (comment by G. Rolph at https://hysplitbbs.arl.noaa.gov/viewtopic.php?t=577, last accessed 15 Mar 2018)."

Furthermore, a second aspect (dust plumes being transported over the rain clouds), we modified the following original paragraph (page 15, line 16):

> "
>
> Relating to (**3**): Wet deposition is the dominant aerosol loss mechanism in tropical latitudes because of their intense precipitation (Huang et al., 2009; Martin et al., 2010a; Abdelkader et al., 2017). According to Fig. 1c, comparatively small scavenging rates are expected for most of the dust's transatlantic passage (i.e., north of 03° N), whereas precipitation rates (on average) increase instantaneously when the air masses meet the ITCZ rain belt. As a measure for the

extent of scavenging rates that the air masses experience in the NE Basin, we calculated the cumulative precipitation along the 3-day BT tracks, $P_{BT}$ , (shown as daily averages). In other words, the intense precipitation in the NE Basin defined if and to what extent the LRT plumes reached the ATTO region. The $P_{BT}$ time series shown in Fig. 2d and its comparison with $M_{1\text{-}10}$ in Fig. 2e clearly underlines this relationship: Virtually all $M_{1\text{-}10}$ pulses correspond with relative minima in the $P_{BT}$ time series. This shows, expectedly, that the dust burden that arrived at the ATTO site was inversely related to the cumulative amount of rain that the corresponding air masses received. In other words, only dust plumes that survived the intense rain-related scavenging had a chance to arrive in the ATTO region. Good examples for this relationship (among many others) are the dust pulses around 18 February 2014 and 06 April 2015."

to the following revised version to explicitly address and clarify the aspects brought up be the referee:

"Relating to (3): Wet deposition is the dominant aerosol loss mechanism in tropical latitudes because of their intense precipitation (Huang et al., 2009; Martin et al., 2010a; Abdelkader et al., 2017). According to Fig. 1c, comparatively small scavenging rates are expected for most of the dust's transatlantic passage (i.e., north of 03° N), whereas precipitation rates (on average) increase instantaneously when the air masses meet the ITCZ rain belt. A direct comparison of the cumulative precipitation, $P_{BT}$, for 3-day vs. 9-day BTs shows that most of the rain (on average ~75 %, see Fig. S6) occurred during the last 3 days of the air mass journey, underlining that the region of the ITCZ belt is most important for aerosol wet deposition. Along these lines, the daily averages of $P_{BT}$ along the 3-day BT tracks represent a measure for the extent of scavenging rates that the air masses experience in the NE Basin. The intense precipitation in the NE Basin defines if and to what extent the LRT plumes reached the ATTO region. The $P_{BT}$ time series shown in Fig. 2d and its comparison with $M_{1\text{-}10}$ in Fig. 2e clearly underlines this relationship: Virtually all $M_{1\text{-}10}$ pulses correspond with relative minima in the $P_{BT}$ time series and vice versa. This shows, expectedly, that the dust burden arriving at ATTO was inversely related to the cumulative amount of rain that the corresponding air masses experienced. In other words, only dust plumes that survived the intense rain-related scavenging had a chance to arrive in the ATTO region. Good examples for this relationship (among many others) are the dust pulses around 18 Feb 2014 and 06 Apr 2015. Note that the HYSPLIT precipitation output, which is calculated per grid cell, does not depend on altitude and cloud cover (see sect. 2.7). Thus, this analysis does not exclude that dust is transported at high altitudes over the precipitating clouds and mixed downwards in the ATTO region. However, the clear inverse relationship between the $P_{BT}$ and $M_{1\text{-}10}$ variability underlines empirically that $P_{BT}$ can be used as a simple but reliable proxy for the extent of rain-related scavenging."

[1.3]  Referee comment: Page 3, line 10 – correct to "focused on "

Author Response: Thanks for pointing out this typo. We have changed it from "focused of" to "focused on".

[1.4]  Referee comment: Page 15, line 9 – correct "this study clearly shows"

Author Response: We also corrected this typo from "this study clearly show" to "this study clearly shows".

---

## Author Comment (AC5) · 9 May 2018

Response to the referees (Daniel Moran-Zuloaga et al., Long-Term study on coarse mode aerosols in the Amazon rain forest with the frequent intrusion of Saharan dust plumes, ACP-2017-1043)

We thank Referee #2 for all the comments and suggestions that indeed will helped us to improve the manuscript. The referees' comments and our responses are outlined in detail below:

[1.1]    Referee comment: For most of figures: the authors preferred to differentiate curves in plots using different colors. There are recommendations to avoid this approach due to color-blinded readers. So, whenever possible, change and prefer figures using different gray tones and different line patterns (dot, crosses, diamonds, trace-dot, etc.). This is the case of the following figures in the manuscript: S1, S2a and S2b, S3, S4, S5, S6 S9, S12; 1, 2, 4a (colors), 4b-d (symbols are equal), 6, 7, 8 (use different symbols), 9, 10e-f.
Reference: https://www.nature.com/articles/nmeth.1618

Author Response: Thanks for pointing this out. We have modified the figures whenever possible. To keep the manuscript consistent with C. Pöhlker et al. (2018) we prefer to keep the original color code of the back trajectory clusters. In figures 4, 6, 8, S3, S7 (formerly S6), and S10 (formerly S9) different marker types were used. The colors in figures 1 and 7 were modified.

[1.2]    Referee comment: 2) - C. Pohlker et al. (2017) is cited several times in the text and is related to important features of this study. However, it is in reference list as "to be submitted". It is not correct to use so often a paper that is not even submitted. The same holds for Saturno et al. (2017) despite it is much less cited than C. Pohlker et al. 2017

Author Response: The studies have been submitted in the meantime and corresponding references have been updated as follows:
-   Saturno, J., Ditas, F., Penning de Vries, M., Holanda, B. A., Pöhlker, M. L., Carbone, S., Walter, D., Bobrowski, N., Brito, J., Chi, X., Gutmann, A., Hrabe de Angelis, I., Machado, L. A. T., Moran-Zuloaga, D., Rüdiger, J., Schneider, J., Schulz, C., Wang, Q., Wendisch, M., Artaxo, P., Wagner, T., Pöschl, U., Andreae, M. O., and Pöhlker, C.: African volcanic emissions influencing atmospheric aerosol particles over the Amazon rain forest, Atmos. Chem. Phys. Discuss., 2017, 1-32, 10.5194/acp-2017-1152, 2017a.

-   Saturno, J., Holanda, B. A., Pöhlker, C., Ditas, F., Wang, Q., Moran-Zuloaga, D., Brito, J., Carbone, S., Cheng, Y., Chi, X., Ditas, J., Hoffmann, T., Hrabe de Angelis, I., Könemann, T., Lavrič, J. V., Ma, N., Ming, J., Paulsen, H., Pöhlker, M. L., Rizzo, L. V., Schlag, P., Su, H., Walter, D., Wolff, S., Zhang, Y., Artaxo, P., Pöschl, U., and Andreae, M. O.: Black and brown carbon over central Amazonia: Long-term aerosol measurements at the ATTO site, Atmos. Chem. Phys. Discuss., 2017, 1-57, 10.5194/acp-2017-1097, 2017b.

-   Pöhlker, C., Walter, D., Paulsen, H., Könemann, T., Rodríguez-Caballero, E., Moran-Zuloaga, D., Brito, J., Carbone, S., Degrendele, C., Després, V., Ditas, F., Holanda, B., Kaiser, J. W., Lammel, G., Lavric, J. V., Ming, J., Pickersgill, D., Pöhlker, M., Praß, M., Ruckteschler, N., Saturno, J., Sörgel, M., Wang, Q., Weber, B., Wolff, S., Artaxo, P., Pöschl, U., and Andreae, M. O.: Land cover and its transformation in the backward trajectory footprint of the Amazon Tall Tower Observatory, Atmospheric Chemistry and Physics, acp-2018-323, submitted, 2018.

[1.3]    Referee comment: - Change "Amazonian" to "amazonian" (first letter lower case) in the whole text. Amazonian is an adjective, not a location. Amazon is a location.

Author Response: In numerous landmark papers focusing on Amazon research, "Amazonian" is used capitalized (e.g., Cochrane and Laurance, 2008; Martin et al., 2010; Davidson et al., 2012; Andreae et al., 2015). We are convinced that this is correct. If not, it will be corrected by ACP's copyeditor.

[1.4]    Referee comment: - p.5 lines 1 and 5: the author refers to "winter dust plumes". It is confusing because the manuscript is dealing with a phenomenon that happens in both hemispheres. Winter in one

hemisphere is summer in the other. Instead use the name of specific months you are referring. Another option, use wet or dry season (with respect to Central Amazonia) whose months are well known.

Author Response: Agree. We have replaced "winter dust plumes" by "the wet season dust plumes".

[1.5]  Referee comment: - p.5 lines 34 and 36: repeated to close in the text the expression "whether and to what extent".

Author Response:  Agree. We deleted the "whether and" and replaced by "Furthermore, to what extent".

[1.6]  Referee comment:  - p.13 line 28: "NE basin" to "NE Amazon Basin".

Author Response: Done. "Amazon" has been added.

[1.7]  Referee comment: - p.15 line 12: P_BT was not defined before.

Author Response:  We have added the following sentence to the experimental section (p. 10, line 13) to define $P_{BT}$:

"Furthermore, the cumulative precipitation along the back trajectory tracks, $P_{BT}$, has been calculated based on the HYSPLIT model output."

[1.8]  Referee comment: - p.16 last line: there is no meaning in M_BCe = 0.02 +- 0.03. This discrepancy is due to the use of Standard Deviation in non-normal data. Use median and Interquartile Range to avoid negative values in the confidence interval.

Author Response: We have replace the mean +/- standard deviation by median with interquartile range (difference between 75 % and 25 % percentiles) throughout the text. Accordingly, the original paragraph in Sect. 3.1, on page 16:

"… the wet season shows clean background concentrations (i.e., $N_{total}$ = 336 ± 209 cm$^{-3}$ and $M_{BCe}$ = 0.02 ± 0.03 µg m$^{-3}$, mean ± 1 std. dev.), whereas highest concentration levels occur in the dry season (i.e., $N_{total}$ = 1508 ± 785 cm$^{-3}$ and $M_{BCe}$ = 0.35 ± 0.20 µg m$^{-3}$). The transitions periods represent an intermediate state in between these extremes (i.e., $N_{total}$ = 790 ± 547 cm$^{-3}$ and $M_{BCe}$ = 0.17 ± 0.20 µg m$^{-3}$). During the LRT season, we observed a clear $M_{BCe}$ enhancement in comparison to the wet season background (i.e., $M_{BCe}$ = 0.17 ± 0.15 µg m$^{-3}$ vs. $M_{BCe}$ = 0.02 ± 0.03 µg m$^{-3}$), …"

has been replaced by

"… the wet season shows clean background concentrations [i.e., median with interquartile range, IQR (25$^{th}$–75$^{th}$ percentiles): $N_{total}$ = 283 (197–420) cm$^{-3}$ and $M_{BCe}$ = 0.02 (0.02–0.04) µg m$^{-3}$], whereas highest concentration levels occur in the dry season [i.e., $N_{total}$ = 1337 (1021–1776) cm$^{-3}$ and $M_{BCe}$ = 0.30 (0.21–0.46) µg m$^{-3}$]. The transitions periods represent an intermediate state in between these extremes [i.e., $N_{total}$ = 663 (448–963) cm$^{-3}$ and $M_{BCe}$ = 0.10 (0.05–0.20) µg m$^{-3}$]. During the LRT season, we observed a clear $M_{BCe}$ enhancement in comparison to the wet season background [i.e., $M_{BCe}$ = 0.14 (0.05–0.24) µg m$^{-3}$ vs. $M_{BCe}$ = 0.02 (0.02–0.04) µg m$^{-3}$] …"

Also in section 3.2, on page 17:

"The $N_{1-10}$ and $M_{1-10}$ levels show a modest increases from the wet season ($N_{1-10}$ = 0.42 ± 0.34 cm$^{-3}$, $M_{1-10}$ = 4.04 ± 2.72 µg m$^{-3}$) over the transition periods ($N_{1-10}$ = 0.81 ± 0.75 cm$^{-3}$, $M_{1-10}$ = 5.24 ± 3.46 µg m$^{-3}$) to the dry season ($N_{1-10}$ = 1.15 ± 0.81 cm$^{-3}$,

$M_{1-10} = 6.47 \pm 2.69$ µg m$^{-3}$). The highest $N_{1-10}$ and $M_{1-10}$ levels clearly occurred during African LRT influence ($N_{1-10} = 2.03 \pm 1.87$ cm$^{-3}$, $M_{1-10} = 11.28 \pm 9.05$ µg m$^{-3}$)."

has been updated to:

"The wet season [$N_{1-10} = 0.3$ (0.2-0.5) cm$^{-3}$, $M_{1-10} = 3.5$ (2.2-5.4) µg m$^{-3}$] over the transition periods [$N_{1-10} = 0.7$ (0.4-1.0) cm$^{-3}$, $M_{1-10} = 4.9$ (3.4-6.5) µg m$^{-3}$]; meanwhile during the dry season [$N_{1-10} = 1.1$ (0.8-1.4) cm$^{-3}$, $M_{1-10} = 6.4$ (4.9-7.8) µg m$^{-3}$]. The highest concentrations for $N_{1-10}$ and $M_{1-10}$ levels clearly occurred during African LRT influence [$N_{1-10} = 1.5$ (0.6-2.8) cm$^{-3}$, $M_{1-10} = 9.1$ (5.2-14.2) µg m$^{-3}$] respectively"

[1.9]   Referee comment: - p.19 lines 18-20: "We propose that the results obtained here for the year 2014 can be regarded as representative for a typical dust deposition scenario in the Amazon region, since 2014 was generally an 'average' year without 20 pronounced precipitation and circulation anomalies (M. Pöhlker et al., 2016; C. Pöhlker et al., 2017)." »>To state that 2014 was an "average year" is a strong affirmation, and should not use a "to be submitted" reference for such an assumption.

Author Response: The fact, that the year 2014 was an "average year" in terms of precipitation is shown and discussed in detail in M. Pöhlker et al., (2016). The manuscript by C. Pöhlker et al. (2018), which provides further details on hydrology and circulation patterns, has been submitted in the meantime and the following reference has been added:

- Pöhlker, C., Walter, D., Paulsen, H., Könemann, T., Rodríguez-Caballero, E., Moran-Zuloaga, D., Brito, J., Carbone, S., Degrendele, C., Després, V., Ditas, F., Holanda, B., Kaiser, J. W., Lammel, G., Lavric, J. V., Ming, J., Pickersgill, D., Pöhkler, M., Praß, M., Ruckteschler, N., Saturno, J., Sörgel, M., Wang, Q., Weber, B., Wolff, S., Artaxo, P., Pöschl, U., and Andreae, M. O.: Land cover and its transformation in the backward trajectory footprint of the Amazon Tall Tower Observatory, Atmospheric Chemistry and Physics, acp-2018-323, submitted, 2018.

[1.10]   Referee comment: References: Rizzolo et al. (2016) was already published as final revised article in ACP: see https://www.atmos-chem-phys.net/17/2673/2017/acp-17-2673-2017.pdf and update it.

Author Response: Thanks for pointing this out. The reference has been update:

- Rizzolo, J. A., Barbosa, C. G. G., Borillo, G. C., Godoi, A. F. L., Souza, R. A. F., Andreoli, R. V., Manzi, A. O., Sá, M. O., Alves, E. G., Pöhlker, C., Angelis, I. H., Ditas, F., Saturno, J., Moran-Zuloaga, D., Rizzo, L. V., Rosário, N. E., Pauliquevis, T., Santos, R. M. N., Yamamoto, C. I., Andreae, M. O., Artaxo, P., Taylor, P. E., and Godoi, R. H. M.: Soluble iron nutrients in Saharan dust over the central Amazon rainforest, Atmos. Chem. Phys., 17, 2673-2687, 10.5194/acp-17-2673-2017, 2017.

[1.11]   Referee comment: - Figure 2: it contains a lot of information in a single figure and it is somehow confusing to understand author's analysis and discussions due to the small size of curves. Improve it separating in different figures with larger sizes.
This figure also cites figure S4. To understand the figure it is mandatory to the reader to see supplement. So, it should not be in the supplement but in the main text. The same holds for Figure 3, 4, 9, 14. Also, (c) lacks BC equivalent in legend.

Author Response: The Fig. 2 underwent various iterations and changes in the course of the preparation of the manuscript. The main purpose of this figure is to directly compare the variability of the key time series. We are not sure if this works well after splitting the figure. Regarding the reference to Fig. S4, we added a legend into Fig. 2, which provides the essential information (wind directions of back

trajectories). Thus, it is not mandatory any more to refer to the supplement. The BC equivalent has been added to the legend in Fig. 2c.

[1.12]   Referee comment: - Figure 4: (a) very confusing to understand the actual meaning of colors in the figure. (c) and (e): change the experimental point symbol for the sake of clarity

Author Response: We have changed the markers as requested. We also added a legend that specifies the back trajectory cluster colors.

[1.13]   Referee comment: - Figure 5: here you explain M_BCe, but it was previously used in Fig. 4 without any explanation. So, move it to previous figure.

Author Response: We specified the pollution tracer also in the caption of Fig. 4 as follows: "Pollution tracers $BC_e$ mass concentration, $M_{BCe}$, and carbon monoxide mole fraction, $c_{CO}$."

[1.14]   Referee comment: - Figure 8: why some points show negative slope, with slope +- uncertainty not compatible with zero?

Author Response: We have checked all six data points with a negative $M_{10\text{-}1}/M_{BCe}$ slope carefully. It turned out that three data points (i.e., the two lowest of 2014 and the lowest of 2015) have data interruptions of the $M_{BCe}$ and $M_{10\text{-}1}$ time series. It appears that the negative slopes resulted from the data gaps. These data points were removed from Fig. 8. The other three data points appear to be true outliers. In these cases, the $M_{BCe}$ was exceptionally low. These data points were left in Fig. 8.

[1.15]   Referee comment: - Figure 9: in the upper legend it cites Moran et al. (in preparation). Not adequate.

Author Response: The reference has been removed.

[1.16]   Referee comment: - Figure 11: include in caption the meaning of gray areas in the map.

Author Response: For clarification, the following sentence has been added to the captions of Fig. 11 and Fig. S10: "The gray areas represent pixels with no satellite data for the corresponding time periods. "

[1.17]   Referee comment: - Figure 12: in the caption: "are show overlay" > "are shown overlay"... and (b) actually refers to figure (d).

Author Response: True. The typos have been corrected.

[1.18]   Referee comment: - Figure 13, in caption: did you mean "NCEP Reanalysis" when writing "NCEP satellite"?

Author Response: Correct – thanks. We have changes "NCEP satellite" to "NCEP reanalysis".

[1.19]   Referee comment: Supplementary material: The manuscript cites too often figures containing in supplementary material. If a figure has to be cited frequently and it is important to the actual

comprehension of the article context of the article it should be moved to the main text. This is the case of figures S4, S6 and S9, which should be inserted in the principal text.

Author Response: In general, we tried to put as many non-essential figures to the supplement as possible to keep the main text (which is already rather long) as short and concise as possible. Referring to the suggestion by the referee: A legend that specifies the color code in Fig. S4 has been added to the main text Figure 2, 4, and 9. Accordingly, it should not be necessary any more to refer to Fig. S4. The Fig. S6 is mentioned in the main text once and is not needed to follow the argumentation. Figure S9 is the overview figure of the second case study. The key results are discussed by means of the first case study, which is entirely placed in the main text. Case study 2 has been added to the supplement since it does not provide new aspects, but rather broadens the data basis and underlines certain aspects for readers that are interested in further details. Accordingly, we are convinced that all figures that are essential to follow the argumentation have been placed in the main text.

 [1.20]   Referee comment: Figure S2: at figure (c) the fitting seems to have forced parameter a = 0. It should be explained and/or justified it. Visual (separately) inspection of black and white experimental points does not seem to be statistically compatible with zero.

Author Response: That is correct, the correlation has been forced through zero, mainly due to the rather small number of data point in the correlation plots. Moreover, both techniques have been validated to deliver zero in the absence of aerosol particles, which emphasizes that no axis intercept is expected. We added the following statement to the caption of Figure S2:
> "Linear regression fits (forced through zero) for both periods in (**c**) confirm overall agreement of both techniques."

[1.21]   Referee comment: Figure S9: Really hard to understand the meaning of colors in part (b).

Author Response: We have added a legend (similar to Figures 2, 4, and 9) that specifies the wind directions of the backward trajectories and may help to increase readability.